# The *Escherichia coli* replication initiator DnaA is titrated on the chromosome

Lorenzo Olivi [1] ✉, Stephan Köstlbacher [1], Christina Ludwig [2], Mees Langendoen[1], Nico J. Claassens [1], Thijs J. G. Ettema [1], John van der Oost [1], Pieter Rein ten Wolde [3], Johannes Hohlbein [4,5] ✉ & Raymond H. J. Staals [1] ✉

DNA replication initiation is orchestrated in bacteria by the replication initiator DnaA. Two models for regulation of DnaA activity in *Escherichia coli* have been proposed: the switch between an active and inactive form, and the titration of DnaA on the chromosome. Although proposed decades ago, experimental evidence of a titration-based control mechanism is still lacking. Here, we first identified a conserved high-density region of binding motifs near the origin of replication, an advantageous trait for titration of DnaA. We then investigated the mobility of DnaA by visualising single proteins inside single cells of wild-type and deletion mutants *E. coli* strains, while monitoring cellular size and DNA content. Our results indicate that the chromosome of *E. coli* controls the free amount of DnaA in a growth rate-dependent fashion. Moreover, they address long-standing questions on the relevance of DnaA titration in stabilising DNA replication by preventing re-initiation events during slow growth.

The correct coordination of DNA replication with cellular growth is crucial for any living organism[1–3]. Remarkably, the orchestration of these processes allows *Escherichia coli* to divide faster than the time it takes to replicate its chromosome[4–7]. This surprising ability was first explained in 1968 by a model stating that *E. coli* initiates one or more rounds of replication before the initial replicative event is completed[4]. Subsequently, an extension of the model introduced the concept of volume of initiation, a specific cellular size per origin of replication[8]. Since then, the coupling of DNA replication and cellular growth has been experimentally confirmed[6,7] and the volume of initiation has been identified as a tightly controlled parameters in the *E. coli* cell cycle[7,9,10].

At the molecular level, *E. coli* initiates DNA replication by unwinding the origin of replication (*oriC*) through the binding of the replication initiator protein DnaA[11]. The regulation and biochemistry of DnaA have been extensively studied[11,12], yet how *E. coli* achieves its characteristic DNA replication behaviour is still a matter of debate[1,11,13–15]. At the expression level, the *dnaA* gene is negatively autoregulated[16] and its near-balanced expression has been reported to

lead to stable intracellular concentrations of DnaA across the cell cycle[17]. At the same time, the population average concentration of DnaA increases at faster growth rates[10]. The DnaA protein binds 9 bp-long DNA sequences, termed DnaA boxes. Both high-affinity and low-affinity boxes exist, with the first class matching a consensus sequence (TTWTNCACA)[18] and the latter carrying multiple variations[19,20]. Two models have been proposed to clarify how the DnaA-mediated unwinding of *oriC* is regulated. First, the initiator titration model[21,22] dictates that DnaA is sequestered by hundreds of high-affinity DnaA boxes present on the chromosome of *E. coli*[14] (Fig. 1, top). Only after enough DnaA is accumulated to saturate these boxes, DnaA can bind the low-affinity sites present at *oriC*[19,23,24] and unwind it. With structural and biochemical characterisation of DnaA advancing, a second model emerged. This switch model (Fig. 1, bottom) is based on the finding that DnaA switches between an ATP-bound state, competent in origin unwinding, and an inactive, ADP-bound state[11]. The interconversion between states is regulated by several mechanisms. Newly synthesised DnaA proteins predominantly bind to ATP due to its higher abundance

[1]Laboratory of Microbiology, Wageningen University & Research, Wageningen, Netherlands. [2]Bavarian Center for Biomolecular Mass Spectrometry (Bay-BioMS), School of Life Sciences, Technical University of Munich, Munich, Germany. [3]Biochemical Networks Group, Institute AMOLF, Amsterdam, Netherlands. [4]Laboratory of Biophysics, Wageningen University & Research, Wageningen, Netherlands. [5]Microspectroscopy Research Facility, Wageningen University & Research, Wageningen, Netherlands. ✉e-mail: lorenzo.olivi@wur.nl; lorenzo.olivi.1994@gmail.com; johannes.hohlbein@wur.nl; raymond.staals@wur.nl

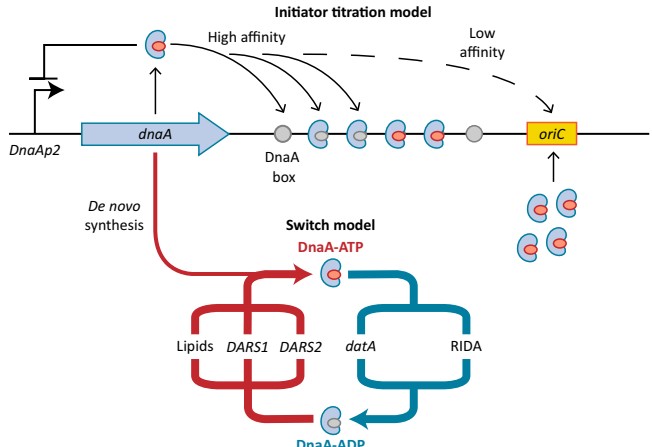

**Fig. 1 | The two models attempting to explain the control of DNA replication in *E. coli*.** DnaA negatively auto-regulates its own expression (left). Newly synthesised DnaA is bound by ATP. In the initiator titration model (top), the initiator protein is sequestered by high-affinity DnaA boxes throughout the chromosome. Only when all boxes are saturated, DnaA can bind to low-affinity boxes present on *oriC*, initiating DNA replication. In the switch model (bottom), DnaA is present in either an ATP-bound, active form, or an ADP-bound, inactive form. Several mechanisms are involved in the switch between the two forms. Once DnaA-ATP accumulates over a specific threshold, it can bind and unwind *oriC*.

in the cell compared to ADP[25]. Further, non-coding sequences called DnaA-reactivating sequences 1 and 2 (*DARS1* and *DARS2*) are involved in recruiting DnaA to nine particular DnaA boxes, where ADP is replaced with ATP[26]. At the same time, a similar re-activating mechanism has been proposed for interactions with acid phospholipids in vitro[27], although this mechanism still lacks confirmation in vivo. Conversely, the non-coding sequence *datA* recruits DnaA with five other DnaA boxes, where *datA*-mediated DnaA-ATP hydrolysis (DDAH) occurs[28]. Finally, the Hda protein associates with the β-clamp of DNA polymerase III during replication to stimulate hydrolysis of DnaA-bound ATP in a process termed regulatory inactivation of DnaA (RIDA)[29]. Both DnaA-ATP and DnaA-ADP have been shown to be competent in binding high-affinity boxes, whereas only DnaA-ATP can bind to low-affinity boxes[30], such as the ones present on *oriC*[19]. Therefore, the switch model dictates that it is the accumulation of specifically the DnaA-ATP form, competent in unwinding of *oriC*, that determines the moment of initiation of DNA replication.

Since the characterisation of DnaA-ADP/ATP interconversion, the switch model has largely replaced the initiator titration model in explaining the replicative behaviour of *E. coli*. Yet, *E. coli* strains carrying deletions of the switch control components *datA*, *DARS1*, *DARS2* and *hda*, either individually or combined, exhibit defects in initiation, but are still viable[17,31,32]. The viability of the quadruple mutant was attributed to the intrinsic ATPase ability of DnaA, while postulating that initiator titration could provide an additional stabilising effect[17]. Moreover, recent single-cell analysis of the oscillation of DnaA-ATP levels in *E. coli* showed that reaching a peak level of DnaA-ATP is a necessary but not sufficient condition to initiate DNA replication[33]. In light of these results, either the accumulation of DnaA-ATP or its binding to the low-affinity binding sites on *oriC* could be delayed by the titration of the initiator on the *E. coli* chromosome. In agreement with this, previous in silico work suggested that concerted DnaA interconversion and its titration on the chromosome can enhance the stability of *E. coli* cell cycles in all growth conditions[34,35], with a larger effect at slower growth rates[36].

The titration of DnaA on the *E. coli* chromosome was hypothesised more than 40 years ago[21] and has been considered as present in many recent modelling[34–36] and experimental studies[17,32,33]. Yet, there is still

no experimental evidence that *E. coli* can control the free concentration of DnaA via titration. To address this gap in knowledge, we first analysed the configuration of *E. coli* genome and showed that DnaA boxes are distributed favourably for titration. Specifically, DnaA boxes accumulate towards the origin of replication, a trait conserved within many *E. coli* strains as well as in *Salmonella enterica*. Revealing the presence of DnaA titration experimentally requires assessing the binding state of this protein in live cells. Previous biochemical assays already probed the interactions of DnaA with DNA[18,37,38], but they all required cell lysis. Additionally, earlier in vivo studies introduced titration sites via additional copies of *oriC*[39] or *datA*[40–42] to obtain insights on the effect of titration on DNA replication control. Yet, these studies were performed only under fast growth conditions and before *datA* had been identified as a DnaA-inactivating locus. Further, the additional titration sites were provided on plasmids, inevitably decoupling DnaA titration from DNA replication. Here, we overcome these limitations by directly probing interactions between DnaA and DNA in live *E. coli* cells via single-particle tracking photoactivatable localisation microscopy (sptPALM)[43]. We generated fusions of the native DnaA with the photoactivatable fluorescent protein PAmCherry2.1[44] in the wild-type *E. coli* MG1655 and in mutant strains in which *datA*, *DARS1* and *DARS2* were deleted either individually or together[45]. Building on our previously established experimental framework[44] (Supplementary Fig. 1), we grew all strains in constant optical density settings and monitored relevant cellular parameters (i.e., cell area and DNA content). By then performing sptPALM, we calculated the overall bound fraction of DnaA, as well as the mobility of the DnaA tracks generated throughout the cell cycle in different growth conditions. In this way, we provide experimental evidence that the *E. coli* chromosome is indeed capable of titrating DnaA, keeping its free fraction low in a growth rate-dependent fashion. Furthermore, our analysis suggests that DnaA titration increases in conditions in which more DnaA-ATP is present and decreases when the cell lacks DnaA reactivating power.

## Results

### DnaA boxes are preferentially accumulated near the origin of replication

A genomic configuration in which DnaA boxes are distributed preferentially towards *oriC* allows for a large fraction of sites to be replicated soon after the start of DNA replication and is thus ideal for optimal titration of DnaA. The hundreds of binding motifs present on the chromosome of *E. coli* are believed to be homogeneously distributed[12,18], yet this assumption has not been tested. We thus compared the number of DnaA boxes per position (radius of circular plot) in windows of 10 centisomes on the *E. coli* chromosome with the number expected when sampling from a uniform distribution (Fig. 2A). We used a list of motifs that includes both the consensus sequence of DnaA box (TTWTNCACA)[18], as well as other motifs functional in *datA*, *DARS1* and *DARS2* (HHMTHCWVH)[45]. We identified an extended region of significant overrepresentation of DnaA boxes between centisomes 98 and 5, specifically revealing a preferred enrichment of DnaA boxes around *oriC*. By evaluating the number of exact matches to permutated DnaA box motifs on the chromosome (Fig. 2A), we further rejected the hypothesis that gradients in genome composition (e.g., GC-content skew) caused the observed enrichment. The same conclusions were reached when using only the consensus sequence (Supplementary Fig. 2A) or when counting DnaA boxes in windows of either 2 or 6 centisomes (Supplementary Fig. 2B). Additionally, we tried excluding *oriC* to test whether its high density of DnaA boxes could bias our analysis. Even without considering the boxes present at the origin, the enrichment in the adjacent region persisted (Supplementary Fig. 2C).

The accumulation of DnaA boxes towards *oriC* was not a characteristic unique to *E. coli* MG1655. We plotted the density of DnaA

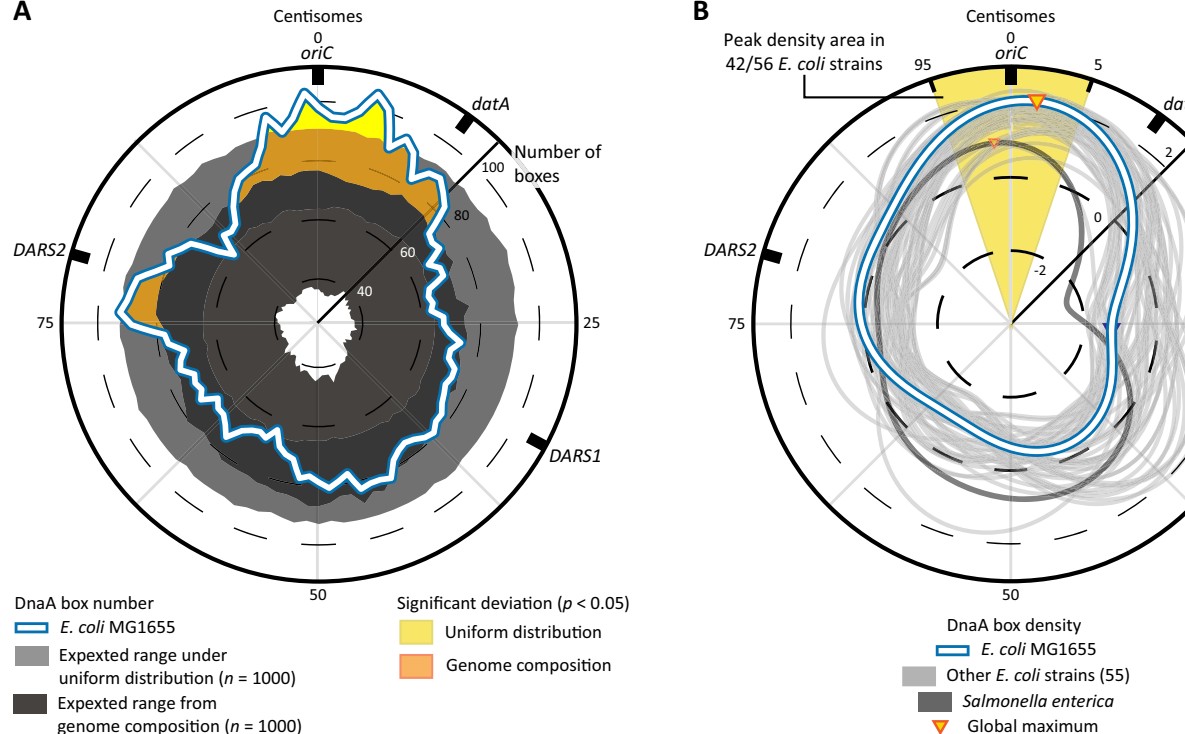

**Fig. 2 | Density peaks of DnaA boxes across the chromosome of *E. coli* strains.**
**A** Circular representation of the *E. coli* MG1655 chromosome, depicting DnaA box numbers at distinct positions. The actual numbers of DnaA boxes on the *E. coli* MG1655 chromosome is depicted as the white line and is compared to expected numbers ($5 \leq p \leq 95$) based on 1000 samples from a uniform distribution (light grey bands), and based on sequence matches to 1000 permutations of DnaA box sequences (dark grey band). Regions highlighted in yellow and orange indicate significant enrichment relative to the uniform null model and the sequence-permuted null model, respectively (one-sided permutation test, pseudo $p < 0.05$, unadjusted for multiple comparisons). The radius of the circle indicates the number of boxes per centisomes, counted within a 10 centisomes window. **B** Circular representation of DnaA box density across multiple *E. coli* genomes and one

singular *Salmonella enterica* chromosome, expressed as Z-scores. The density of *E. coli* MG1655 is depicted as the white line, while other 55 *E. coli* strains are in grey and *S. enterica*, considered as a distant relative species, is depicted in dark grey. For *E. coli* MG1655 and *S. enterica*, the global maximum is also plotted. The radius of the circle indicates the number of standard deviations away from the mean (Z-score) per centisome, counted within a 10 centisomes window. See Supplementary Fig. 3 for density plots of each individual *E. coli* strain. In all panels, the outer ring represents the relative distance from the origin of replication, expressed in centisomes and with the 0 at *oriC*. In all panels, we used a TTWTNCACA motif for the consensus DnaA box and a HHMTHCWVH motif to include the boxes present among *datA*, *DARS1* and *DARS2* in the analysis. Source data are provided as a Source Data file.

boxes across the chromosomes of 56 *E. coli* strains and obtained a Z-score (indicated from the radius of the circular plot) (Fig. 2B). As for our previous analysis, we included both consensus motifs and other types of boxes found in *datA, DARS1* and *DARS2*, for an average of $902 \pm 77$ boxes per *E. coli* genome (see Table S4 in Supplementary Information for strain-specific numbers). We found that most *E. coli* chromosomes ($n = 42$, 74% of all analysed strains) have their global density maximum of DnaA boxes within 5 centisomes of the *oriC*. Further expanding this analysis, we observed that 93% of the genomes had either a local or a global maximum within 10 centisomes from *oriC* (see also Supplementary Fig. 3), with this region having a median DnaA box density two standard deviations above the chromosomal mean (Z-score median = 2.1, interquartile range 1.8–2.2). Interestingly, a similar distribution pattern is also observed in the distant relative *Salmonella enterica* (Fig. 2B), suggesting an evolutionary conservation of this genomic arrangement beyond *E. coli*.

Our analysis highlights an accumulation of DnaA boxes near the origin of replication. The significant overrepresentation of box sequences compared to the genome composition further suggests positive selection for DnaA boxes on the chromosome (Supplementary Fig. 2D). Finally, the preferential accumulation of these motifs on the lagging strand of DNA replication (Supplementary Fig. 2E) hints at cooperation between initiator titration and the switch-related mechanism of RIDA[29], which functions through DNA polymerase III β-clamps[46] left on the lagging strands of replication[47]. Notably, the

abundance of DnaA boxes near *oriC* means that a considerable fraction of these motifs is replicated in the early stages of the DNA replication elongation step. As a result, both the total number of DnaA boxes and their concentration can quickly increase after initiation, a feature that could help initiator titration during fast growth[36].

## Growth rate-dependent titration of DnaA on the *E. coli* chromosome

After observing that DnaA boxes are preferentially arranged close to *oriC*, we moved on to seek experimental confirmation of the ability of the *E. coli* chromosome to titrate DnaA. To this end, we generated several fusions of DnaA in domain II[48,49] with photoactivatable or photoswitchable fluorescent proteins. The fusions were created by replacing amino acids 87-106[48,49] with the different fluorescent proteins (see Methods for details on the construction process). We ultimately selected PAmCherry2.1[44] for its low impact on DNA content (Supplementary Fig. 4B). Follow up characterisation confirmed that the *dnaA-PAmCherry2.1* mutation did not altered the cell area (Supplementary Fig. 4C) nor the DnaA expression levels compared to the wild-type background (Supplementary Fig. 4D). We additionally generated a plasmid carrying a copy of the *dnaAp* promoter driving transcription of the LacZ β-galactosidase. Previous studies have used the expression of LacZ as a proxy for the ratio of DnaA-ATP/DnaA-ADP in *E. coli* cells[17,50], based on the fact that DnaA-ATP is more efficient in repressing its own promoter[12,16]. Once again, we observed no

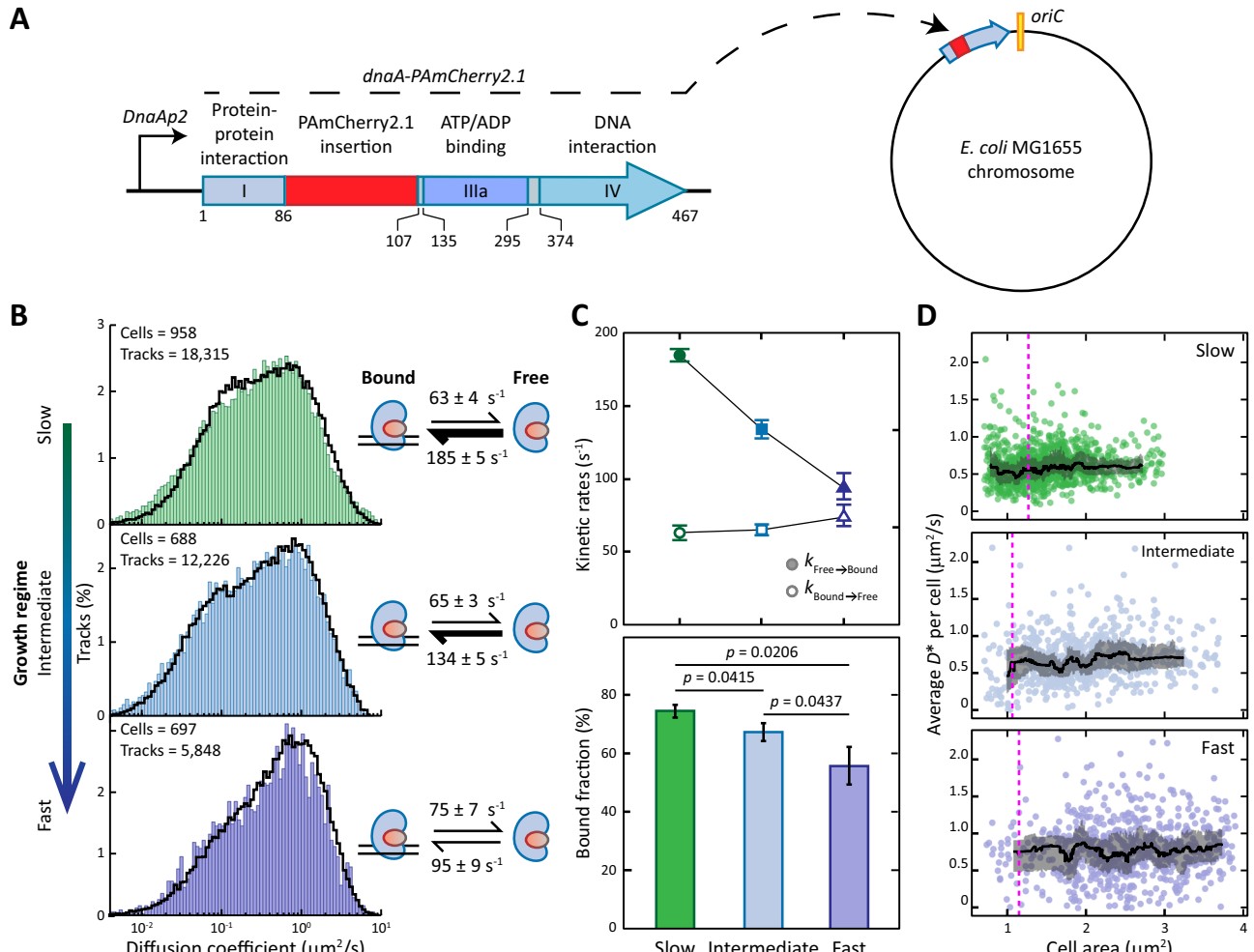

**Fig. 3 | SptPALM of DnaA in live *E. coli* MG1655 cells. A** Photoactivatable fluorescent protein PAmCherry2.1 was fused to DnaA in its native chromosomal locus by replacing amino acids 87-106 in Domain II of DnaA. The domains of DnaA are indicated with Roman numerals. Domain II and IIIb are depicted in light grey, yet are not specifically indicated for visualisation purposes due to their small size. On the bottom, the codon numbers of each domain of the wild-type DnaA are indicated, not accounting for the PAmCherry2.1 insertion. **B** Distributions of diffusion coefficients of DnaA in live *E. coli* cells grown in either slow, intermediate or fast growth regimes. The number of cells and tracks per histogram is shown in the upper left corner. Histograms are fitted (black line) with a theoretical description of 250,000 particles moving between a free and a bound state. On the right, kinetic rates describing the fitting are provided. **C** Trends of the kinetic rates $k_{Free \rightarrow Bound}$ and $k_{Bound \rightarrow Free}$ and average bound fraction of DnaA in different growth regimes. Data are presented as mean values ± SD, derived mathematically from the fitting of diffusional distribution or from kinetic rates in (**B**). *P*-values are calculated by a two-sided T test. **D** Trends of DnaA mobility throughout the cell cycle. The average diffusion coefficient of each cell is plotted against the area of the cell, with the black line indicating the rolling median of the population. The grey ribbons represent the interquartile range. The magenta vertical line indicates the area at which cells initiate DNA replication. See Supplementary Fig. 4G for the area distribution of the cells. Source data are provided as a Source Data file.

difference between wild-type *E. coli* and our *E. coli* MG1655 *dnaA-PAmCherry2.1* in all growth conditions (Supplementary Fig. 4E). Building on our previously established experimental framework[44] (Supplementary Fig. 1), we grew the obtained *E. coli* MG1655 *dnaA-PAmCherry2.1* strain (Fig. 3A) at a constant optical density to maintain cells exponentially growing in either a slow (doubling time 187 ± 32 min), intermediate (doubling time 62 ± 9 min) or fast growth regime (doubling time 31 ± 3 min). We then performed sptPALM (Supplementary Movie 1) and generated diffusional histograms of DnaA in different conditions. We analysed the distributions with Monte-Carlo diffusion distribution analysis (MC-DDA) and obtained kinetic rates governing transitions between a free and a DNA-bound state of DnaA (Fig. 3B). In this way, we observed a decreasing trend of the association rate $k_{Free \rightarrow Bound}$ from slow to fast growth regime, whereas the dissociation rate $k_{Bound \rightarrow Free}$ remained constant (Fig. 3C, top). We then used the kinetic rates to estimate the average DnaA bound fraction as $\frac{k_{Free \rightarrow Bound}}{k_{Free \rightarrow Bound} + k_{Bound \rightarrow Free}}$. The overall bound fraction of

DnaA decreased from 75 ± 2% in the slow growth regime, to 67 ± 3% in the intermediate and 55 ± 7% in the fast growth regime (Fig. 3C, bottom). The DNA-bound fraction in fast growth is at the lower end of a previously reported range of 55–89% for other DNA-binding proteins in *E. coli* grown in similar growth conditions[44,51]. The extracted $k_{Bound \rightarrow Free}$ rates suggest short-lived interactions of DnaA with the chromosome (average $t_{Bound} = 16 ± 2$ ms), consistent with previous reports[52]. The binding of DnaA to DNA has been reported to be temperature-dependent in vitro, with weaker binding occurring at lower temperature[53]. Therefore, our single-particle experiments performed at room temperature could have led to an underestimation of the bound fractions. However, this temperature-dependent effect was constant throughout all our experiments and does not invalidate the observed growth rate-dependent behaviour. We further note that, together with changing the carbon source, intermediate growth was achieved by lowering the culture temperature to 30 °C (see Methods). Whereas changes in culture temperature affect growth rate, these changes have been reported to have a smaller impact on cellular mass[10,54]. Still, *E. coli*

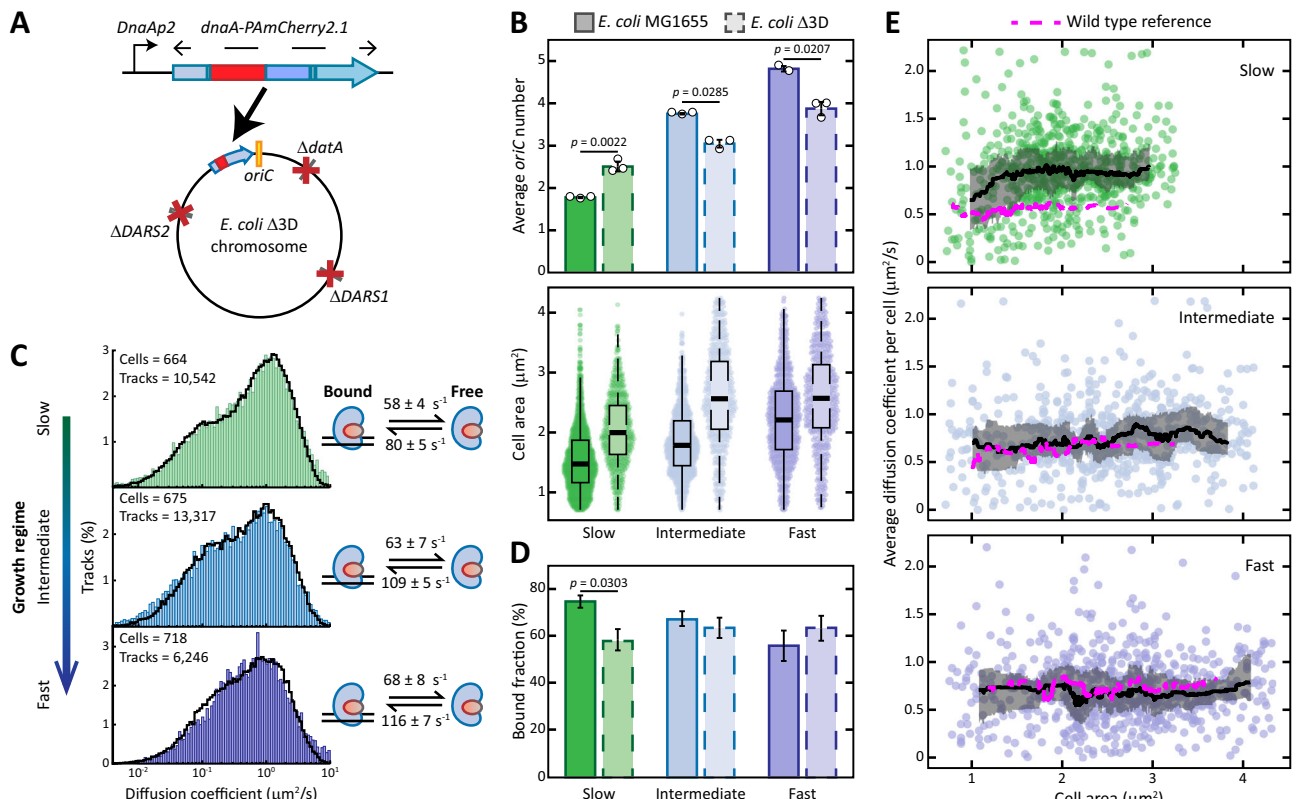

**Fig. 4 | The effect of the Δ*DARS1*, Δ*DARS2*, Δ*datA* triple deletion on the binding of DnaA and the DNA replication of *E. coli*. A** The *dnaA::PAmCherry2.1* mutation was introduced in the chromosomal locus of *dnaA* in the *E. coli* Δ3D strain. **B** Measurements of DNA content (top) (see Supplementary Fig. 5A for a complete list of replicates) and cellular area (bottom) of either the wild-type or the mutant strain for cells grown in slow ($n_{WT} = 1409$ cells, $n_{Δ3D} = 1056$ cells), intermediate ($n_{WT} = 790$ cells, $n_{Δ3D} = 469$ cells) or fast growth regime ($n_{WT} = 693$ cells, $n_{Δ3D} = 480$ cells). For the bar graph, data are presented as mean values ± SD, $n = 3$ biologically independent replicates. *P*-values are calculated by a two-sided T test. Box plots indicate median (middle line), 25th and 75th percentile (box) and 5th and 95th percentile (whiskers) as well as outliers (single points). **C** The diffusion coefficient histograms of DnaA obtained from sptPALM of *E. coli* Δ3D cells growing either in slow, intermediate or fast growth regime. Histograms are fitted (black line) with a theoretical description of 250,000 particles moving between a free and a bound

state. The number of cells and tracks per histogram is shown in the upper left corner. On the right, the kinetic rates describing the fitting are provided. **D** Average bound fraction of DnaA in each growth regime for either the wild-type *E. coli* MG1655 or the triple deletion mutant *E. coli* Δ3D strain. Data are presented as mean values ± SD, derived mathematically from kinetic rates in (**C**). *P*-values are calculated by a two-sided T test. **E** Trends of DnaA mobility throughout the cell cycle either in slow ($n_{WT} = 958$ cells, $n_{Δ3D} = 664$ cells), intermediate ($n_{WT} = 688$ cells, $n_{Δ3D} = 659$ cells) or fast growth regime ($n_{WT} = 697$ cells, $n_{Δ3D} = 705$ cells). The average diffusion coefficient of each cell is plotted against the area of the cell, with the black line indicating the rolling median of the population. The grey ribbons represent the interquartile range, whereas the dotted magenta lines represent the wild-type references. See Supplementary Fig. 5F for the area distribution of the cells. Source data are provided as a Source Data file.

cells exhibited the characteristic trend of larger cell size with increasing growth rate[7,10], with the measured areas progressively increasing from slow to fast growth conditions (Fig. 4B).

We then calculated an average of the diffusion coefficients for each cell and compared it with the respective size of each cell (Fig. 3D). In this way, we provided a proxy for the change in mobility of DnaA throughout the cell cycle in each growth condition. This area-dependent analysis further extended our previous observations on the average bound fraction of DnaA to the mobility of this protein in different phases of the cell cycle (Fig. 3D). Cells in slow growth regime exhibited a low average diffusion coefficient (~0.5 μm²/s) throughout the cell cycle, whereas in intermediate and fast growth regime DnaA had an overall higher mobility (around 0.7−0.8 μm²/s).

Our sptPALM assay revealed that the chromosome can effectively sequester DnaA in a growth rate-dependent fashion, as illustrated by the average DnaA bound fraction and the protein mobility through the cell cycle. This capacity is the highest in slow growth regime and progressively diminishes until a minimum during fast growth. The observed reduction in the bound fraction with increasing growth rate (Fig. 3C) is likely the result of a faster accumulation of cellular volume. To maintain a constant intracellular concentration[17], the synthesis rate of DnaA must

increase with growth rate. Since DnaA concentrations are higher in *E. coli* cells rapidly dividing[10], the increase in DnaA expression is also more pronounced at higher growth rates. In contrast, the synthesis of new titration sites per origin is set by the speed of DNA replication and is independent of growth rate[7]. As a result, the synthesis rate of DnaA increases faster with growth rate than the synthesis rate of titration sites, leading to a reduction in the effective titration power of the chromosome. These observations are consistent with previous mathematical modelling of a titration control mechanism[36].

## *DARS1, DARS2* and *datA* account for a large fraction of the chromosome titration power

The capacity of *E. coli* chromosome to sequester DnaA hints at initiator titration being a possible mechanism for controlling the activity of DnaA. The genome of this bacterium has hundreds of DnaA boxes that can recruit DnaA, peculiarly arranged to have a higher density towards the origin (Fig. 2). Whereas the potential function of most of these boxes is still unknown, six of them are at the *oriC*, while 26 more are located among the switch control loci *DARS1, DARS2* and *datA*[45]. To investigate the contribution of these boxes to the state of DnaA, we introduced the *dnaA-PAmCherry2.1* mutation in the triple

deletion strain *E. coli* Δ3D (Δ*datA*, Δ*DARS1*, Δ*DARS2*)[45] (Fig. 4A). We then characterised its phenotype at a population and a single-molecule level.

We first measured the DNA content (see also Supplementary Fig. 5A) and cell area of *E. coli* Δ3D in the different growth regimes (Fig. 4B). We observed a reduction in DNA content and an increase in cellular area in fast and intermediate growth regimes. The anti-correlation between the change in average cell size and the change in average number of origins upon deletion of loci mediating DnaA interconversion is consistent with previous reports[17,45] and with the idea that cells initiate a new round of replication after reaching a well-defined size[6,7,55]. This size is then larger in *E. coli* Δ3D than in *E. coli* MG1655 (Fig. 4B) due to the slight decrease in DnaA-ATP levels (Supplementary Fig. 5B) caused by the triple mutation. Interestingly, this anti-correlation was lost when *E. coli* Δ3D was cultured in slow growth regime. The median cellular area increased from $1.5\,\mu m^2$ in the wild-type to $2.2\,\mu m^2$ in *E. coli* Δ3D, pointing at delayed initiation. The *dnaAp* promoter was also more active in *E. coli* Δ3D than in the wild-type (Supplementary Fig. 5B), suggesting lower levels of DnaA-ATP as observed during intermediate and fast growth. The change in activity was milder compared to the other growth condition, likely due to a smaller contribution of RIDA and *DARS2* during slow growth. However, DNA content analysis revealed cells with a higher number of origins than the wild-type. More specifically, during slow growth *E. coli* Δ3D seems to cycle between 2 and 4 copies of the origin, rather than 1 to 2 as in the wild-type (Supplementary Fig. 5A). The triple mutant grew similarly to the wild-type strain in slow and fast growth regime and slower in intermediate growth regime (Supplementary Fig. 5C). From the number of cells with specific numbers of origins, we also obtained an asynchrony index and observed that this value was higher in *E. coli* Δ3D compared to the wild-type strain in all growth conditions (Supplementary Fig. 5D).

We then performed sptPALM on *E. coli* Δ3D *dnaA*-PAmCherry2.1 for cells grown in either slow, intermediate or fast growth regime (Fig. 4C). The absence of *datA, DARS1* and *DARS2* impacted the interaction of DnaA with the chromosome only during slow growth. In this condition, the bound fraction decreased from $75 \pm 2\%$ in the wild-type to $58 \pm 2\%$ in *E. coli* Δ3D (Fig. 4D). The average mobility of DnaA increased during the first part of the cell cycle, before stabilising at around $1\,\mu m^2/s$ (Fig. 4E). On the other hand, kinetic rates, bound fraction and DnaA mobility in *E. coli* Δ3D remained similar to the wild-type in intermediate and fast growth regimes (Fig. 4D, E). The near-wild-type levels of DnaA in *E. coli* Δ3D (Supplementary Fig. 5E) indicate that the decrease in DnaA bound fraction and increase in its mobility are likely due to a loss of important titration sites, rather than to an accumulation of extra DnaA proteins.

Altogether, the deletion of *datA, DARS1* and *DARS2* led to different phenotypes in the different growth conditions. During slow growth, we observed a decreased bound fraction and increased mobility of DnaA, despite the higher DNA content indicating that more DnaA boxes are present compared to the wild-type condition. These observations lead us to believe that the contribution of the three switch control loci on titration is substantial under these conditions. More interestingly, the higher mobility of DnaA in this condition was also correlated with an increase in both the average number of origins and cell size. The increase in both average cell size and DNA content can be reconciled by noting that the activity of the *dnaAp* promoter increased during slow growth (Supplementary Fig. 5B). This increase points at a lower abundance of DnaA-ATP in *E. coli* Δ3D compared to the wild-type, leading to delayed initiations and thus higher values of cell area. Still, the higher DNA content is in apparent contradiction with the lower abundance of DnaA-ATP and could be explained by the presence of frequent re-initiation events. As such, these results indicate that regulation of the DNA-bound state of DnaA could be important for the control of replication during slow growth.

## The DnaA-ATP/DnaA-ADP ratio affects DnaA binding to the chromosome

We next set out to understand whether the increased availability of free DnaA had an impact on the increase of DNA content in *E. coli* Δ3D through reduced initiator titration or whether the phenotype could be solely explained through the removal of switch components. To this end, we introduced single deletions of either *datA, DARS1* or *DARS2* and further investigated the behaviour of *E. coli* during slow growth.

As before, we performed sptPALM and derived kinetic rates through fitting (Fig. 5A). The changes in bound fraction observed in the single deletion mutants did not add up to yield the major decrease observed in *E. coli* Δ3D (Fig. 5B). The deletion of *datA* caused an increase in the average DnaA bound fraction (from $75 \pm 2\%$ to $82 \pm 3\%$), whereas removing *DARS1* caused a reduction to around 69%. The Δ*DARS2* mutation had no significant impact on DnaA bound fraction. By measuring the DNA content (Fig. 5C; see also Supplementary Fig. 6A) and cell size of the strains (Fig. 5D) we observed the expected anti-correlation in the *datA* and *DARS1* single deletion strains. The absence of *datA* led to an increase in the average number of origins, smaller cells and reduced activity of the *dnaAp* promoter (Supplementary Fig. 6B). All these observations are consistent with the idea that *datA* inactivates DnaA and its deletion lowers the initiation volume[36]. Notably, the average number of origins of the Δ*datA* population was lower than the one of *E. coli* Δ3D (2.1 versus 2.5). At the same time, deletion of *DARS1* led to larger cells with a lower number of origins than *E. coli* MG1655 and more active *dnaAp* promoter (Supplementary Fig. 6B), in line with the reactivating role of this locus. Removing *DARS1* thus leads to an increase in initiation volume[36]. The deletion of *DARS2* seemed not to impact the average number of origins of *E. coli* ($1.77 \pm 0.004$ compared to $1.78 \pm 0.016$) nor the activity of the *dnaAp* promoter when compared to the wild-type (Supplementary Fig. 6B). Such results are in line with the hypothesis that *DARS2* is inactive during slow growth[36] due to the lack of the protein Fis[56,57]. Yet, the Δ*DARS2* mutation led to a slight increase in the average cell size. All single deletion strains maintained near-wild-type levels of DnaA expression (Supplementary Fig. 6C). Moreover, they showed near-wild-type values of asynchrony index (Supplementary Fig. 6D) and mostly cycled between one and two origins (Supplementary Fig. 6A), pointing at a low frequency of re-initiation events.

Our analysis of single deletion mutants highlighted a correlation between the DnaA-ATP/DnaA-ADP ratio and the bound fraction of DnaA. Specifically, mutations that led to a higher fraction of DnaA-ATP (Δ*datA*) caused more DnaA to be titrated on the chromosome, while the opposite was observed in the Δ*DARS1* mutant strain. Such changes can be explained by the fact that DnaA-ATP better interacts with low-affinity DnaA boxes[19,20,30] than DnaA-ADP and can form dimers or oligomers[58]. Differences in DNA content and removal of specific loci also change the average number of DnaA boxes in the cell. On a phenotypical level, deletions of *datA* and *DARS1* behaved as predicted by a switch-only model, exhibiting the characteristic anti-correlation between average cell size and average DNA content. These observations are consistent with a model in which replication is initiated once per cell cycle at a well-defined volume, whose threshold value is predominantly set by the balance of activation and deactivation[35,36]. On the other hand, the severely re-initiating phenotype of *E. coli* Δ3D could not be explained solely by perturbation of factors involved in the switch mechanism. Rather, it revealed a complex behaviour that needs to be explained through a combination of DnaA interconversion and its titration on the *E. coli* chromosome.

## Changes in DnaA ATP/ADP ratio or titration do not impact its function as transcription factor

Next to initiating DNA replication, DnaA is also a transcription factor involved in the regulation of several genes[59,60]. Previous reports showed how the activity of transcription factors can be modulated by

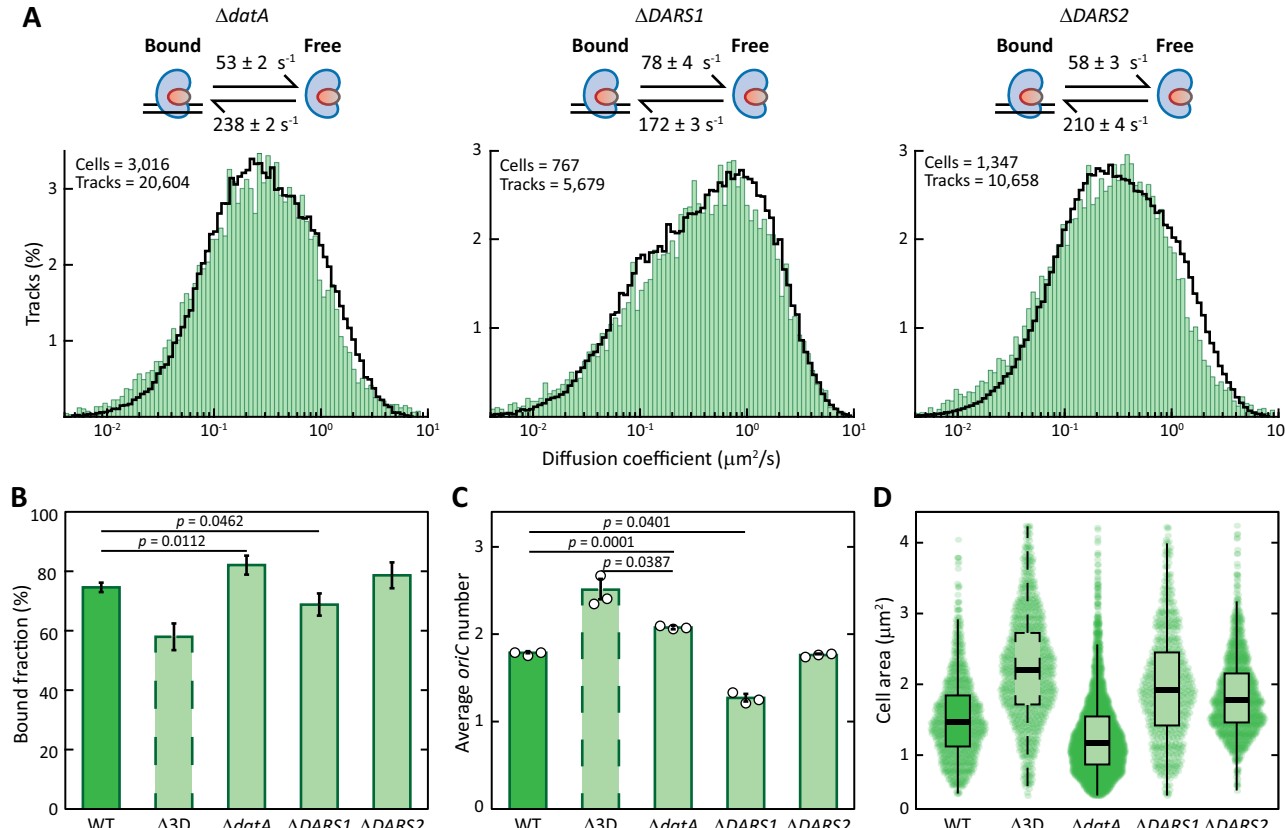

**Fig. 5 | Population- and single-molecule-level phenotypic characterization of single *datA*, *DARS1* or *DARS2* deletions in slow growth regime. A** Diffusion coefficient histograms of DnaA obtained from sptPALM with cells growing in slow growth regime in the absence of either *datA*, *DARS1* or *DARS2*. Histograms are fitted (black line) with a theoretical description of 250,000 particles moving between a free and a bound state. The number of cells and tracks per histogram is shown in the upper left corner. On top, the kinetic rates describing the fitting are provided for each growth regime. **B** Average bound fraction of DnaA, as estimated from the kinetic rates, for wild-type, triple deletion and the three single deletion *E. coli* mutants. Data are presented as mean values ± SD, derived mathematically from kinetic rates in (**A**). *P*-values are calculated by a two-sided T test. **C, D** Comparison of the average number of origins of replication (**C**) and the cell area of the population ($n_{\Delta datA}$ = 3016 cells, $n_{\Delta DARS1}$ = 760 cells, $n_{\Delta DARS2}$ = 1300 cells) (**D**). See Supplementary Fig. 6A for a complete list of replicates of DNA content measurements via flow cytometry. For bar graphs, data are presented as mean values ± SD, n = 3 biologically independent replicates. Box plots indicate median (middle line), 25th and 75th percentile (box) and 5th and 95th percentile (whiskers) as well as outliers (single points). *P*-values are calculated by a two-sided T test. Source data are provided as a Source Data file.

introducing additional binding sites functioning as titration spots[61]. Additionally, negative autoregulation is a characteristic shared between DnaA and a variety of other *E. coli* transcription factors[62]. As such, we investigated whether the titration of DnaA on the *E. coli* chromosome could function as a control mechanism for its activity as a transcriptional regulator. DnaA has been reported to activate the transcription of the *nrdA, polA, glpD* and *fliC* genes[59,63–65], while it acts as a repressor of the *mioC, uvrB, aldA, guaB, proS, rpoH, iraD* and *dnaA* genes[59,66–72]. Recently, chromatin immunoprecipitation experiments performed during fast growth provided evidence that the *purH, rne* and *nrdD* genes are also regulated by DnaA[73]. Notably, only four of these genes (*mioC, polA, purH* and *dnaA*) fall within the chromosomal region with a high density of DnaA boxes (Fig. 6A). In total, the DnaA boxes on the promoter region of these genes account for 14 of the 72 motifs found around *oriC*; the total increases to 22 when considering *oriC*. Thus, we exclude the possibility that the previously observed enrichment in boxes towards the origin can originate uniquely from the clustering of genes that are part of DnaA regulon in this region.

Nevertheless, we examined the proteomes of wild-type *E. coli* and *E. coli* Δ3D in the different growth conditions to obtain insights into how the triple deletion altered the global proteome (Fig. 6B). We argued that, in the case in which titration plays a role specifically in the transcription factor activity of DnaA, the decrease in DnaA bound fraction in *E. coli* Δ3D during slow growth would generate unique

changes in expression of the DnaA regulon. Despite large changes in the global proteome, we noticed that the majority of the DnaA regulon remained at wild-type levels upon deletion of *datA, DARS1* and *DARS2* in all growth regimes. The only exceptions were the flagellar protein FliC and the glycerol 3-phosphate dehydrogenase GlpD, which are overexpressed in all the tested growth conditions. Both the *fliC* and *glpD* genes are known to be positively regulated by DnaA. As such, it is surprising that mutations that lower the DnaA-ATP fraction, and thus its capacity to oligomerise on DNA[58], would result in their overexpression. Yet, the notion that these genes are part of the DnaA regulon, at least during fast growth, has been recently called into question[73]. We also detected an enrichment in the expression of the anaerobic ribonucleoside-triphosphate reductase NrdD during intermediate and fast growth. Next, we analysed the proteomes of the single deletion mutants cultured in slow growth regime (Fig. 6C). Single deletions of *datA* or *DARS1*, which led to marked changes in DnaA bound fraction (Fig. 5B), DNA content (Fig. 5C) and *dnaAp* activity (Supplementary Fig. 6B) did not substantially alter the global proteome of *E. coli* (Fig. 6C). Conversely, the deletion of *DARS2* generated a large change in *E. coli* expression profile, including the overexpression of FliC, GlpD and NrdD. After performing principal component analysis, we further noticed that *E. coli* ΔDARS2 and *E. coli* Δ3D clustered closely together (Fig. 6D), meaning that both strains exhibit similar changes in protein expression. This observation was

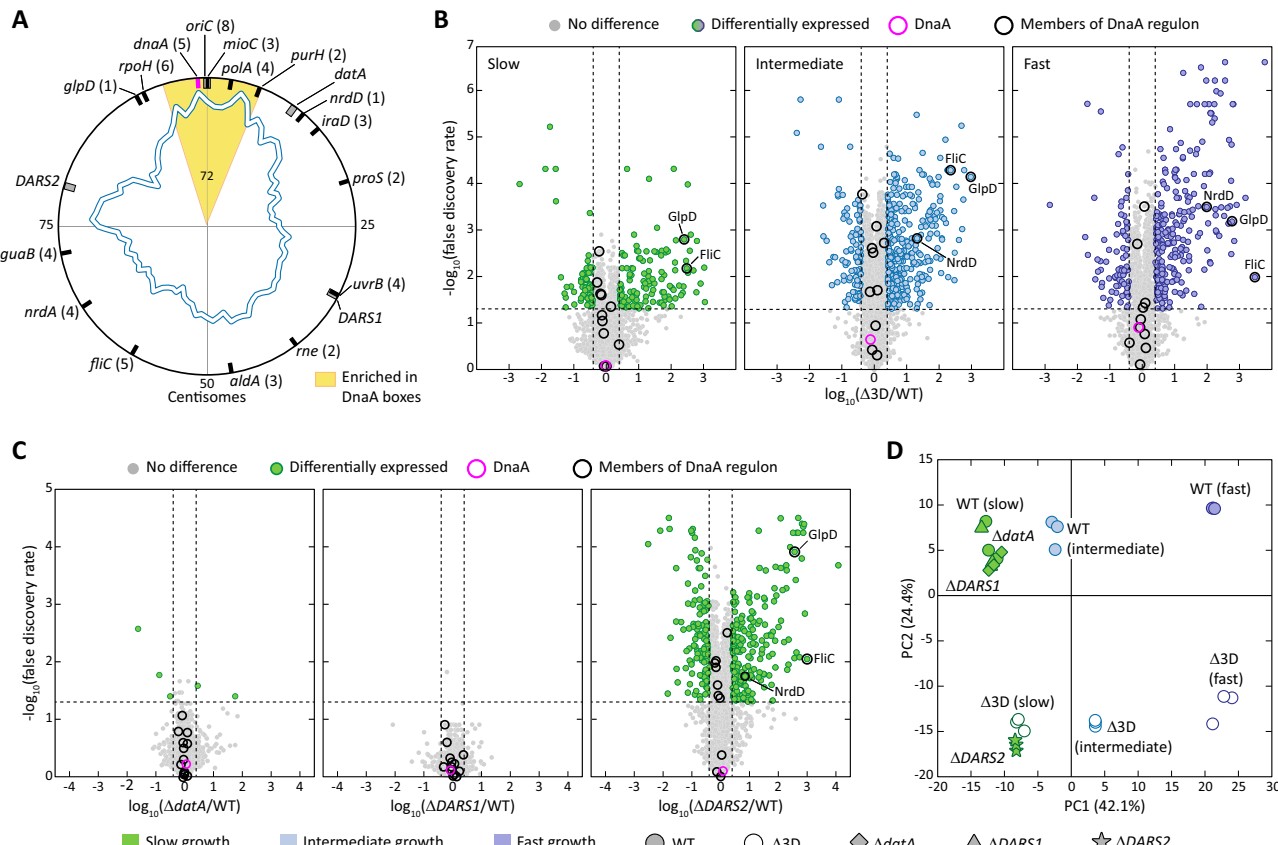

**Fig. 6 | The effect of ΔdatA, ΔDARS1 and ΔDARS2 deletion on the proteome of *E. coli*. A** Graphical representation of the genomic position of the DnaA regulon (black lines), of the *dnaA* gene (magenta line) and of known DnaA interaction loci (grey boxes). The number between brackets indicate the number of DnaA boxes on each gene. The white and blue line represents the count of DnaA boxes on the chromosome of *E. coli* MG1655, while the yellow area represents the region with significant DnaA boxes enrichment around *oriC*. Distances from the origin are expressed in centisomes, starting at *oriC*. **B**, **C** Global changes in the proteomes of (**B**) *E. coli* Δ3D in either slow (green), intermediate (blue) or fast (purple) growth regime or (**C**) *E. coli* ΔdatA, *E. coli* ΔDARS1 and *E. coli* ΔDARS2 during slow growth. In

all cases, the proteomes are compared to *E. coli* MG1655 *dnaA-PAmCherry2.1* in the same growth condition. Log$_{10}$ of fold changes in protein intensity are plotted against significance (-log$_{10}$(false discovery rate)), with cutoffs placed at 2.5 fold-change in expression and a false discovery rate of <0.05. Proteins outside the cutoffs are coloured according to the growth regime, while other proteins are in grey. Members of the DnaA regulon are circled in black, whereas DnaA is circled in magenta. **D** Principal component analysis showing all biological replicates for *E. coli* MG1655 *dnaA-PAmCherry2.1* (WT), *E. coli* Δ3D, *E. coli* ΔdatA, *E. coli* ΔDARS1 and *E. coli* ΔDARS2 in all relevant growth conditions. Source data are provided as a Source Data file. The complete set of protein intensity signals is available in PRIDE.

---

surprising, considering that the deletion of *DARS2* during slow growth did not alter DnaA bound fraction (Fig. 5B), DNA content (Fig. 5C) or *dnaAp* activity (Supplementary Fig. 6B). Yet, it serves as an explanation of the observed change in *E. coli* cell size upon its deletion (Fig. 5D).

Overall, mutations altering the DnaA-ATP fraction of DnaA (i.e., *dnaAp* activity) and/or its titration on *E. coli* chromosome (i.e., DnaA bound fraction) had no effect on the expression of DnaA regulon. Rather, our proteomics analysis hints at a yet unclear role of *DARS2* in the control of gene expression in *E. coli*. Whereas further investigating this role is beyond the scope of this study, we can conclude that switch and titration mechanisms seem to have little-to-no impact on the transcriptional regulator activity of DnaA during slow growth. Nevertheless, the binding motifs present on the promoter regions of the DnaA-regulated genes could play a role both in their transcription and in stabilising DNA replication via titration of DnaA.

## Discussion

In this work we present a systematic characterisation of DnaA behaviour in *E. coli* based on direct visualisation of DnaA. We confirm previous reports of transient interactions of DnaA with DNA[52] with an improved time resolution (10 ms against the previous 41 ms), while further enlarging the range of tested growth conditions and genetic backgrounds and implementing a cell size-dependent analysis. We

provide a genomic basis and long-awaited experimental evidence of the titration capacity of the *E. coli* chromosome proposed more than 40 years ago[22]. Finally, we show that the state of the nucleotide bound to DnaA affects its binding on the chromosome of *E. coli* and that altering the titration power of the chromosome can impact the control of DNA replication during slow growth. On the other hand, these changes did not impact the activity of DnaA as transcription factor.

While assessing the state of DnaA in the wild-type *E. coli* MG1655 (Fig. 3), we observed that the association rate $k_{\text{Free}\rightarrow\text{Bound}}$ constantly decreased from slow to fast growth regime, while the dissociation rate $k_{\text{Bound}\rightarrow\text{Free}}$ remained largely constant. As a result, the bound fraction of DnaA was highest during the slow growth regime and reached a minimum in the fast growth regime (Fig. 3C). This phenomenon is the result of two previously described *E. coli* characteristics: (i) near-balanced biosynthesis[17,74], leading to constant intracellular DnaA concentration during the cell cycle, and (ii) the approximately constant speed of DNA replication elongation[7]. Maintaining constant DnaA concentrations requires increasing the expression of the *dnaA* gene[10,74] in conditions in which the cellular volume increases faster, i.e. during higher growth rates[6]. On the other hand, the rate at which new DnaA boxes are replicated per origin is independent of the growth rate, due to the nearly constant speed of DNA replication[7]. Therefore, in increasingly faster growth conditions, the rate of new DnaA synthesis

progressively exceeds the rate by which DnaA boxes accumulate[36]. Moreover, the total concentration of DnaA is higher during faster growth[10]. Taking these features into consideration, our observed decreasing trend for the association rate indicates an excess of total DnaA proteins over the number of available titration sites at the steady-state. As a result, the chromosome ability to maintain a low concentration of free DnaA is impacted at higher growth rates. Still, the bound fraction accounted for more than 50% of visible DnaA proteins in all experimental conditions (Figs. 3C, 4D, 5B). Whereas the number and rate of synthesis of DnaA boxes per origin is constant across growth rates, the same does not hold for the total number of DnaA boxes per cell. First, the DNA content of *E. coli* cells increases with increasing growth rates (Fig. 4A). Second, cells eventually enter a regime of overlapping replication forks, in which the synthesis of new DNA proceeds through parallel replication rounds. The resulting increase in the total number of DnaA boxes per cell and in the rate at which these boxes are synthesised during progressively faster growth likely has an impact on the bound fraction of DnaA. Moreover, we hypothesise that the binding of DnaA to boxes scattered throughout the chromosome is also responsible for the overall high bound fractions, with pairs of closely spaced boxes[73] and specific loci (e.g. *mioC*[75] and *oriC*[76–78]) mediating more stable forms of binding. The preferential accumulation of boxes towards *oriC* (Fig. 2B) then means that new DnaA boxes can rapidly be synthesised after DNA replication initiation, enhancing the titration power of the chromosome at all growth rates. This genomic configuration and the trends of association rates and DnaA bound fractions in different growth regimes (Fig. 3C) are in accordance with previous modelling efforts[34,36] predicting the existence of a titration-based control mechanism.

The original initiator titration model[22] has recently been expanded to include the interconversion of DnaA[17,35,36]. Our experimental evidence supports these updated models, while providing new insights to complement them. In these models, the titration of both forms of DnaA on the chromosome[12,37,79] keeps the free DnaA concentration relatively low throughout the cell cycle, enhancing the synchrony of replication firing and preventing re-initiation events[34]. Due to the growth rate-dependent decrease in the DnaA bound fraction (Fig. 3), this mechanism is predicted to be prevalent during slow growth[34]. In line with this prediction, we report low mobility of DnaA across all observed cell size values in wild-type *E. coli* during slow growth (Fig. 3C) and we observed no re-initiation events (Supplementary Fig. 5A). After all DnaA boxes are saturated, specifically DnaA-ATP binds to the low-affinity motifs on *oriC*[19,23,24], thereby initiating replication. In the 10 min following replication initiation, SeqA binds to *oriC* and *dnaAp* to prevent subsequent firing of the same origin and transcription of new DnaA, respectively[80]. The repression of *dnaA* transcription enacted by SeqA could act in concert with the synthesis of new DnaA boxes via replication to temporarily enhance titration and reduce overall DnaA activity. This process would be favoured by the particular chromosomal configuration reported here (Fig. 2), leading to a rapid accumulation of new boxes. At the same time, RIDA and *datA* inactivate DnaA by promoting ATP hydrolysis, effectively preventing new rounds of initiation. Titration can also play a role in enhancing the action of RIDA. In RIDA, the protein Hda is loaded onto DNA polymerase β-clamps that are left on the lagging strand of DNA replication after the synthesis of Okazaki fragments[47]. From there, Hda inactivates nearby DnaA-ATP by stimulating the hydrolysis of ATP[81]. The striking preference for accumulating DnaA boxes on the lagging strand of each replichore (Supplementary Fig. 2E) likely increases interactions with the chromosome in proximity of Hda proteins, ensuring efficient inactivation of DnaA.

Initiator titration models often consider that both forms of DnaA are titrated equally by the DnaA boxes on the chromosome[17]. Instead, our analysis hints at changes in the DnaA-ATP/DnaA-ADP ratio influencing the titration of DnaA on the chromosome of *E. coli*, at least

during slow growth. Such effects could then have relevance in the control of DNA replication. The Δ*datA* mutation led to both an increase in DnaA-ATP/DnaA-ADP ratio (Supplementary Fig. 6B) and in the bound fraction of DnaA (Fig. 5). The increase in DnaA titration could affect replication initiation by limiting the binding of DnaA-ATP to *oriC* and thus re-initiation events (Fig. 7, top), consistent with the low fraction of cells with more than two origins we observed (Supplementary Fig. 6A). Conversely, the deletion of *DARS1* resulted in a lower abundance of DnaA-ATP (Supplementary Fig. 6B) and caused a decrease in the bound fraction of DnaA (Fig. 5). In this sense, when the cell lacks a major DnaA reactivating component, a decrease in titration could lead to higher amounts of free DnaA-ATP and mitigate delays in replication initiation (Fig. 7, middle), thus providing robustness. A role of titration in complementing interconversion of DnaA is also consistent with the fact that reaching peak levels of DnaA-ATP is a necessary but not sufficient condition to initiate replication[33].

The peculiar phenotype of *E. coli* Δ3D during slow growth, where both DNA content and cell size increase (Fig. 4), can also be explained by considering the combined action of titration and switch mechanisms. In this mutant, the deletions of *datA* and *DARS1* leave the cell in a state of DnaA-ATP deficiency, as indicated by the increased activity of the *dnaAp* promoter in our bulk assay (Supplementary Fig. 6B). This effect also leads to a matching decrease in titration (see Δ*DARS1* mutant), which is further impacted by the loss of *DARS1, DARS2* and especially *datA*[40,82] as titration loci. Due to the lack of reactivating loci, the only mechanism left for the cell to accumulate sufficient DnaA-ATP is through synthesis of new DnaA[25]. The accumulation of new DnaA-ATP then leads to the increase in DnaA mobility that we observed at the beginning of *E. coli* Δ3D cell cycle (Fig. 4D). After sufficient DnaA-ATP is accumulated and replication is initiated, new DnaA boxes are rapidly synthesised, thereby stabilising DnaA mobility (Fig. 4C) and lowering its activity. At this point, *E. coli* is mostly relying on the action of RIDA to inactivate DnaA, together with its intrinsic ATPase activity[17]. In this scenario, most DnaA is still freely diffusing rather than bound on the chromosome (Fig. 4D) and modest accumulation of more DnaA-ATP could then stimulate cells to initiate new rounds of replication after the release of SeqA from *dnaAp* and *oriC*[17] (Fig. 7, bottom). Such frequent re-initiations would then explain the observed increase in DNA content in *E. coli* Δ3D during slow growth. It is possible that the decreased rate of DnaA-ATP hydrolysis caused by the absence of *datA* is sufficient to explain the re-initiating phenotype of the triple deletion mutant. It should further be noted that the *dnaAp* activity assay employed here relies on bulk measurements that could mask transient peaks in the active form of DnaA in key moments of the cell cycle. However, if high DnaA-ATP levels would be the sole determinant of re-initiation frequency, *E. coli* Δ*datA* would show the highest DNA content out of all the strains examined during slow growth. In *E. coli* Δ*datA*, the effect of the absence of the inactivating locus is further aggravated by the presence of the reactivating *DARS1* locus, leading to the highest DnaA-ATP levels observed (Supplementary Fig. 6B). Yet, *E. coli* Δ*datA* showed a lower number of origins than *E. coli* Δ3D (Fig. 5C) and near-wild-type values of asynchrony index (Supplementary Fig. 6D). We believe that this surprising difference can also serve as indirect evidence for the role of DnaA titration on the coordination of DNA replication.

Overall, our findings provide important experimental support of previous models in which initiator titration stabilises replication cycles[32,36] and suppresses intrinsic noise in DNA replication initiation. A titration-based control mechanism was hypothesised to be an ancient way of controlling DNA replication in slow-growing ancestors of *E. coli*[32,35]. Interestingly, the peculiar chromosomal configuration that we found in *E. coli* MG1655, with a high representation of DnaA boxes arranged close to the chromosomal *oriC*, was present in most of the analysed strains of this species. The conservation of this arrangement perhaps hints at initiator titration being a primordial form of DNA replication control. In the future, specifically altering the titration

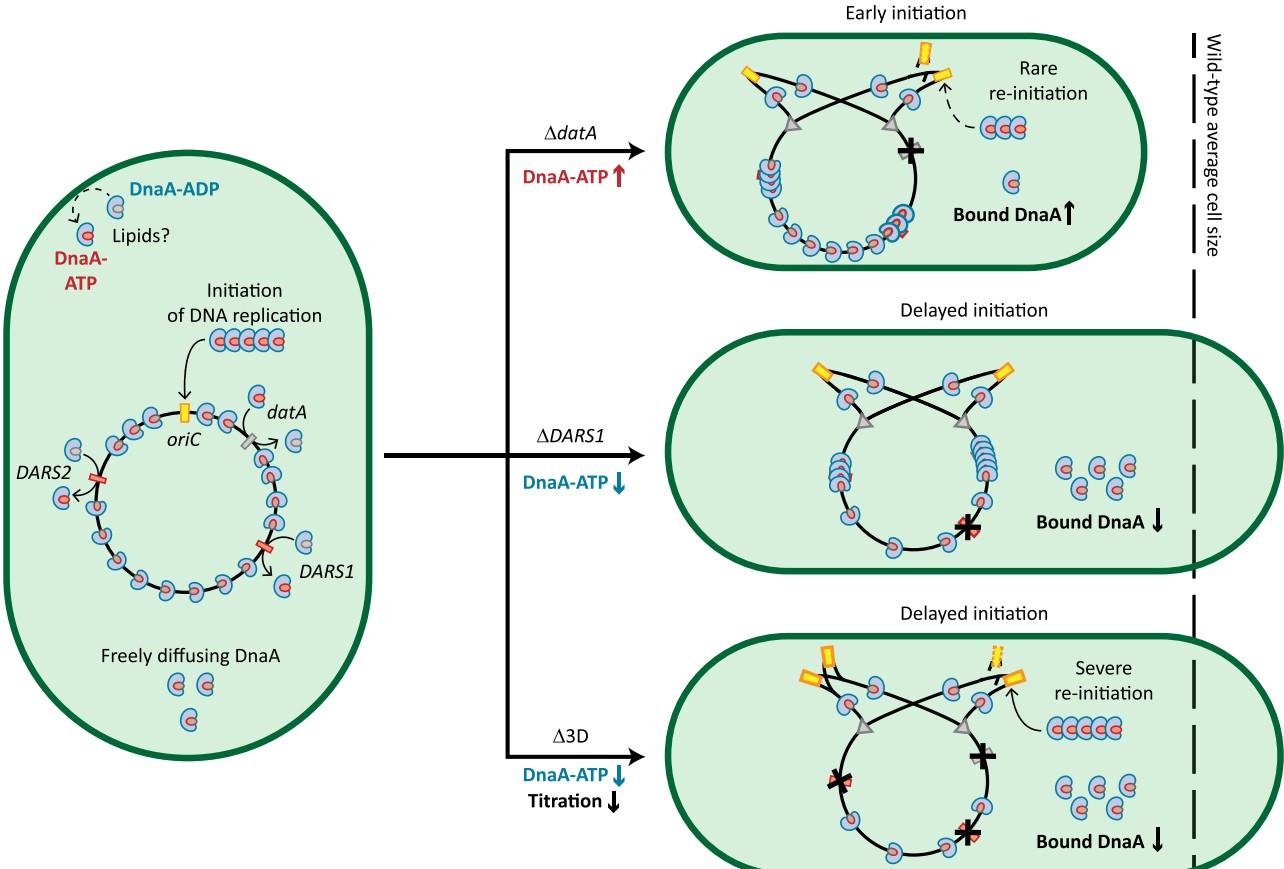

**Fig. 7 | The proposed role of titration in controlling re-initiation events.** First, DnaA-ATP is accumulated to levels allowing DNA replication initiation (left). In *E. coli* Δ*datA* (top) the level of DnaA-ATP is higher, leading to early initiations and overall smaller cells. The increased titration can remove DnaA-ATP from the cytosol and minimise re-initiation events. In *E. coli* Δ*DARS1* (middle) the level of DnaA-ATP is lower than wild-type. Therefore, the cell needs more time to accumulate enough DnaA-ATP, leading to delayed initiation and larger cells. The decrease in titration can help mitigate this delay, by leading to higher levels of free DnaA-ATP competent in origin binding. In *E. coli* Δ3D (bottom) the level of DnaA-ATP is lower, leading again to longer times needed to accumulate DnaA-ATP, delayed initiation and larger cells. At the same time, titration is severely hampered. Therefore, when the threshold of DnaA-ATP is met, active DnaA remains free in the cytosol and can bind once again to *oriC*, leading to frequent re-initiations.

of DnaA by removing binding motifs outside of *datA*, *DARS1* and *DARS2* could shine further light on the role of this control system also in setting the volume of initiation. Extending previous attempts of chromatin immunoprecipitation of *E. coli* DnaA[73] to different growth rates and genetic backgrounds could also provide a better understanding of the usage of each box. Finally, implementing microfluidics-based cultivation strategies[83] could then allow to probe variations in DnaA abundance[17] and the DnaA-ATP/DnaA-ADP ratio[33], as well as to connect changes in DnaA mobility with key moments of the *E. coli* cell cycle. In any case, our experimental pipeline may serve as a solid basis for any future research on this elusive form of control, for which direct visualisation of the initiator protein DnaA is posed to be essential.

## Methods
### Computational identification of chromosomal DnaA box densities

Nucleotide fasta files of several *E. coli* strains genomes[45], as well as the reference genome of *Salmonella enterica* were downloaded from RefSeq[84]. Identifiers NC_017644.1, NC_017663.1, and NC_020518.1 were removed from RefSeq and therefore excluded from analysis. A custom-made Python script (matchBox.py) was developed to scan circular chromosomes and identify exact matches to the DnaA box motifs, either on the forward or reverse strand. The consensus DnaA box sequence (TTWTNCACA)[18] and other sequences reported as active in DnaA binding across *datA, DARS1* and *DARS2* (HHMTHCWVH)[45] were

included in the list of potential matches. We accounted for the circular nature of *E. coli* and *S. enterica* chromosomes by allowing detection of motifs near the origin and terminus. Matches were identified and assigned a sequence ID, position, orientation and motif. A complete list of the matches for each genome is available in the Source Data file. The distribution of DnaA box motifs was then analysed and visualised with R v4.4.1[85]. The relative positions of DnaA box motifs within each chromosome was calculated by scaling to centisomes, i.e. dividing the total length of each genome in 100 equal intervals[86], starting from the position of *oriC*, determined via the DoriC database[87]. The null hypothesis of DnaA boxes being homogenously distributed was generated by repeatedly ($n = 1000$) simulating DnaA box positions over the chromosome from a uniform distribution in R v4.4.1[85] with the 'runif' function. We calculated the number of boxes per position in a sliding window approach by moving over the scaled chromosome in either 2, 6 or 10 centisome windows, taking 1 centisome steps. Pseudo $p$-values were obtained based on the rank of the real box number per position compared to the simulated values from a uniform distribution with a significance cutoff of $p$-value < 0.05. To additionally test whether the non-uniform distribution of DnaA boxes derived from the genome compositional background (i.e., GC-content), we created 1000 permutations of the used DnaA box sequences using a custom-made Python script (shuffleBoxes.py). We searched the chromosome for identical matches to the permutated boxes and calculated the number of motifs per position in a sliding window approach as for the

uniform distribution test. Pseudo *p*-values were calculated and used as before to obtain significance cutoffs.

Circular statistics was employed to estimate the density of DnaA boxes along the genomes, using the circular v0.5-0 package[88], and the von Mises kernel to smooth the density of motifs. The density estimates were subsequently scaled to Z-score to indicate the number of standard deviations the value is away from the mean density of boxes across the chromosome (Z-score = 0). Extreme density values were extracted with the zoo v1.8-12 package[89] 'rollapply' function, with the largest Z-score assigned as the global maximum for that genome. Data was then visualised with the ggplot2 v3.5.1 package[90].

### Bacterial strains, plasmids and growth conditions

A list of bacterial strains used in this study is provided in Supplementary Information (Table S1). *Escherichia coli* DH5α (NEB) was used for general plasmid propagation and standard molecular techniques. The wild-type *Escherichia coli* MG1655 and mutant strains lacking the genomic loci *datA, DARS1* and *DARS2* either individually or in combination[45] were used to investigate DnaA interactions with the bacterial chromosome.

All *E. coli* strains were routinely grown at 30 °C or 37 °C and 200 rpm in LB liquid medium (10 g/L tryptone, 10 g/L NaCl, 5 g/L yeast extract) for strain propagation. M9 liquid medium (5 g/L $KH_2PO_4$, 0.5 g/L NaCl, 6.78 g/L $Na_2HPO_4$, 1 g/L $NH_4Cl$, 4.98 mg/L $FeCl_3$, 0.84 mg/L $ZnCl_2$, 0.13 mg/L $CuCl_2 \cdot 2H_2O$, 0.1 mg/L $CoCl_2 \cdot 6H_2O$, 0.1 mg/L $H_3BO_3$, 0.016 mg/L $MnCl \cdot 4H_2O$, 0.5 g/L $MgSO_4 \cdot 7H_2O$, 11 mg/L $CaCl_2$) was used during constant density cultivations or single-particle tracking. Growth conditions to achieve different growth regimes were: M9 supplemented with 4 g/L glucose and 1x RPMI 1640 amino acids at 37 °C for fast growth regime; M9 supplemented with 4 g/L succinate and 1x RPMI 1640 amino acids at 30 °C for intermediate growth regime; and M9 supplemented with 4 g/L acetate at 37 °C for slow growth regime. Media were eventually supplemented with agar (15 g/L) to obtain solid media and with antibiotics for plasmid propagation (50 mg/L kanamycin, 100 mg/L spectinomycin, 100 mg/L ampicillin, 25 mg/L chloramphenicol).

Phosphate-buffered saline solution (PBS) (8 g/L NaCl, 0.2 g/L KCl, 1.42 g/L $Na_2HPO_4$, 0.24 g/L $KH_2PO_4$) was used to wash cells during sample preparation.

### Plasmid construction

A list of all constructs used in this study is provided in Supplementary Information (Table S2). The pCas and pTarget system[91] was used to obtain genomic mutants of *E. coli* MG1655. The pTarget_dnaA-PAFP plasmid series were constructed from pTarget[91]. The pTarget_dnaA-PAFP plasmid series expressed a single-guide RNA targeting the chromosomal *dnaA* gene of *E. coli* in its domain II and a repair template flanked by 50 bp-long homology arms. The repair template consisted of the gene encoding one of four photoactivatable or photo-convertible fluorescent proteins (PAFP), replacing bases 259-312 of the native *dnaA* gene[48,49]. The tested PAFP were Dronpa2[92] (BG25452), mEos4b[93] (BG25454), mMaple3[94] (BG25455) and PAmCherry2.1, a M10L[95] mutant of PAmCherry2[96]. Sequences were codon-optimised using the IDT Codon Optimisation Tool (IDT) and DNA fragments coding for Dronpa2, mEos4b and mMaple3 were chemically synthesised (IDT). The sequences of the chemically synthesised fragments are provided in Supplementary Information (Table S3). The sequence of the codon-optimised PAmCherry2.1 was retrieved from the pLbdCas12a-PAmCherry2.1_scrambled plasmid[44].

The pDnaA211-PAmCherry2.1 plasmid was based on a pBeloBAC11 plasmid[97] modified to express proteins of interest via the TetR-repressible promoter PLtetO-1 and using a bicistronic design[98]. The plasmid contained the mutant allele *dnaA211*, encoding a M426T mutation reported to abolish binding to DNA[99]. The mutation was introduced via site-directed mutagenesis using primers BG36136 and

BG36137. The gene encoding PAmCherry2.1 was used to replace bases 259-312 of the DnaA211 gene.

The pdnaAp-LacZ plasmid was based on the design of the pTACDNAA plasmid[17,50], using a pSC101 plasmid containing the *lacZ* gene under the control of *dnaAp*, the native *dnaA* promoter. Both *dnaAp* and *lacZ* were amplified from the genome of *E. coli* MG1655, using primers BG35613 and BG35614 for *dnaAp* and BG36203 and BG36204 for *lacZ*.

For cloning purposes, DNA fragments were amplified by PCR using Q5® High-Fidelity DNA Polymerase (NEB) following manufacturer instructions. Specific oligonucleotides were designed and synthesised (IDT) to introduce proper overhangs for assembly. Assembly was performed using NEBuilder® HiFi DNA Assembly Master Mix (NEB).

### Electro-competent cells preparation and transformation procedure

Pre-cultures of *E. coli* were grown overnight in LB liquid medium at 37 °C and 200 rpm. Cells were made electrocompetent by re-inoculating and culturing them at 37 °C and 200 rpm in LB. Strains carrying either the pCas9[91], the pCas9_ampR or the pSIJ8[100] plasmid were instead cultured at 30 °C and 200 rpm in LB, supplemented with L-arabinose (final concentration 10 mM) to induce expression of the λ-Red system. Upon reaching $OD_{600nm}$ of 0.4, the cells were cooled down to 4 °C, washed one time with 1 culture volume of ice-cold Milli-Q water and two times with 0.5 culture volumes of an ice-cold 10% V/V glycerol solution in Milli-Q water. Cells were then suspended in ice-cold 10% glycerol to a final volume of 400 μL for each 100 mL of initial culture volume and dispensed in 40 μL aliquots. All washing steps were performed at 4 °C, by centrifugation for 10 min at 3000 x*g*.

Electroporation was performed in ice-cold 2 mm electroporation cuvettes at 2500 V, 200 Ω and 25μF. Immediately after electroporation, cells were recovered in LB medium. Recovery was performed either at 30 °C, 750 rpm for 2 h for cells harbouring either the pCas9, the pCas9_ampR or the pSIJ8 plasmid or at 37 °C, 750 rpm for 1 h in all other instances. Following recovery, cells were spread on LB solid medium supplemented with appropriate antibiotics for plasmid maintenance and incubated overnight at the appropriate temperature.

### Creation of the *dnaA-PAFP* fusion mutants

The different photoactivatable or photoconvertible proteins were knocked in domain II of DnaA, replacing base pairs 259-312 of the native *dnaA* gene[48,49]. To this end, a previously reported procedure was followed, consisting of a two-plasmid system[91]. Briefly, electro-competent cells of wild-type *E. coli* MG1655 harbouring the pCas plasmid were transformed with one of the four pTarget_dnaA-PAFP plasmids. Correct insertion of the PAFP was confirmed first by PCR using primers BG21364 and BG21365 and then by Sanger sequencing of the locus. The same approach was used for the generation of the *dnaA-PAFP* fusion mutant in the *E. coli* Δ3D background. In this case, the pCas_ampR was used instead of the original pCas plasmid and only the pTarget_dnaA-PAmCherry2.1 plasmid was used to introduce the mutation. After insertion of *PAmCherry2.1* gene in *E. coli* MG1655 and *E. coli* Δ3D, whole genome sequencing was performed to confirm that the modified locus was the only source of DnaA.

In the case of the single deletion mutants, electrocompetent cells of *E. coli* MG1655 *dnaA-PAmCherry2.1* harbouring the pSIJ8 plasmid[100] were transformed with a linear fragment replacing either *DARS1* or *DARS2* with a chloramphenicol resistance cassette or *datA* with a kanamycin resistance cassette. The linear fragments were obtained by amplifying the genomic locus of *E. coli* mutant strains carrying either the Δ*datA::kanR* (primers BG23801 and BG23802), Δ*DARS1::cat* (primers BG23803 and BG23804) or Δ*DARS2::cat* allele[45] (primers BG23805 and BG23806). Transformant colonies were first selected in the appropriate resistance and the locus of interest was then further

confirmed by PCR. The sequences of the oligonucleotides are provided in Supplementary Information (Table S3).

## Turbidostat cultivation of cell cultures

*E. coli* strains were grown overnight in 5 mL of LB medium at 37 °C and 180 rpm. The densely-grown culture was then used to start a turbidostat cultivation using the Chi.Bio platform[101]. Experiments were started according to manufacturer instructions for fluidic lines, electrical connections and user interface. Cultures were started at an $OD_{600nm}$ of 0.05 in M9 medium. Both the M9 supplementation and the temperature were selected according to the specific growth regime to obtain. Default settings were employed for stirring and frequency of optical density measurements. A 650 nm laser diode was used to monitor the optical density of the culture, to minimise accidental photoactivation of PAmCherry2.1. The target optical density was set to 0.4 AU (corresponding to an $OD_{600nm}$ of ~0.2) and the media reservoir was filled with M9 supplemented as the cultivation chambers. Supplemented M9 medium was pumped in the vessel from the reservoir to maintain a constant optical density. Cells were balanced in exponential growth at the target optical density for at least ten generations before 10 mL of culture was harvested for imaging. The remainder was subjected to replication run-out for DNA content measurements.

In a turbidostat, the dilution rate ($D$) is equal to the growth rate ($\mu$). For each cultivation, the total volume ($V_{SteadyState}$) and time ($t_{SteadyState}$) spent at the steady-state optical density were used to calculate the flow rate $F = \frac{V_{SteadyState}}{t_{SteadyState}}$. $F$ was then used together with the cultivation volume ($V_{Culture} = 20$ mL) to calculate dilution rate and thus growth rate, following $\mu = D = \frac{F}{V_{Culture}}$.

## Sample preparation and single-particle tracking photo-activatable localisation microscopy

To perform sptPALM of DnaA in live *E. coli*, 10 mL of cells were collected in a 50 mL Falcon tube from turbidostat cultures growing in one of the three different growth regimes (slow, intermediate, fast). Cells were then washed three times in PBS before removing the supernatant and resuspending the pellet in 50 μL of PBS. 1–2 μL of the final cell suspension were immobilised on M9 agarose pads between two heat-treated glass coverslips (#1.5H, 170 μm thickness).

All sptPALM experiments were performed at room temperature using the miCube open microscopy framework[102]. Briefly, the microscope mounted an Omicron laser engine, a Nikon TIRF objective (100x, oil immersion, 1.49 NA, HP-SR) and an Andor Zyla 4.2 PLUS camera running at a 10 Hz for brightfield imaging acquisition and 100 Hz for sptPALM. For each imaging experiment, 300 frames were acquired at 100 ms intervals with brightfield illumination using a commercial LED light (INREDA, IKEA, Sweden). For sptPALM, individual videos of 30,000 frames were acquired at 10 ms intervals. Multiple videos were collected for each field of view, until exhaustion of fluorophores. The Single Molecule Imaging Laser Engine (SMILE) software was used to control the lasers (https://hohlbeinlab.github.io/miCube/LaserTrack_Arduino.html). A 561 nm laser with ~0.12 W/cm² power output was used for HiLo-to-TIRF illumination with 4 ms stroboscopic illumination in the middle of 10 frames. A 405 nm laser was used to activate the PAmCherry2.1 fluorophores. The 405 nm laser was initially provided with low-power (μW/cm² range) and with a 0.5 ms stroboscopic illumination at the beginning of 10 ms frames[103]. Both the power and the stroboscopic illumination were progressively increased throughout the imaging until exhaustion of fluorophores. Raw data was acquired using Micro-Manager[104]. During acquisition, a $2 \times 2$ binning was used, yielding an effective pixel size of $119 \times 119$ nm. The excitation field of ~$30 \times 30$ μm was restricted to regions of interest of $256 \times 256$ pixels or smaller during imaging. The first 500 frames of each video were discarded, to prevent attempted localisation of overlapping fluorophores and to pre-bleach fluorescent contaminants in and around the cells.

## Cells segmentation and cell area and volume estimation

Cell segmentation was performed as previously described[44] using ImageJ[105]/Fiji[106]. Briefly, watershed-based segmentation[107] (http://imagej.net/Interactive_Watershed) was performed to obtain pixel-accurate cell outlines from the brightfield images. From there, a value for the area in pixels of each identified cell was obtained and the camera pixel size ($119 \times 119$ nm) was used to generate the corresponding area measurements in μm². The brightfield images, together with their segmented counterparts are available on Zenodo (https://doi.org/10.5281/zenodo.13930507). A list of obtained cellular area is available in the Source Data file.

When cell volume was needed to estimate volume of initiation, previously reported[108] aspect ratios ($AR$) of *E. coli* cells cultured in growth conditions similar to the ones in this study were used (Slow: $\mu = 0.2$ h$^{-1}$ and $AR = 3.43$; Intermediate: $\mu = 0.7$ h$^{-1}$ and $AR = 4.14$; Fast: $\mu = 1.4$ h$^{-1}$ and AR = 4.3). *E. coli* cells were considered as cylinders with a hemisphere at each pole and a relationship between area of *E. coli* ($A$) and radius ($r$) of the sphere and cylinder was obtained as $r = \sqrt{\frac{A}{\pi + 4 \cdot AR - 4}}$. From here, volume values were obtained as $V = (2 \cdot AR - 0.7) \cdot \pi \cdot \sqrt{\frac{A}{\pi + 4 \cdot AR - 4}}^3$. The volume of initiation ($V^*$) was then obtained using the mathematical relationship $V^* = \frac{\langle V \rangle}{\ln 2 \cdot n_{ori}}$ between average cellular volume ($\langle V \rangle$) and the average number of origins ($n_{oriC}$). $V^*$ values were then converted back to area values using the relationship previously mentioned between the two values.

## Localisation of fluorophores and tracking of single particles

Single-molecule localisation was performed via the ImageJ/Fiji plugin ThunderSTORM[109], with added maximum likelihood estimation-based single-molecule localisation algorithms. First, a 50-frame temporal median filter[110] (https://github.com/HohlbeinLab/FTM2) was applied to the sptPALM movies to correct background intensities[111]. Image filtering was performed through a difference-of-Gaussians filter (Sigma1 = 2 px, Sigma2 = 8 px). The approximate localisation of molecules was determined via a local maximum with peak intensity threshold of $std(Wave.F1) \cdot 1.2$ and 8-neighbourhood connectivity. Sub-pixel localisation was performed through Gaussian-based maximum likelihood estimation, with a fit radius of 4 pixels (Sigma = 1.5 px). A custom-written, MATLAB-based pipeline was used to process and analyse the imaging data. A complete list of localisations for each replicate and condition, together with the custom-written MATLAB pipeline is available on Zenodo (https://doi.org/10.5281/zenodo.13930507).

Different output files from ThunderSTORM were combined when multiple videos had to be recorded for the same field of view. Localisations were then assigned a cell ID if they fell inside a cell and were discarded if not. Single, valid localisations were linked into tracks according to spatial and temporal distances. The tracking procedure was performed as previously reported[102] and yielded the number of tracks observed in each single cell and an overall apparent diffusion coefficient distribution. For each track featuring at least three localisations, the apparent diffusion coefficient $D^*$ was obtained by calculating the mean square displacement between the first $n$ steps and taking the average of that, where $n$ is the number of localisations minus one. The diffusion coefficients of all tracks were then collected into 85 logarithmic-divided bins from $D^* = 0.04$ μm²/s to $D^* = 10$ μm²/s.

## Monte-Carlo diffusion distribution analysis of DnaA

To interpret the diffusional distributions obtained for DnaA-PAmCherry2.1 fusion mutants, a set number of proteins was simulated moving between a bound and a free state in a two linear state model (50,000 for the fit, 250,000 for the visualisation), using Monte-Carlo diffusion distribution analysis (MC-DDA)[44]. The distributions were then fitted with a general Levenberg-Marquardt procedure in MATLAB, yielding kinetic rates with a 95% confidence

interval. A single value for localisation uncertainty was used ($\sigma = 0.035\,\mu m$, or $D^{*}_{\text{Immobile}} = 0.12\,\mu m^2/s$). DnaA was assigned a $D^{*}_{\text{Free}}$ diffusion constant value for its free state and a $D^{*}_{\text{Bound}}$ diffusion constant value for its bound state (see below for an explanation of the values). Additionally, initial rates $k_i$ governing the transitions between the free and bound state ($k_{\text{Free} \rightarrow \text{Bound}}$, $k_{\text{Bound} \rightarrow \text{Free}}$) were assigned to the proteins. The proteins were randomly placed inside a cell, simulated as a cylinder (length $2\,\mu m$ and radius $0.5\,\mu m$) with two hemispheres at its poles (radius $0.5\,\mu m$). Each protein is randomly put in one of the two different states, based on the probability set by their kinetic rates $\left( p_{\text{Free}} = \frac{k_{\text{Bound} \rightarrow \text{Free}}}{k_{\text{Bound} \rightarrow \text{Free}} + k_{\text{Free} \rightarrow \text{Bound}}}, p_{\text{Bound}} = \frac{k_{\text{Free} \rightarrow \text{Bound}}}{k_{\text{Bound} \rightarrow \text{Free}} + k_{\text{Free} \rightarrow \text{Bound}}} \right)$. From there, the proteins are given a time before they are changed to a different state, defined as state-change time $t_{\text{Change}}$ and calculated as $t_{\text{Change}} = \frac{\log(rand)}{-k}$, where $rand$ is an evenly distributed random number and $k$ is the kinetic rate governing the transition between the current and the next state. The movement of each protein is simulated with over-sampling with regards to the frame-time ($t_{\text{Frame}} = 10\,ms$, $t_{\text{Step}} = 0.1\,ms$) and their state is recorded every $t_{\text{Frame}}$ interval. In their free state, each DnaA protein moves for a distance ($s$) equal to a randomly sampled normal distribution, centred around $D^{*}_{\text{Free}}$ and with $s = \sqrt{2 \cdot D^{*}_{\text{Free}} \cdot t_{\text{step}}}$. At every step, the $t_{\text{Change}}$ is subtracted with the $t_{\text{Step}}$. If the value becomes $\leq 0$, the protein switches its state and new diffusion coefficient and $t_{\text{Change}}$ are assigned. Every 10 ms, the current location of the proteins is convoluted with a random localisation error, taken from a randomly sampled normal distribution with a localisation uncertainty of $\sigma = 0.035\,\mu m$. Each simulated protein had a pre-determined number of localisations, leading to simulated tracks ranging from 1 to 8 steps. The number of tracks of each length follows an exponential decay with a mean track length of three steps, following previous experimental observations of PAmCherry2[112]. The $D^{*}_{\text{Free}}$ value assigned to DnaA was 2.7 $\mu m^2/s$, based on its hydrodynamic radius and accounting for cytoplasmic retardation due to crowding and viscosity ($\sim 20x$)[113]. The value of the DnaA hydrodynamic radius ($\sim 2.48$ nm) was obtained using the in-silico tool HullRad[114], providing as input a predicted crystal structure of the DnaA-PAmCherry2.1 fusion mutant, generated through AlphaFold[115]/ColabFold[116]. This diffusion value is similar to previous estimates of diffusion coefficients of proteins moving within the bacterial cytoplasm[117–119]. We further confirmed this value by using the pDnaA211-PAmCherry2.1 plasmid to express DnaA211[99], a DnaA mutant impaired in DNA binding and thus solely moving in a free state. We then analysed the diffusional distributions of the protein as described above, simulating a static species ($k_{\text{Free} \rightarrow \text{Bound}}$ $= k_{\text{Bound} \rightarrow \text{Free}} = 0\,s^{-1}$) and estimating diffusion coefficients that would best fit the experimental data. In this way, we obtained a diffusion coefficient of $2.6 \pm 0.2\,\mu m^2/s$ across all growth regimes (Supplementary Fig. 4F). In our set-up, proteins bound to DNA appear as immobile[44,102]; therefore, DnaA was considered as immobile in its DNA-bound state, with a diffusion coefficient solely determined by the localisation uncertainty ($D^{*}_{\text{Bound}} = D^{*}_{\text{Immobile}} = 0.12\,\mu m^2/s$). The bound fraction of DnaA was then calculated using the obtained kinetic rates as $p_{\text{Bound}} = \frac{k_{\text{Free} \rightarrow \text{bound}}}{k_{\text{Bound} \rightarrow \text{free}} + k_{\text{Free} \rightarrow \text{bound}}}$. The result was then multiplied by 100 to display it as a percentage.

### Analysis of DnaA behaviour throughout the cell cycle

To study the difference in behaviour of DnaA across the cell cycle, we assigned to each cell a single value of average diffusion coefficient. This value was obtained by summing the diffusion coefficient of tracks a single cell contained and then dividing it by the number of tracks. We used the area to probe the different phases of the cell cycle, by plotting the obtained average diffusion coefficient over the area of the cell. We then calculated and displayed rolling medians, and 25 and 75% quantiles in R v4.4.1 with the zoo v1.8-12 package[89],

using a rolling window size of '$k = 75$'. Distributions of cell areas were also obtained.

### Flow cytometry estimation of number of origins

To obtain the average number of origins present in an *E. coli* population, both reference cells (wild-type in slow growth regime, 1 to 2 copies) and experimental cells (mutant strains in different growth regimes) samples were prepared[7]. In both cases, turbidostat cultures of *E. coli* were subjected to run-out replication. Rifampicin (150 µg/mL final concentration) and cephalexin (15 µg/mL final concentration) were added, the turbidostat fermentation was stopped and cells were left incubating with constant stirring and temperature for additional 6-6.5 h. Then, 2 mL of the final culture were first washed in the same volume of PBS, subsequently added to 18 mL of ice-cold 70% V/V ethanol and stored at 4 °C for at least 12 h[120]. Reference cells were obtained from turbidostat cultures of wild-type *E. coli* MG1655 in slow growth regime, where cells typically cycle between 1 and 2 copies of the chromosome[6]. Experimental cells were obtained for all strains harbouring the *dnaA-PAmCherry2.1* gene in different growth regimes.

Both reference and experimental cells samples were washed once with PBS and then resuspended in 1 mL of PBS in an Eppendorf tube. The membrane of the reference cells was then stained with MitoTracker™ Deep Red FM (Invitrogen™) to a final concentration of 1 µg/mL and left incubating for 1–1.5 h at room temperature, in the dark, with constant shaking. Reference cells were then washed two times with PBS and finally resuspended in 1 mL of PBS. For absolute DNA content measurement, 25 µL of reference cells were mixed with 75 µL of experimental cells and the cell mixture was stained with 20 µL of 100x Quant-iT™ PicoGreen™ dsDNA reagent (Invitrogen™) in 25% DMSO. The cells were incubated together with the nucleic acid dye for 30–60 min at room temperature in the dark. Finally, 200 µL of PBS was added to the samples.

Analysis of DNA content was performed with an Attune™ NxT Flow Cytometer (Invitrogen™) and the Attune™ proprietary software. For each sample, forward and side scatter measurements were obtained, together with emission filtered with a 530/30 nm transmission filter for DNA content measurement (BL1-H channel, voltage = 260) and emission filtered with a 695/40 nm transmission filter for MitoTracker™ measurement (YL3-H channel, voltage = 450). Reference cells were separated from experimental cells by gating the far-red emission. At least 10,000 single cells events were collected for the experimental cells. The DNA content of both the reference cells and the experimental cells populations were represented as a histogram versus fluorescence on the green channel. The intensity of the highest peak for the population of reference cells, representing two copies of the chromosomes after run-out replication, was used to quantify absolute number of origins. The intensity maximum of each peak was identified and the number of cells in a 500 A.U. range centred around the peak was used to calculate the average number of origins, asynchrony index and relative abundance of cells with one origin of each sample.

### Proteomic sample generation

To perform proteomics, 10 mL of turbidostat cultures of *E. coli* strains in different growth conditions were harvested. Cells were then washed twice in PBS before being pelleted and bacteria were lysed with 100% trifluoracetic acid (TFA) according to the SPEED protocol[121] with slight adaptations[122]. Briefly, 30 µL of 100% TFA was added to the pellet, incubated for 5 min at 55 °C and neutralised with 270 µL of 2 M Tris (pH was not adjusted). Lysed samples were quantified via Bradford assay. 50 µg total protein amount per sample were reduced and alkylated (9 mM tris(2-carboxyethyl)phosphine (TCEP) and 40 mM chloroacetamide (CAA)). Protein digestion was performed with a Trypsin (Roche)-to-protein ratio of 1:50 overnight at 37 °C. Samples were

desalted by solid-phase extraction (C18) and stored after lyophilisation at −80 °C.

## Proteomic data acquisition

Peptides were analysed on a Vanquish Neo UHPLC (micro-flow configuration; Thermo Fisher Scientific) coupled to an Orbitrap Exploris 480 mass spectrometer (Thermo Fisher Scientific). Around 25 µg peptides were applied directly onto a commercially available Acclaim PepMap 100 C18 column (2 µm particle size, 1 mm ID x 150 mm, 100 Å pore size; Thermo Fisher Scientific) and separated using a 60 min linear gradient ranging from 3% to 28% solvent B (0.1% FA, 3% DMSO in ACN) in solvent A (0.1% FA, 3% DMSO in HPLC grade water) at a flow rate of 50 µL/min. The mass spectrometer was operated in data-independent acquisition (DIA) mode. MS1 full scans (360–1300 m/z) were acquired with a resolution of 120,000, a normalized AGC target value of 100% and a custom maximum injection time. The cycle time was set to 3 seconds. MS2 scans (200–1800 m/z) were acquired over 40 DIA segments with widths adjusted to the precursor density (see Supplementary Table 8 in Wu et al.[123]). The scan resolution in the Orbitrap was set to 15,000 with a normalized AGC target value of 100% and a custom maximum injection time. The HCD collision energy was set to 30%.

## Proteomic data analysis

The mass spectrometric raw files were analyzed with DIA-NN[124] (version 1.8) using the Uniprot reference fasta file for *E. coli* K-12 (UP000000625, download 04.06.2023, 4403 protein entries) appended with common contaminants. FASTA digest for library-free search and deep learning-based spectra was enabled. N-term M excision, C carbamidomethylation, Ox(M), and Ac(N-term) were activated. The peptide length range was set from 7 to 30, precursor charge ranged from 2 to 6, precursor m/z ranged from 360 to 1300 and fragment ion m/z ranged from 200 to 1800. Usage of isotopologues and match-between runs (MBR) were enabled. Precursor FDR was set to 1%. All other parameters were kept as default. The DIA-NN report table "report.pg_matrix" was imported into the data visualisation platform omicsViewer[125], where proteins were filtered for being quantified in all three replicate measurements of at least one experimental condition. Finally, remaining missing values in the data matrix were imputed by a protein-specific constant value, which was defined as the lowest detected value over all samples divided by two. Additionally, a maximal imputed intensity value was defined as 15% quantile of the protein distribution from the complete dataset.

## LacZ β-galactosidase assay

To use the activity of the LacZ β-galactosidase as a proxy for DnaA-ATP/DnaA-ADP ratio, 10 mL of turbidostat cultures of *E. coli* strains carrying the pdnaAp-LacZ plasmid in different growth conditions were harvested. Cells were permeabilised using B-PER™ Bacterial Protein Extraction Reagent (ThermoFisher Scientific™) following manufacturer instructions. Permeabilised cells were then diluted 10 times before addition of *o*-nitrophenyl-β-D-galactopyranoside to a final concentration of 0.7 mg/mL. The β-galactosidase activity was determined as previously described[126]. Briefly, absorbance at 420 nm was measured using a Synergy H1 microplate reader (BioTek™) and then normalised for the average signal obtained from the wild-type strain in the same growth condition.

## Reporting summary

Further information on research design is available in the Nature Portfolio Reporting Summary linked to this article.

## Data availability

All the localisation data and related brightfield images used during sptPALM in this study have been deposited on Zenodo (https://doi. org/10.5281/zenodo.13939193). A list of the matches to the DnaA box sequences on the genome of all analysed *E. coli* strains and all cellular and single-molecule parameters generated in this study are provided in the Source Data file. The mass spectrometric raw files, as well as the DIA-NN output files have been deposited to the ProteomeXchange Consortium via the PRIDE partner repository and can be accessed using the identifier PXD064841. All the mutant *E. coli* strains carrying the *dnaA-PAmCherry2.1* locus are available upon request. Source data are provided with this paper.

## Code availability

Scripts used in this study to analyse position of DnaA boxes across genomes and to generate single-particle tracking data have been deposited on Zenodo (https://doi.org/10.5281/zenodo.13939193). The scripts used to analyse the disposition of DnaA boxes on the genomes of *E. coli* and *S. enterica* are also available on GitHub (https://github.com/stephkoest/Ecoli_titration).

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

## Acknowledgements

We want to thank Dr. Mareike Berger for the inspiring discussions at every stage of the study. We would like to thank prof. Anders Løbner-Olesen (University of Copenhagen) for kindly providing us with the *E. coli* strains carrying deletions for *datA, DARS1* and *DARS2*, alone or in combination. We would like to thank Franziska Hackbarth for excellent technical assistance and maintenance of mass spectrometers. L.O. thanks Ricardo Villegas-Warren for his help in generating the predicted crystal structure of DnaA-PAmCherry2.1, Tim Althuis for his guidance in preparing the samples for the proteomics analysis and Dr. Charlotte Koster for her guidance in performing the LacZ assays. L.O., N.J.C., J.vd.O and P.R.t.W. acknowledge financial support from The Netherlands Organization of Scientific Research (NWO/OCW) Gravitation program Building a Synthetic Cell (BaSyC) (024.003.019). R.H.J.S. is supported by a VIDI grant (VI.Vidi.203.074) from NWO. T.J.G.E. is supported by a European Research Council Consolidator Grant (817834), a VICI grant from the Netherlands Organization of Scientific Research (VI.C.192.016) and a Volkswagen Foundation 'Life' grant (96725). The Exploris 480 mass spectrometer was funded in part by the German Research Foundation (INST 95/1435-1 FUGG).

## Author contributions

L.O. designed the research. L.O., C.L. and M.L. performed experiments. S.K. generated in silico data. S.K., J.H. and C.L. wrote the scripts used to generate and analyse data. L.O., S.K. and C.L. analysed the data. L.O. and S.K. drafted the manuscript and produced figures. L.O., J.H., R.H.J.S. supervised the work. L.O., S.K., C.L., M.L., N.J.C., T.J.G.E., J.v.d.O., P.R.t.W., J.H., R.H.J.S. contributed to editing the manuscript.

## Competing interests

The authors declare no competing interests.
