## [Transparent Peer Review file · Nature Communications]

The *Escherichia coli* replication initiator DnaA is titrated on the chromosome.

Corresponding Author: Dr Raymond Staals

Version 0:

Reviewer comments:

Reviewer #1

(Remarks to the Author)

In this work, the authors were able to label the DnaA protein of *E. coli* with a fluorescent protein without significantly perturbing its activity in vivo. This has allowed them to carry out super-resolution microscopy to measure the diffusion constant of the fluorescently labeled protein and therefore estimate the fraction of DnaA that is free in the cytoplasm from the amount that is bound to the DNA and the kinetics of association and dissociation. The aim is to find experimental evidence for a titration-based control mechanism of the initiation of DNA replication.

The first observation is that the bound fraction and the kinetic rates depend on the growth rate of the cells.

Despite the increase in DNA, and the number of DnaA's binding sites, per cell at fast growth, cell volume increases faster, resulting in a decrease in DNA concentration with increasing growth rate (Bremer, H., and Dennis, P. (1996). Modulation of Chemical Composition and Other Parameters of the Cell by Growth Rate. In *Escherichia coli and Salmonella typhimurium: cellular and Molecular Biology*, F. C. Neidhardt, J. L. Ingraham, K. B. Low, B. Magasanik, M. Schaechter, and H. E. Umbarger, eds. (American Society for Microbiology), pp. 1553–1569.).

Could the decrease in DNA concentration suffice to explain the change in DnaA mobility observed in these experiments? (Figure 3C)

How does the amount of DnaA change with growth rate? Or in the presence of the mutations? The bound fraction will change just by the presence of increasing concentrations of DnaA, as the high affinity sites on the DNA become saturated and more DnaA is found in the cytoplasm.

Is it possible to use the fluorescently labeled DnaA (not necessarily this fluorescent protein, which is specific to the single molecule tracking experiments) to estimate the DnaA concentration and therefore the total DnaA to DNA ratio in the different growth conditions, at the single cell level?

How does the DnaA-ATP to DnaA-ADP ratio change? The total amount of DnaA is not proportional to growth rate, Zheng et al show that the population average DnaA concentration increases both at faster and at slower growth (Zheng et al Nat Micro. 2020). Skarstad has also shown that there is an excess of DnaA at slow growth. The amount of DnaA-ATP relative to DnaA-ADP however is more difficult to measure. At slower growth rates RIDA plays a smaller role, because it is associated with ongoing DNA replication, therefore the effect of the *datA* mutation is stronger, as an alternative pathway to increase ATP hydrolysis rate (see the work of Skarstad and Lobner-Olesen).

Are the results of these experiments affected by the DnaA-ATP to DnaA-ADP ratio?

If there were an increased fraction of DnaA-ATP at slow growth, it could also result in a higher bound fraction, and the deletion of *datA* would increase the fraction of DnaA-ATP at slow growth. This would agree with the results shown in Figures 3C and 5B. Suggesting that the results might also reflect the change in the amount of free DnaA-ATP.

Are there other experiments that can help determine the binding activity of DnaA-ATP vs DnaA-ADP? Did the authors try to do a control with the DnaACOS mutant?

More context would be useful to compare these results with previous data on the behavior of DNA-binding proteins and how these results can help uncover a possible role of titration in DNA regulation.

How does the data on DnaA compare with that of other DNA binding proteins? (Stracy, M., Schweizer, J., Sherratt, D.J.,

Kapanidis, A.N., Uphoff, S., and Lesterlin, C. (2021). Transient non-specific DNA binding dominates the target search of bacterial DNA-binding proteins. *Molecular Cell* 81, 1499-1514.e6. <https://doi.org/10.1016/j.molcel.2021.01.039>).

DnaA is a DNA binding protein; therefore, it is not surprising that a significant amount of it is found on the DNA, this result in itself is not evidence for a titration effect. The main challenge is to determine whether DnaA binding to the genome can have an effect on its activity for the initiation of DNA replication.

Titration on genomic DNA is a general property of DNA binding proteins and can particularly affect the activity of transcription factors. The study of Gao et al shows one way in which one can study the role of titration, by adding what they call “decoy” DNA binding sites (Gao, R., Helfant, L.J., Wu, T., Li, Z., Brokaw, S.E., and Stock, A.M. (2021). A balancing act in transcription regulation by response regulators: titration of transcription factor activity by decoy DNA binding sites. *Nucleic Acids Res* 49, 11537–11549. <https://doi.org/10.1093/nar/gkab935>). In the case of DnaA these could be the DnaA sites found at the *mioC* promoter for example.

DnaA is not very different from other transcription factors, and titration could play a role that is independent from initiation of DNA replication. Indeed, the coupling of DnaA autoregulation of its own expression with a genome titration effect would help maintain the amount of active DnaA proportional to the amount of DNA in the cell. This could be useful in the regulation not only of initiation but also in the expression of DNA repair genes for example (Wurihan, Gezi, Brambilla, E., Wang, S., Sun, H., Fan, L., Shi, Y., Sclavi, B., and Morigen, (2018). DnaA and LexA Proteins Regulate Transcription of the *uvrB* Gene in *Escherichia coli*: The Role of DnaA in the Control of the SOS Regulon. *Front Microbiol* 9, 1212. <https://doi.org/10.3389/fmicb.2018.01212>).

The results are presented with little context of previous experimental studies on the role of titration in DnaA activity.

It is known that the activity of the DnaA protein plays a crucial role in the initiation of DNA replication. It has been shown that its nucleotide bound state determines whether it is in an active (ATP bound) or inactive (ADP bound form), and several of the molecular mechanisms that can regulate the ratio of DnaA-ATP to DnaA-ADP have been identified. Previously, different theoretical models have considered an additional regulatory mechanism, the titration of DnaA on the DNA, which can change the amount of DnaA-ATP that is available to bind to the origin, and to the other regulatory sites.

Earlier work created an artificial titration situation by adding different amounts of DnaA binding sites. These experiments showed that in response to titration on additional sites, the expression of the DnaA gene increased due to derepression. Thus, compensating for the decreased DnaA activity. (Hansen, F.G., Koefoed, S., Sorensen, L., and Atlung, T. (1987). Titration of DnaA protein by *oriC* DnaA-boxes increases *dnaA* gene expression in *Escherichia coli*. *EMBO J* 6, 255–258.) [This citation and the two below should be added in line 73]. In fact, they note that the cell’s growth rate is quite robust in the presence additional DnaA sites or copies of *oriC* on plasmids and minichromosomes. The derepression of the expression of the DnaA gene suggests that titration using *oriC* DnaA boxes decreases specifically the activity DnaA-ATP over that of DnaA-ADP.

Later, the Skarstad group tried to measure the effect of titration by increasing the copy number of the *datA* site, which we now know that not only binds DnaA but also induces hydrolysis of its ATP (Morigen, Lobner-Olesen, A., and Skarstad, K. (2003). Titration of the *Escherichia coli* DnaA protein to excess *datA* sites causes destabilization of replication forks, delayed replication initiation and delayed cell division. *Mol.Microbiol.* 50, 349–362; Morigen, Boye, E., Skarstad, K., and Lobner-Olesen, A. (2001). Regulation of chromosomal replication by DnaA protein availability in *Escherichia coli*: effects of the *datA* region. *Biochim.Biophys.Acta* 1521, 73–80.). These studies showed that increased copies of *datA* could decrease the activity of DnaA, affecting initiation of DNA replication and the replication of minichromosomes when present. They also showed that the autoregulation of DnaA expression could compensate for the titration effect, but only when the *datA* copy number was low. Therefore, it is possible that *datA* can both titrate and inactivate DnaA.

However, the thing that was missing from these studies was a way to control the timing of the increase in copy number of the additional DnaA sites, since they were placed on plasmids and not on the chromosome.

As the authors of this paper point out, the preferential positions of the high affinity sites near the origin means that their copy number increases right after DNA replication has begun. During a window of time of approximately 10 minutes following initiation, the expression of the newly replicated *dnaA* gene is repressed by the *SeqA* protein, therefore delaying a possible derepression. These sites could therefore contribute to a transient decrease in DnaA activity (together with dilution rate and ATP hydrolysis via *RIDA* and *datA*) while the origin is also bound by *SeqA*.

The authors searched the genome for DnaA binding sites that matched the consensus sequence.
Line 108-109 what are the other motifs used ?

Line 110, of the boxes found between centrosomes 98 and 5, how many are part of the origin, the *mioC* promoter and the DnaA promoter? It might be interesting to know whether they are near the origin because of the function of the genes (Sobetzko, P., Travers, A., and Muskhelishvili, G. (2012). Gene order and chromosome dynamics coordinate spatiotemporal gene expression during the bacterial growth cycle. *Proceedings of the National Academy of Sciences* 109, E42–E50. <https://doi.org/10.1073/pnas.1108229109>.) or because of the need for titration sites.

DnaA-ATP can bind to the DNA as dimers and oligomers, but not DnaA-ADP (Erzberger, J.P., Mott, M.L., and Berger, J.M. (2006). Structural basis for ATP-dependent DnaA assembly and replication-origin remodeling. *Nat Struct Mol Biol* 13, 676–683. <https://doi.org/10.1038/nsmb1115>.). For example, Stringer et al (*Microbiology* 2024) recently identified a number of such

closely spaced sites, which they show bind DnaA better than individual sites in vivo. Maybe this dimer motif would be a better predictor of the pattern of DnaA binding along the genome. Indeed, Stringer et al show that single sites have a very weak signal (with the caveat that in their experiments cells were grown in LB, where the DNA is at a low concentration and DnaA has a higher number of possible targets, despite its higher concentration). Does the replication of these sites decrease the amount of DnaA-ATP available to inhibit a reinitiation of DNA replication, which would otherwise cause genome instability via fork collapse?

If titration via the sites near the origins worked mainly by trapping DnaA-ATP after initiation of DNA replication has taken place, then one should see an increased fraction of DnaA on the chromosome at a specific cell volume following the initiation of DNA replication.

Would it be possible to note on the cell area plots in Figure 3D the cell size for initiation of DNA replication?

The initiation of DNA replication seems to be correlated better with cell Volume rather than cell Area (see for example: Govers, S.K., Campos, M., Tyagi, B., Laloux, G., and Jacobs-Wagner, C. (2024). Apparent simplicity and emergent robustness in the control of the Escherichia coli cell cycle. *Cell Syst* 15, 19-36.e5. <https://doi.org/10.1016/j.cels.2023.12.001>). Would the result change if plotted as a function of cell volume?

Why is the average diffusion coefficient plotted as a function of cell size instead of the kinetic rates or the bound fraction? Maybe there are not enough tracks per cell to measure the bound fraction? Could the cells be binned as a function of their size?

The data presented in this paper shows that DnaA binds to the DNA, and that the mutations of the genomic sites affecting its activity also change the fraction that is bound, however it does not support a specific role of titration in DNA replication regulation.

In the specific case of DnaA, titration could play a double role: (i) in creating a step-like function increase in the activity of DnaA and therefore result in improved synchrony of initiation at the different origins and (ii) in decreasing the activity of DnaA after initiation has taken place by the increase of DnaA sites resulting from the replication process. However, data on the role of titration has been elusive. The difficulty in measuring a specific effect of titration on initiation of DNA replication is due to (i) the difficulty in measuring the activity of DnaA in vivo, due to the presence of both the ATP and the ADP bound forms with different DNA binding properties (ii) the redundancy of multiple regulatory mechanisms required to keep the initiation process coupled to cell growth under all growth conditions, some being more important under slow growth for example or in response to stress, (iii) DnaA's binding sites on the genome also have other specific functions, apart from the origin, acting to regulate gene expression or the nature of the nucleotide bound to DnaA itself. The current title just describes what is known for most DNA binding proteins, they will spend a significant time bound more or less specifically to the DNA. In itself, this result is not very surprising or useful to understand how DnaA activity is regulated. Therefore, more experiments are needed in order to be able to say that titration of DnaA plays a specific role in the regulation of DNA replication.

Additional concerns:

Lines 37-39. In the introduction, and elsewhere, the authors put on the same plane conclusions obtained via experimental methods and theoretical models, which however still need to be experimentally confirmed. In addition, they cite results obtained as a function of growth conditions, which however have no data about the cell cycle.

The same can be said on line 146, the cited paper is a mathematical model, the text should reflect this. For example: "a feature that could help initiator titration during fast growth".

There is nothing wrong with theoretical models, they are very useful to interpret data and guide future experiments, and however, one cannot say that DnaA expression is balanced with cell growth because a model says so. In general, there is no point of invoking two different models, the switch and titration models, as if there were two exclusive realities. It seems an outdated description of what we know about the regulation of DnaA activity. For example, the *datA* site can probably do both. An alternative would be to say that it is known that regulation of the DnaA-ATP to DnaA-ADP ratio is important for regulation of initiation, it remains to be determined whether titration can also play a role, and in which growth conditions.

Line 60: both models can be based on the accumulation of DnaA-ATP determining the moment of initiation of DNA replication. The difference between the models is that the accumulation of DnaA-ATP is delayed by the binding of the genomic sites, until it passes a threshold level, and on the mechanisms that can inhibit re-initiation after duplication of the origin region. In both cases, there is a peak in the DnaA active form. The amount of active form is the amount of DnaA-ATP that is available to bind to the origin, or other sites on the genome, as measured by Iuliani et al. (2024).

Line 188-189, incomplete sentence. It is understood from the context, but it might be better to complete it by specifying which mechanism is being referred to.

Line 191-192 does this refer to fully consensus sites? Dimer sites?

Line 200, maybe it would be more clear to say something like: cells initiate a new round of replication upon reaching a well-defined size that is higher in the mutants than in the wild type. Consistent with a decrease in DnaA activity. (which is the

opposite of what one would expect if there were less titration of DnaA but especially less conversion of DnaA-ADP to DnaA-ATP)

Lines 202-203, the phenotype of larger cell size could also be due to a longer C period and a higher number of origins is consistent with rifampicin resistant initiations in the flow cytometry experiments due to increased amounts of DnaA-ATP and delayed division (see the following and references therein: Flåtten, I., Fossum-Raunehaug, S., Taipale, R., Martinsen, S., and Skarstad, K. (2015). The DnaA Protein Is Not the Limiting Factor for Initiation of Replication in *Escherichia coli*. *PLOS Genetics* 11, e1005276. <https://doi.org/10.1371/journal.pgen.1005276>). Rifampicin resistant reinitiations in cells whose cell division has been inhibited by cephalixin can result from a protection of the DnaA promoter from rifampicin by DnaA itself (see Skarstad).

Lines 235-236, these results would also be consistent with a scenario where there is more DnaA-ATP in the cell (because of less RIDA at slow growth and no *datA* (also see Skartad's paper above)) and all the sites are saturated.

Line 296, Shank et al's study is on *B. subtilis* DnaA, long distant cousin with very different mechanisms of regulation. This is not a valid comparison, or at least it should be specified.

Lines 300-301, unfortunately the results of this study do not allow for an improved understanding of the impact of titration on DNA replication compared to previous work.

Lines 306-307: (i) balanced biosynthesis has not been shown directly; furthermore, the concentration of DnaA has been shown to change as a function of growth rate (Zheng et al 2020) (ii) the speed of DNA replication is not constant as a function of growth rate (Michelsen, O., Teixeira de Mattos, M.J., Jensen, P.R., and Hansen, F.G. (2003). Precise determinations of C and D periods by flow cytometry in *Escherichia coli* K-12 and B/r. *Microbiology (Reading)* 149, 1001–1010. <https://doi.org/10.1099/mic.0.26058-0>.) see also Skarstad's work on the *datA* mutation.

Line 311, the rate at which new DnaA boxes are replicated per origin is different from the speed at which new boxes are replicated -per cell- or -per volume- due to the increase in the origin to terminus ratio with growth rate.

Line 320-321, the presence of the high affinity boxes towards *oriC* just means that the genes regulated by DnaA need to be near the origin in order to keep their copy number increasing with growth rate. This fact alone does not support the existence of a titration-based form of control.

There is no evidence that these boxes are responsible for the binding of over half of DnaA proteins in the cell.

Lines 338-339, see the comments above about lines 202-203.

Lines 345-347, the regulatory mechanisms at fast and slow growth are not the same, therefore one should not expect the same phenotype.

While the cells were grown at 37°C, the measurements were carried out at room temperature. The binding of DnaA-ATP to DNA is temperature dependent, binding with a lower affinity at lower temperatures (Saggiaro, C., Olliver, A., and Sclavi, B. (2013). Temperature-dependence of the DnaA-DNA interaction and its effect on the autoregulation of *dnaA* expression. *Biochem. J.* 449, 333–341. <https://doi.org/10.1042/BJ20120876>.) This suggests that these measurements could overestimate the amount of free DnaA in the cell.

The ideal, but maybe impossible, experiment would be to measure the diffusion constant of DnaA of cells growing in their growth medium, instead of PBS, at a temperature between 30 and 37°C, as a function of time and thus follow the change in diffusion constant in the same cell as the cell size increases.

Even more ideally, one would need a fluorophore specific to the nucleotide bound state of DnaA, something that fluorescence only when DnaA dimerizes or oligomerizes (FRET?).

Several citations in the methods are wrong. Something must have happened when the methods were pasted into the main document.

(Remarks on code availability)

My level of expertise does not allow me to properly review code.

Reviewer #2

(Remarks to the Author)

This paper provides experimental evidence supporting the titration of the replication initiator protein DnaA in *Escherichia coli* and explores how elements regulating the DnaA-ATP/DnaA-ADP switch (*datA*, DARS1, and DARS2) affect DnaA titration and initiation. The authors address three main questions:

1. Distribution of DnaA boxes on the chromosome: Using a comparative genomic approach, the researchers analyzed the *E. coli* MG1655 genome alongside other *E. coli* strains, a related species (*Salmonella enterica*), and random permutations. They identified a conserved enrichment of DnaA boxes near the origin of replication (*oriC*), suggesting a universal bias.

2. Fraction of DnaA bound to the chromosome: Employing single-particle tracking photoactivatable localization microscopy (sptPALM), the authors measured the mobility of individual DnaA proteins. They found that in wild-type *E. coli*, 55-75% of DnaA proteins are chromosome-bound, with the bound fraction decreasing as growth rate increases.

3. Effects of deleting switching elements (*datA*, *DARS1*, *DARS2*): In a mutant lacking all three elements ($\Delta 3D$), the bound fraction of DnaA remained at ~60% across growth regimes. The researchers also noted that the combined effects of these deletions on DnaA binding were not simply additive.

This study fills a critical gap by providing direct quantitative evidence of DnaA titration in live cells and highlights its importance as a major mechanism for replication initiation control. The experimental results are robust, though the interpretations of growth-regime-specific mechanisms are somewhat intricate in the Discussion section. A straightforward takeaway is that initiator titration is significant in all conditions and mutants tested (with over 50% of DnaA proteins bound), but its strength varies moderately with growth rate and the presence of switching elements. Overall, we recommend publication in *Nature Communications* once the authors have addressed the following minor concerns.

Minor concerns:

1. Figure 2A lacks clarity regarding the meaning of the radius in the circular plot. From the context, it seems to be something like the number of DnaA boxes per unit length of the chromosome or the DnaA box density, but this is not explicitly defined. Please define the radius of the circle in Figure 2A explicitly in the main text and in the figure legend to avoid misinterpretation.

2. Figure 2B: are these all consensus DnaA boxes (i.e., TTWTNCACA)? Additionally, providing the total number of DnaA boxes in each strain would add valuable context.

3. Line 136-137: "...rejecting previous assumptions on their homogenous distribution.12,19" This is inaccurate and unnecessarily strong. First, neither Ref. 12 nor Ref. 19 assumed a homogeneous distribution in their study. Rather, both Ref. 12 and Ref. 19 considered the biased distribution of DnaA boxes near *ori*. Ref. 19 only stated that high-affinity DnaA boxes are randomly distributed, but not for all consensus ones. Second, while heterogeneity exists, the distribution in Figure 2A does not seem to significantly deviate from a homogeneous distribution. Based on this result, it still seems valid to assume a homogenous distribution as a zeroth-order approximation in some physical models such as in Ref. 34 and 35. The emphasis on heterogeneity seems overstated, raising the question of whether incorporating it is crucial for future modeling efforts. A similar overstatement appears in Line 110 and should be addressed for consistency.

4. Line 213-214: "...while the mobility of DnaA increased throughout the whole cell cycle (Figure 4E)." This statement seems inconsistent with the data. Indeed, in Figure 4E top panel (slow growth), the diffusivity only increases when the cell size is small, and becomes flat for most of the cell cycle. We suspect that the short increasing phase after cell birth is right before initiations when cells need to accumulate a large amount of free DnaA to have enough DnaA-ATP because DnaA-ATP fraction is low due to the absence of *DARS1* and *DARS2* in this mutant.

5. Line 369 and 383, also Figure 6: The concept of controlling the first replication event is underexplained and we do not understand the meaning of distinguishing it from other replication events.

6. Including supplementary videos showing examples of DnaA movement with different diffusivity would greatly aid in visualizing the findings and improving reader comprehension.

(Remarks on code availability)

Reviewer #3

(Remarks to the Author)

This study used newly developed single cell methodology and fluorescence-labeled DnaA to quantitatively analyze intracellular movements of DnaA in growing cells with various doubling times. Also, the authors mathematically deduced DnaA box density for the whole genome of *E. coli* species. Results of these two are not necessarily linked together in this study, but are proposed to be consistent with the idea that the initiator titration model is partly valid for regulation of replication initiation in slowly growing cells. Challenging experiments were well performed and interpretation to the results are overall reasonable. However, I have some questions as follows and am not very convincing to reliability of the values of DnaA bound fraction and a part of the conclusion.

1. Figure 2 and Supplementary Figure 2AC: Include explanation to the numbers of DnaA boxes (20, 40, 60 and 100) indicated by circles in Results or Figure legend. Methods describes 'We calculated the number of boxes per position in a sliding window approach by moving over the scaled chromosome in a 10 centisome window, taking centisome steps.' However, a brief explanation is also required in Results or Figure legend.

2. Relating above: I feel that the 10 centisome window is too long. Show several data using several different lengths of the window. Comparing those data are required to make a conclusion for DnaA box density bias

3. Relating above: The oriC region should have exceptionally high DnaA box density. This should affect the density levels of the oriC-proximal regions by the window. This is not preferable for the intended analysis of the initiator titration model. Thus, these calculations should be done excluding the oriC region.

4. Figure 3A: The codon no. and domain structure should be indicated to dnaA.

5. Figure 3A: The cellular number of DnaA-PAmCherry2.1 molecules in MG1655 dnaA-PAmCherry2.1 cells growing in the indicated three conditions should be shown by immunoblotting or similar experiments. Also, are DnaA-PAmCherry2.1 molecules stable in cells? Degradation products, if present at a certain level, would adversely affect the microscopic data.

6. Figure 3C and Results lines 156-164: Understanding how the bound fractions (%) are determined is very difficult. At least brief explanation is required in this part, although Methods includes detailed explanation (lines 587-595).

7. Relating above: The authors assume certain values of diffusion constants/coefficients for distinguishing the bound and free DnaA molecules (2.7 microm²/s and 0.12 microm²/s), based on theoretical assumptions. This is OK, but other different values for these parameters should be used for calculations for comparison.

8. Relating above: The diffusion constants of free DnaA should be experimentally deduced using DnaA mutant defective in DnaA box binding. DnaA-PAmCherry2.1 variant with such a mutation should be expressed in MG1655 cells and be analyzed.

9. The chapter of lines 187-241 and Figure 4 and 5: The duration of chromosomal replication is thought to be relatively constant in cells with different growth rate at the same temperature. So, there is another hypothesis; i.e., In delta-3D cells growing slowly, the ATP-DnaA level is inferred to be elevated because RIDA works only for a limited period in the long doubling time. In contrast, in delta-3D cells growing at intermediate and fast rates with the shorter doubling time, RIDA works relatively more efficiently, reducing the ATP-DnaA level and repressing overinitiation. This hypothesis can explain the cause of the overinitiation in delta-3D cells growing slowly. The increased oriC copies move for partition in cells, which enhances movement of DnaA bound to the chromosome. In panel D, the indicated 'bound' DnaA does not necessarily mean DNA-bound molecules but indeed does slowly moving molecules. Such chromosome dynamics for partition of the oriC copies and then chromosomes can explain partial enhancement of movements of DnaA molecules even bound to the chromosome, which can be recognized as free DnaA molecules in the diffusion constants/coefficients. Note that this hypothesis is valid also to explain the results of Figure 5. Thus, these data do not necessarily mean the DnaA titration is reduced in the delta-3D cells growing slowly. Thus, the chapter should be revised fundamentally.

10. Relating above: lines 235-231. If overinitiation occurs and the initiator titration model is valid, the elevated copies of the oriC-proximal region (as shown in Figure 2) should titrate DnaA more. Thus, logic of this part is inconsistent. The datA, DARS1 and DARS2 are not included in the DnaA box-dense region (Figure 2). Overall plot of this study is confusing.

11. Relating above: Efficient titration of DnaA by datA, DARS1 and DARS2 is difficult to imagine because of the limited number of DnaA boxes. Indeed Ogawa et al, demonstrates that datA binds at most 30 DnaA molecules (Genes to Cells (2009) 14, 329–341). However, the authors should perform ChIP experiments to assess DnaA binding to these loci and oriC (as a control) in cells with different growth rates.

(Remarks on code availability)

Version 1:

Reviewer comments:

Reviewer #1

(Remarks to the Author)

The authors have addressed most of the comments, however there are still a couple of issues with this new version for the manuscript.

Line 382: "Both the fliC and glpD genes are known to be positively regulated by DnaA. As such, it is surprising that mutations that lower the DnaA-ATP fraction and thus its capacity to oligomerise on DNA would result in their overexpression."

The authors should cite the following study : Stringer et al (Microbiology 2024): "We did not detect an association of DnaA with previously described sites upstream of aldA, glpD, fliC, proS and iraD, or within guaB, strongly suggesting that the associated genes are not part of the DnaA regulon."

The authors do not describe how they define the DnaA regulon. Did they compare the DnaA binding sites found by Stringer et al. in addition to their bioinformatics search? How many of these are in promoter regions? How do these compare to the protein expression profiles?

In their rebuttal the authors say: Our analysis did not reveal major changes in DnaA abundance at different growth rates or in the different genetic backgrounds used in our study (Supplementary Figure 4D, 5D and 6C). Still, we refer to Zheng and colleagues (Nat. Micro., 2020) for a more precise estimation of these numbers across growth rates.

From Materials and Methods:

“Growth conditions to achieve different growth regimes were: M9 supplemented with 4 g/L glucose and 1x RPMI 1640 amino acids at 37 °C for fast growth regime; M9 supplemented with 4 g/L succinate and 1x RPMI 1640 amino acids at 30 °C for intermediate growth regime; and M9 supplemented with 4 g/L acetate at 37 °C for slow growth regime.”

I refer the authors to Figure 3 from Zheng et al as an example of how temperature might not be the best way to achieve different growth rates for comparison of DnaA activity. They should cite them and point out in their results section that the intermediate growth rate was obtained by lowering the temperature. In addition to the known temperature dependence of DnaA activity.

The authors should cite the following study when comparing fast and slow growth conditions: Stepankiw, N., Kaidow, A., Boye, E., and Bates, D. (2009). The right half of the Escherichia coli replication origin is not essential for viability, but facilitates multi-forked replication. *Mol Microbiol* 74, 467–79. doi: 10.1111/j.1365-2958.2009.06877.x

In the rebuttal the authors say:

“We also believe that DnaA cannot just be assimilated to any other DNA binding protein: its specific form of control (near-balanced expression) is not a trait confirmed for all DNA binding protein, as well as the non-monotonic dependence of its concentration on growth rate.”

A lot of transcription factors (40%) are negatively autoregulated to maintain a near-balanced expression, it is not unique of DnaA, and the growth rate dependence of its concentration is not related to the titration effect but the increase in origins per cell.

See: Rosenfeld, N., Elowitz, M. B., and Alon, U. (2002). Negative autoregulation speeds the response times of transcription networks. *J Mol Biol* 323, 785–93. doi: 10.1016/S0022-2836(02)00994-4

“the synthesis of new titration sites per origin is set by the speed of DNA replication and is independent of growth rate. As a result, the synthesis rate of DnaA increases faster with growth rate than the synthesis rate of titration sites, leading to a reduction in the effective titration power of the chromosome.”

The synthesis of new titration sites is set by the speed of DNA replication, yes, but also by the number of sites.

The number of origins per cell increases with increasing growth rate.

It also increases at slow growth in the 3D strain. However, the bound fraction decreases and the average diffusion coefficient increases.

In the first round of review I made the following comment:

Line 311, the rate at which new DnaA boxes are replicated per origin is different from the speed at which new boxes are replicated -per cell- or -per volume- due to the increase in the origin to terminus ratio with growth rate.

Our response: We agree with the Reviewer, yet we are unsure as of how they would like us to acknowledge this fact in the text.

The -number- of titrating sites is dependent on growth rate. How does this affect the regulation of DnaA activity? It should at least be mentioned in the discussion.

(Remarks on code availability)

Reviewer #2

(Remarks to the Author)

The authors have addressed all our concerns.

(Remarks on code availability)

Reviewer #3

(Remarks to the Author)

Overall the authors made significant improvements by additional experiments and revising the text. However, the following points should be further considered.

1. Structure of DnaA-mCherry. Descriptions in the main text and Figure 3A are inconsistent. Unlike the explanation in the text (lines 171-173), Figure 3A and its legend indicated deletion of DnaA domain IIIb, in addition to DnaA domain II. Deletion of DnaA domain IIIb should be lethal. Precise structure of the dnaA-mCherry region on the genome and strain construction process should be described.

2. As I mentioned in the previous comments, I am not still fully convinced in the idea that "The decreased bound fraction and increased mobility of DnaA suggest that the contribution of the three switch control loci on titration is substantial during slow growth". (lines 295-296). Suggested experiments were not performed. This might be OK, but as the result, the idea was not reinforced. There should be a reasonable possibility that due to the lack of *datA*, the rate of DnaA-ATP hydrolysis which is caused only by RIDA and the intrinsic ATPase is somehow decreased, causing re-initiations. Although the overall level of the ATP form is decreased in the 3D mutant, the ATP-DnaA level should be elevated temporarily even in a delayed manner during the cell cycle. If the decreasing rate of the level is slowed, re-initiations can occur. These ideas and the results by Ogawa et al. (2009) (as indicated in the e previous comments) should be discussed in the text.

(Remarks on code availability)

We would like to thank all reviewers for their careful assessment of our work and the encouraging feedback. Our responses to the referees can be found in blue, whereas any cited text from the main text is in red.

Among other changes that are detailed below, please note that the revised manuscript includes proteomics analyses and assays on the activity of the β -galactosidase LacZ under the control of *dnaA* promoter for all the strains cited in the original version. The proteomics analyses were performed by Dr. Christina Ludwig from the Bavarian Center for Biomolecular Mass Spectrometry (BayBioMS) at the Technical University of Munich. We decided to add her name to the authors list. Accordingly, we added new sections to the Materials and Methods (“Proteomic sample generation”, “Proteomic data acquisition”, “Proteomic data analysis” and “ β -galactosidase activity assay”). The Discussion section has been substantially revised as a result of all the new experimental efforts we performed during the revision process.

Moreover, during the draft of the original manuscript we mistakenly plotted the DnaA box densities of *E. coli* BW2952 instead of *E. coli* MG1655 in Figure 2A and Supplementary Figure 2. Whereas the mistake did not impact any of the conclusions we reached in the original submission, we now modified the plots in Figure 2A and all panels of Supplementary Figure 2 to use data from *E. coli* MG1655 as originally intended.

Reviewer #1

In this work, the authors were able to label the DnaA protein of *E. coli* with a fluorescent protein without significantly perturbing its activity in vivo. This has allowed them to carry out super-resolution microscopy to measure the diffusion constant of the fluorescently labeled protein and therefore estimate the fraction of DnaA that is free in the cytoplasm from the amount that is bound to the DNA and the kinetics of association and dissociation. The aim is to find experimental evidence for a titration-based control mechanism of the initiation of DNA replication.

The first observation is that the bound fraction and the kinetic rates depend on the growth rate of the cells.

Despite the increase in DNA, and the number of DnaA’s binding sites, per cell at fast growth, cell volume increases faster, resulting in a decrease in DNA concentration with increasing growth rate (Bremer, H., and Dennis, P. (1996). Modulation of Chemical Composition and Other Parameters of the Cell by Growth Rate. In *Escherichia coli* and *Salmonella typhimurium*: cellular and Molecular Biology, F. C. Neidhardt, J. L. Ingraham, K. B. Low, B. Magasanik, M. Schaechter, and H. E. Umbarger, eds. (American Society for Microbiology), pp. 1553–1569.).

Could the decrease in DNA concentration suffice to explain the change in DnaA mobility observed in these experiments? (Figure 3C)

Our response: We would like to thank the Reviewer for assessing our work and for their suggestions. We agree that the decrease in DNA concentration is part of the reason for the growth rate-dependent decrease in DnaA bound fraction in *E. coli* MG1655 *dnaA-PAmCherry2.1*. In fact,

we already mentioned this consideration in the Discussion section. To increase readability, we modified the Result section to read:

The observed reduction in the bound fraction with increasing growth rate (Figure 3C) is likely the result of a faster accumulation of cellular volume. To maintain a constant intracellular concentration¹⁷, the synthesis rate of DnaA must increase with growth rate. The fact that DnaA concentrations are then higher in *E. coli* cells rapidly dividing¹⁰ also mean that the increase in DnaA expression is more pronounced at higher growth rates. In contrast, the synthesis of new titration sites per origin is set by the speed of DNA replication and is independent of growth rate⁷. As a result, the synthesis rate of DnaA increases faster with growth rate than the synthesis rate of titration sites, leading to a reduction in the effective titration power of the chromosome. These observations are consistent with previous mathematical modelling of a titration control mechanism³⁶.

How does the amount of DnaA change with growth rate? Or in the presence of the mutations? The bound fraction will change just by the presence of increasing concentrations of DnaA, as the high affinity sites on the DNA become saturated and more DnaA is found in the cytoplasm. Is it possible to use the fluorescently labeled DnaA (not necessarily this fluorescent protein, which is specific to the single molecule tracking experiments) to estimate the DnaA concentration and therefore the total DnaA to DNA ratio in the different growth conditions, at the single cell level?

Our response: A change in DnaA expression due to growth rate or the presence of mutations can indeed be the cause of the observed decrease in the bound fraction. Previously, it was shown that the intracellular concentration of DnaA increases with growth rate (Zheng *et al*, *Nat. Micro.* 2020). We used this reference to explain our observed growth rate-dependent decrease of DnaA bound fraction both in the Discussion section and in the newly added section in the Results (see response above).

As correctly mentioned, we would have to employ a different fluorescent protein for expression analysis due to the photoactivatable properties of the used PAmCherry2.1. Using a different fluorescent protein is likely to generate a new phenotype, as indicated by the different impact of the various fluorescent proteins we tested here (Supplementary Figure 3B), making comparisons to the strain carrying the DnaA-PAmCherry2.1 allele unreliable. Because of these concerns, we decided to perform proteomic analysis on all the strains to obtain insights into the relative levels of DnaA. Our analysis did not reveal major changes in DnaA abundance at different growth rates or in the different genetic backgrounds used in our study (Supplementary Figure 4D, 5D and 6C). Still, we refer to Zheng and colleagues (*Nat. Micro.*, 2020) for a more precise estimation of these numbers across growth rates.

How does the DnaA-ATP to DnaA-ADP ratio change? The total amount of DnaA is not proportional to growth rate, Zheng *et al* show that the population average DnaA concentration increases both at faster and at slower growth (Zheng *et al* *Nat Micro.* 2020). Skarstad has also shown that there is an excess of DnaA at slow growth. The amount of DnaA-ATP relative to DnaA-ADP however is more difficult to measure.

At slower growth rates RIDA plays a smaller role, because it is associated with ongoing DNA replication, therefore the effect of the *datA* mutation is stronger, as an alternative pathway to increase ATP hydrolysis rate (see the work of Skarstad and Lobner-Olesen).

Are the results of these experiments affected by the DnaA-ATP to DnaA-ADP ratio?

If there were an increased fraction of DnaA-ATP at slow growth, it could also result in a higher bound fraction, and the deletion of *datA* would increase the fraction of DnaA-ATP at slow growth. This would agree with the results shown in Figures 3C and 5B. Suggesting that the results might also reflect the change in the amount of free DnaA-ATP.

Our response: We agree that the ratio of DnaA-ATP to DnaA-ADP could also affect the titration of DnaA on the chromosome of *E. coli*. Due to the technical challenges associated with directly measuring the DnaA-ATP/DnaA-ADP ratio, we decided to collect additional insights by using the expression from the DnaA promoter as a readout of the DnaA-ATP/DnaA-ADP ratio (Charbon *et al*, *Mol. Microbiol.* 2011; Boesen *et al.*, *Proc. Natl. Acad. Sci. U.S.A.* 2024). The rationale of this experimental setup is that the *dnaAp* promoter is more strongly repressed by DnaA-ATP compared to DnaA-ADP. As such, the expression of a reporter gene (e.g. *lacZ*) placed downstream of this promoter varies with the ratio of DnaA-ATP/DnaA-ADP, with transcription increasing when the ratio decreases and more DnaA-ADP is present. We thus generated a low-copy number plasmid expressing the LacZ β -galactosidase via the DnaA promoter and measured the LacZ activity in all the original strains and growth conditions. Whereas the differences in the copy number of the plasmid-borne *lacZ* gene in different growth rates prevents us to make exact comparisons between them, it is possible to compare differences in LacZ activity, and thus DnaA-ATP/DnaA-ADP levels between different genetic backgrounds (Supplementary Figures 4E, 5B and 6B).

The single deletion mutants that we analysed in this study however provide insights on how the titration of DnaA is impacted by the nucleotide it is bound to. The deletion of *datA* caused an increase in DNA content (Figure 5C) and lower LacZ activity (Supplementary Figure 6B), both consistent with an increase in the DnaA-ATP fraction. At the same time, the bound fraction of DnaA increased. Conversely, the deletion of *DARS1* led to a decrease in DNA content (Figure 5C) and higher LacZ activity (Supplementary Figure 6B) consistent with a decrease in the DnaA-ATP fraction. In the same background, the bound fraction of DnaA decreased. We included these observations in the revised Results section, which now reads:

Our analysis of single deletion mutants highlighted a correlation between the DnaA-ATP/DnaA-ADP ratio and the bound fraction of DnaA. Specifically, mutations that led to a higher fraction of DnaA-ATP ($\Delta datA$) caused more DnaA to be titrated on the chromosome, while the opposite was observed in the $\Delta DARS1$ mutant strain. Such changes can be explained by the fact that DnaA-ATP better interacts with low-affinity DnaA boxes^{19,20,30} than DnaA-ADP and can form dimers or oligomers⁵⁷. Differences in DNA content and removal of specific loci also change the average number of DnaA boxes in the cell.

Are there other experiments that can help determine the binding activity of DnaA-ATP vs DnaA-ADP? Did the authors try to do a control with the DnaACOS mutant?

Our response: We did not try to use DnaAcos as a control, as the expression of DnaAcos has been reported to be lethal in *E. coli* due to over-initiation of DNA replication (Katayama, *J. Biol. Chem.*, 1994) even in presence of wild-type DnaA (Katayama and Kornberg, *J. Biol. Chem.*, 1994). Moreover, DnaAcos is impaired in its ability to bind nucleotides. We thus believe that the DNA-binding behaviour of such mutant cannot be directly related to the one of DnaA-ATP or DnaA-ADP.

More context would be useful to compare these results with previous data on the behavior of DNA-binding proteins and how these results can help uncover a possible role of titration in DNA regulation. How does the data on DnaA compare with that of other DNA binding proteins? (Stracy, M., Schweizer, J., Sherratt, D.J., Kapanidis, A.N., Uphoff, S., and Lesterlin, C. (2021). Transient non-specific DNA binding dominates the target search of bacterial DNA-binding proteins. *Molecular Cell* 81, 1499-1514.e6. <https://doi.org/10.1016/j.molcel.2021.01.039>).

Our response: The study mentioned by the Reviewer observed the bound fraction of various DNA-binding proteins in *E. coli* grown in rich defined medium, obtaining values for bound fractions between 55-89%. Their growth condition can be compared only to our fast growth regime condition, in which we observe a bound fraction of 55% for DnaA, at the lower end of the mentioned range. We have incorporated this point in the Results section as follows:

The DNA-bound fraction in fast growth is at the lower end of a previously reported range of 55-89% for other DNA-binding proteins in *E. coli* grown in similar growth conditions^{44,51}.

DnaA is a DNA binding protein; therefore, it is not surprising that a significant amount of it is found on the DNA, this result in itself is not evidence for a titration effect. The main challenge is to determine whether DnaA binding to the genome can have an effect on its activity for the initiation of DNA replication.

Titration on genomic DNA is a general property of DNA binding proteins and can particularly affect the activity of transcription factors. The study of Gao et al shows one way in which one can study the role of titration, by adding what they call “decoy” DNA binding sites (Gao, R., Helfant, L.J., Wu, T., Li, Z., Brokaw, S.E., and Stock, A.M. (2021). A balancing act in transcription regulation by response regulators: titration of transcription factor activity by decoy DNA binding sites. *Nucleic Acids Res* 49, 11537–11549. <https://doi.org/10.1093/nar/gkab935>). In the case of DnaA these could be the DnaA sites found at the *mioC* promoter for example.

DnaA is not very different from other transcription factors, and titration could play a role that is independent from initiation of DNA replication. Indeed, the coupling of DnaA autoregulation of its own expression with a genome titration effect would help maintain the amount of active DnaA proportional to the amount of DNA in the cell. This could be useful in the regulation not only of initiation but also in the expression of DNA repair genes for example (Wurihan, Gezi, Brambilla, E., Wang, S., Sun, H., Fan, L., Shi, Y., Sclavi, B., and Morigen, (2018). DnaA and LexA Proteins Regulate Transcription of the *uvrB* Gene in

Escherichia coli: The Role of DnaA in the Control of the SOS Regulon. *Front Microbiol* 9, 1212. <https://doi.org/10.3389/fmicb.2018.01212>).

Our response: Via our proteomics analysis, we compared protein expression between wild-type *E. coli* and *E. coli* $\Delta 3D$ in all growth condition, as well as between wild-type *E. coli* and the single deletion mutants during slow growth. Our new findings are described in more detail in the new Results section “Changes in DnaA ATP/ADP ratio or titration do not impact its function as transcription factor”. Here is a brief summary concerning the main points of the Reviewer.

Whereas we observed major changes in the global proteomes, the majority of the DnaA regulon did not significantly change in expression due to mutations that alter DnaA activity (Figure 6B-C). The only exception was for FliC and GlpD, both of which are significantly enriched in all growth conditions for *E. coli* $\Delta 3D$, despite differences in DNA content and DnaA bound fraction in the conditions.

To further investigate the role of titration and/or DnaA-ATP/DnaA-ADP ratio, we also analysed changes in proteome in our single deletion mutants during slow growth. We observed that the proteome is not affected by changes in either the DnaA-ATP/DnaA-ADP ratio or its bound fraction. Both $\Delta datA$ and $\Delta DARS1$ mutations led to changes in both the DnaA bound fraction and DnaA-ATP/DnaA-ADP ratio (Supplementary Figure 6B), but not in the global proteome of *E. coli* (Figure 6C). Rather, the deletion of *DARS2* seems to be linked with most of the changes observed in the global proteome found in *E. coli* $\Delta 3D$ (Figure 6C-D). Such an intriguing observation warrants further investigation into the role of this locus on the transcriptional regulation activity of DnaA and of *E. coli* in general that is beyond the scope of this study.

The lack of correlation between the DnaA bound fraction or the DnaA-ATP/DnaA-ADP ratio and the proteome of *E. coli* prompted us to conclude that both the titration and switch mechanisms do not have a major impact on the transcriptional regulation activity of DnaA. On the other hand, the obtained trend of decreased DnaA bound fraction at higher growth rate is consistent with mathematical modelling of a titration mechanisms in *E. coli* cells (Berger & ten Wolde, *Nat. Commun.*, 2022).

The results are presented with little context of previous experimental studies on the role of titration in DnaA activity.

It is known that the activity of the DnaA protein plays a crucial role in the initiation of DNA replication. It has been shown that its nucleotide bound state determines whether it is in an active (ATP bound) or inactive (ADP bound form), and several of the molecular mechanisms that can regulate the ratio of DnaA-ATP to DnaA-ADP have been identified. Previously, different theoretical models have considered an additional regulatory mechanism, the titration of DnaA on the DNA, which can change the amount of DnaA-ATP that is available to bind to the origin, and to the other regulatory sites. Earlier work created an artificial titration situation by adding different amounts of DnaA binding sites. These experiments showed that in response to titration on additional sites, the expression of the DnaA gene increased due to derepression. Thus, compensating for the decreased DnaA activity. (Hansen, F.G., Koefoed, S., Sorensen, L., and Atlung, T. (1987). Titration of DnaA protein by oriC DnaA-boxes increases dnaA gene expression in Escherichia coli. *EMBO J* 6, 255–258.) [This citation and the two below should be

added in line 73]. In fact, they note that the cell's growth rate is quite robust in the presence additional DnaA sites or copies of *oriC* on plasmids and minichromosomes. The derepression of the expression of the DnaA gene suggests that titration using *oriC* DnaA boxes decreases specifically the activity DnaA-ATP over that of DnaA-ADP.

Later, the Skarstad group tried to measure the effect of titration by increasing the copy number of the *datA* site, which we now know that not only binds DnaA but also induces hydrolysis of its ATP (Morigen, Lobner-Olesen, A., and Skarstad, K. (2003). Titration of the Escherichia coli DnaA protein to excess *datA* sites causes destabilization of replication forks, delayed replication initiation and delayed cell division. *Mol.Microbiol.* 50, 349–362; Morigen, Boye, E., Skarstad, K., and Lobner-Olesen, A. (2001). Regulation of chromosomal replication by DnaA protein availability in Escherichia coli: effects of the *datA* region. *Biochim.Biophys.Acta* 1521, 73–80.). These studies showed that increased copies of *datA* could decrease the activity of DnaA, affecting initiation of DNA replication and the replication of minichromosomes when present. They also showed that the autoregulation of DnaA expression could compensate for the titration effect, but only when the *datA* copy number was low. Therefore, it is possible that *datA* can both titrate and inactivate DnaA.

However, the thing that was missing from these studies was a way to control the timing of the increase in copy number of the additional DnaA sites, since they were placed on plasmids and not on the chromosome.

Our response: We added the mentioned references in our extended Introduction. The text now reads:

Additionally, earlier *in vivo* studies introduced titration sites via additional copies of *oriC*³⁹ or *datA*^{40–42} to obtain insights on the effect of titration on DNA replication control. Yet, these studies were performed only under fast growth conditions and before *datA* had been identified as a DnaA-inactivating locus. Further, the additional titration sites were provided on plasmids, inevitably decoupling DnaA titration from DNA replication.

As the authors of this paper point out, the preferential positions of the high affinity sites near the origin means that their copy number increases right after DNA replication has begun. During a window of time of approximately 10 minutes following initiation, the expression of the newly replicated *dnaA* gene is repressed by the SeqA protein, therefore delaying a possible derepression. These sites could therefore contribute to a transient decrease in DnaA activity (together with dilution rate and ATP hydrolysis via RIDA and *datA*) while the origin is also bound by SeqA.

The authors searched the genome for DnaA binding sites that matched the consensus sequence. Line 108-109 what are the other motifs used ?

Our response: We modified the text to include a degenerate sequence of the other motifs searched. The text reads:

We used a list of motifs that include both the consensus sequence of DnaA box (TTWTNCACA)¹⁸, as well as other motifs functional in *datA*, *DARS1* and *DARS2* (HHMTHCWWH)⁴⁵.

Line 110, of the boxes found between centrosomes 98 and 5, how many are part of the origin, the *miuC* promoter and the *DnaA* promoter? It might be interesting to know whether they are near the origin because of the function of the genes (Sobetzko, P., Travers, A., and Muskhelishvili, G. (2012). Gene order and chromosome dynamics coordinate spatiotemporal gene expression during the bacterial growth cycle. *Proceedings of the National Academy of Sciences* 109, E42–E50. <https://doi.org/10.1073/pnas.1108229109>.) or because of the need for titration sites.

Our response: In the indicated region there are a total of 72 boxes, 12 of which comes from either *polA* (4 boxes), *miuC* (3 boxes) or *dnaA* (5 boxes). This number increases from 12 to 20 if counting also includes the boxes present on *oriC*. None of the genes are differently expressed in the different genetic background. We added these considerations on the new section regarding the transcriptional activity of *DnaA* “Changes in *DnaA* ATP/ADP ratio or titration do not impact its function as transcription factor”.

Regarding their position on the chromosome, we believe that the two propositions are not mutually exclusive: the position of the genes in the enriched region can be selected because of both their function and the need of titrating *DnaA*.

DnaA-ATP can bind to the DNA as dimers and oligomers, but not *DnaA*-ADP (Erzberger, J.P., Mott, M.L., and Berger, J.M. (2006). Structural basis for ATP-dependent *DnaA* assembly and replication-origin remodeling. *Nat Struct Mol Biol* 13, 676–683. <https://doi.org/10.1038/nsmb1115>.) For example, Stringer et al (Microbiology 2024) recently identified a number of such closely spaced sites, which they show bind *DnaA* better than individual sites in vivo. Maybe this dimer motif would be a better predictor of the pattern of *DnaA* binding along the genome. Indeed, Stringer et al show that single sites have a very weak signal (with the caveat that in their experiments cells were grown in LB, where the DNA is at a low concentration and *DnaA* has a higher number of possible targets, despite its higher concentration). Does the replication of these sites decrease the amount of *DnaA*-ATP available to inhibit a reinitiation of DNA replication, which would otherwise cause genome instability via fork collapse?

Our response: The binding of *DnaA*-ATP immediately following replication could indeed maintain the free concentration of *DnaA*-ATP low enough to prevent re-initiation events as soon as the eclipse period of *SeqA* is over. We now mention this as a possibility in our revised Discussion. Specifically, the text reads:

The repression of *dnaA* transcription enacted by *SeqA* could act in concert with the synthesis of new *DnaA* boxes via replication to temporarily enhance titration and reduce overall *DnaA* activity. This process would be favoured by the particular chromosomal configuration reported here (Figure 2), leading to a rapid accumulation of new boxes.

If titration via the sites near the origins worked mainly by trapping *DnaA*-ATP after initiation of DNA replication has taken place, then one should see an increased fraction of *DnaA* on the chromosome at a specific cell volume following the initiation of DNA replication.

Our response: The reviewer is correct in mentioning that this is an expected behaviour of titration as a form of control, together with a predicted sharp increase in the free fraction immediately

preceding replication initiation (Berger & ten Wolde, *Nat. Commun.*, 2022). Unfortunately, technical limitations of sptPALM, such as the short fluorophore lifetime (tens of milliseconds) and the limited number of tracks per cell, prevent us to confirm or disprove such behaviours.

Would it be possible to note on the cell area plots in Figure 3D the cell size for initiation of DNA replication? The initiation of DNA replication seems to be correlated better with cell Volume rather than cell Area (see for example: Govers, S.K., Campos, M., Tyagi, B., Laloux, G., and Jacobs-Wagner, C. (2024). Apparent simplicity and emergent robustness in the control of the *Escherichia coli* cell cycle. *Cell Syst* 15, 19-36.e5. <https://doi.org/10.1016/j.cels.2023.12.001>). Would the result change if plotted as a function of cell volume?

Our response: Our analysis framework makes use of Watershed segmentation, a type of image analysis that does not infer shape of the cell but simply segments separate objects in the provided input image and provides an area value for each identified object. Because of this, we did not obtain length and width values needed to calculate cell volume and we thus decided to use the direct measurement we obtained (cell area) for our plots. We argued that this does not considerably impact our analysis, as cell area is a commonly used proxy for cell cycle progression (see, e.g., Knöppel *et al.*, *Proc. Natl. Acad. Sci. U.S.A.* 2023).

However, following the reference provided by the Reviewer, we used aspect ratios (AR) specific to the observed growth rates (Slow: $\mu = 0.2 \text{ h}^{-1}$ and $AR = 3.43$; Intermediate: $\mu = 0.7 \text{ h}^{-1}$ and $AR = 4.14$; Fast: $\mu = 1.4 \text{ h}^{-1}$ and $AR = 4.3$) to convert our area value to volumes, assuming that the shape of *E. coli* cells can be considered as a cylinder with an hemispheres at each pole. Considering the relationship between the length L and width W of $L = AR \cdot W$, we obtain that $r = \sqrt{\frac{A}{\pi + 4 \cdot AR - 4}}$ and

thus $V = (2 \cdot AR - 0.7) \cdot \pi \cdot \sqrt{\frac{A}{\pi + 4 \cdot AR - 4}}^3$. Following this relationship, we turned the area measurements of *E. coli* MG1655 *dnaA-PAmCherry2.1* into volume. Using these values, we obtained the volume of initiation (V_{init}) using the mathematical relationship between this value, the average cellular volume (V) and the average number of origins (n_{ori}) of $V_{init} = \frac{V}{\ln 2 n_{ori}}$ (Si *et al.*, *Curr. Biol.*, 2017). We then converted the value back to cellular area and added a reference to the obtained area of initiation in Figure 3D.

Also, we updated Figure 3D as a function of cell volume as shown below:

Evidently, following the average cellular diffusion coefficient as a function of cell area or cell volume does not change our original conclusions. We thus did not add this extra analysis to the revised manuscript. We note that a similar type of analysis cannot be performed for *E. coli* $\Delta 3D$, as the change in cell size means that the same aspect ratio of the wild-type strain cannot be employed.

Why is the average diffusion coefficient plotted as a function of cell size instead of the kinetic rates or the bound fraction? Maybe there are not enough tracks per cell to measure the bound fraction? Could the cells be binned as a function of their size?

Our response: The Reviewer is correct in noting that we used the average cellular diffusion coefficient instead of kinetic rates in Figures 3D and 4E due to the limited number of tracks. We generally consider 5000 tracks as the minimal number to generate a reliable histogram on which we can apply our analysis framework. For this reason, we would have had to collect cells in subsets with largely different cell sizes to obtain reliable statistics. We argued that by using the average cellular diffusion coefficient, we could probe cell cycle-related differences in DnaA behaviour with better resolution and more reliably.

The data presented in this paper shows that DnaA binds to the DNA, and that the mutations of the genomic sites affecting its activity also change the fraction that is bound, however it does not support a specific role of titration in DNA replication regulation.

In the specific case of DnaA, titration could play a double role: (i) in creating a step-like function increase in the activity of DnaA and therefore result in improved synchrony of initiation at the different origins and (ii) in decreasing the activity of DnaA after initiation has taken place by the increase of DnaA sites resulting from the replication process. However, data on the role of titration has been elusive. The difficulty in

measuring a specific effect of titration on initiation of DNA replication is due to (i) the difficulty in measuring the activity of DnaA *in vivo*, due to the presence of both the ATP and the ADP bound forms with different DNA binding properties (ii) the redundancy of multiple regulatory mechanisms required to keep the initiation process coupled to cell growth under all growth conditions, some being more important under slow growth for example or in response to stress, (iii) DnaA's binding sites on the genome also have other specific functions, apart from the origin, acting to regulate gene expression or the nature of the nucleotide bound to DnaA itself.

The current title just describes what is known for most DNA binding proteins, they will spend a significant time bound more or less specifically to the DNA. In itself, this result is not very surprising or useful to understand how DnaA activity is regulated. Therefore, more experiments are needed in order to be able to say that titration of DnaA plays a specific role in the regulation of DNA replication.

Our response: We agree with the Reviewer that our study does not conclusively attribute a specific role of DnaA titration in setting the timing of DNA replication initiation, due to the observed dual role of *datA*, *DARS1* and/or *DARS2* which we also point out in our Discussion. However, we believe our data still provides novel, direct experimental evidence for the capacity of *E. coli* chromosome to bind and titrate the initiator protein. This biological insight had so far only been hypothesised to take place and was only supported by mathematical models and indirect experimental evidence. Specifically, the observed decrease in the bound fraction with higher growth rate is consistent with previous mathematical modelling, which predicts that this relationship results from the titration mechanism. Moreover, our work shows how the bound fraction of DnaA varies based on the DnaA-ATP/DnaA-ADP ratio, a feature that could help stabilise cell cycles in conditions in which the switch mechanism is perturbed. We elaborated on this scenario in the revised Discussion.

We also believe that DnaA cannot just be assimilated to any other DNA binding protein: its specific form of control (near-balanced expression) is not a trait confirmed for all DNA binding protein, as well as the non-monotonic dependence of its concentration on growth rate. Moreover, although the DNA-binding behaviour of many proteins has been characterised as the Reviewer already mentioned (for instance: Stracy *et al.*, *Mol. Cell.* 2020), so far there was no knowledge on the behaviour of DnaA and, more generally, limited information is available on how the DNA-bound fraction of proteins changes during different growth regimes. In the future, follow-up improvement and expansion of our experimental setup will address the gap of knowledge still present and highlighted by Reviewer 1.

Additional concerns:

Lines 37-39. In the introduction, and elsewhere, the authors put on the same plane conclusions obtained via experimental methods and theoretical models, which however still need to be experimentally confirmed. In addition, they cite results obtained as a function of growth conditions, which however have no data about the cell cycle.

Our response: We agree that the phrasing of this sentence can be misleading when trying to distinguish between behaviour related to growth rate or cell cycle. For this reason, the text now reads:

At the expression level, the *dnaA* gene is negatively autoregulated¹⁶ and its near-balanced expression has been reported to lead to stable intracellular concentrations of DnaA across the cell cycle¹⁷. At the same time, the population average concentration of DnaA increases at faster growth rates¹⁰.

The same can be said on line 146, the cited paper is a mathematical model, the text should reflect this. For example: “a feature that could help initiator titration during fast growth”.

Our response: We implemented the suggestion by modifying the sentence in question as:

As a result, both the total number of DnaA boxes and their concentration can quickly increase after initiation, a feature that could help initiator titration during fast growth³⁶.

There is nothing wrong with theoretical models, they are very useful to interpret data and guide future experiments, and however, one cannot say that DnaA expression is balanced with cell growth because a model says so. In general, there is no point of invoking two different models, the switch and titration models, as if there were two exclusive realities. It seems an outdated description of what we know about the regulation of DnaA activity. For example, the *datA* site can probably do both. An alternative would be to say that it is known that regulation of the DnaA-ATP to DnaA-ADP ratio is important for regulation of initiation, it remains to be determined whether titration can also play a role, and in which growth conditions.

Line 60: both models can be based on the accumulation of DnaA-ATP determining the moment of initiation of DNA replication. The difference between the models is that the accumulation of DnaA-ATP is delayed by the binding of the genomic sites, until it passes a threshold level, and on the mechanisms that can inhibit re-initiation after duplication of the origin region. In both cases, there is a peak in the DnaA active form. The amount of active form is the amount of DnaA-ATP that is available to bind to the origin, or other sites on the genome, as measured by Iuliani et al. (2024).

Our response: We agree with the Reviewer that the two proposed mechanisms of DnaA control are not mutually exclusive, yet are likely to act in concert to establish constant cell cycles in *E. coli*. For this reason, we modified the text from line 64, which now reads:

The viability of the quadruple mutant was attributed to the intrinsic ATPase ability of DnaA, while postulating that initiator titration could provide an additional stabilising effect¹⁷. Moreover, recent single-cell analysis of the oscillation of DnaA-ATP levels in *E. coli* showed that reaching a peak level of DnaA-ATP is a necessary but not sufficient condition to initiate DNA replication³³. In light of these results, either the accumulation of DnaA-ATP or its binding to the low-affinity binding sites on *oriC* could be delayed by the titration of the initiator on the *E. coli* chromosome. In agreement with this, previous *in silico* work suggested that concerted DnaA interconversion and its titration on the chromosome can enhance the stability of *E. coli* cell cycles in all growth conditions^{34,35}, with a larger effect at slower growth rates³⁶.

Line 188-189, incomplete sentence. It is understood from the context, but it might be better to complete it by specifying which mechanism is being referred to.

Our response: We modified the text, now reading:

The capacity of *E. coli* chromosome to sequester DnaA hints at initiator titration being a possible mechanism for controlling the activity of DnaA.

Line 191-192 does this refer to fully consensus sites? Dimer sites?

Our response: The text originally referred to the number of DnaA boxes with consensus motifs. We modified the text to account for both types of boxes, it now reads:

The genome of this bacterium has hundreds of DnaA boxes that can recruit DnaA, peculiarly arranged to have a higher density towards the origin (Figure 2). Whereas the potential function of most of these boxes is still unknown, six of them are at the *oriC*, while 26 more are located among the switch control loci *DARS1*, *DARS2* and *datA*⁴⁵.

Line 200, maybe it would be more clear to say something like: cells initiate a new round of replication upon reaching a well-defined size that is higher in the mutants than in the wild type. Consistent with a decrease in DnaA activity. (which is the opposite of what one would expect if there were less titration of DnaA but especially less conversion of DnaA-ADP to DnaA-ATP)

Our response: Following our experimental characterisation of the DnaA-ATP/DnaA-ADP ratio using the activity of the *dnaAp* promoter as a proxy, we modified the text as:

The anti-correlation between the change in average cell size and the change in average number of origins upon deletion of loci mediating DnaA interconversion is consistent with previous reports^{17,45} and with the idea that cells initiate a new round of replication after reaching a well-defined size^{6,7,54}. This size is then larger in *E. coli* Δ 3D than in *E. coli* MG1655 (Figure 4B) due to the slight decrease in DnaA-ATP levels (Supplementary Figure 5B) caused by the triple mutation.

Lines 202-203, the phenotype of larger cell size could also be due to a longer C period and a higher number of origins is consistent with rifampicin resistant initiations in the flow cytometry experiments due to increased amounts of DnaA-ATP and delayed division (see the following and references therein: Flåtten, I., Fossum-Raunehaug, S., Taipale, R., Martinsen, S., and Skarstad, K. (2015). The DnaA Protein Is Not the Limiting Factor for Initiation of Replication in Escherichia coli. PLOS Genetics 11, e1005276. <https://doi.org/10.1371/journal.pgen.1005276>). Rifampicin resistant reinitiations in cells whose cell division has been inhibited by cephalixin can result from a protection of the DnaA promoter from rifampicin by DnaA itself (see Skarstad).

Our response: In their studies, the Skarstad group observed rifampicin-resistant phenotypes in the presence of large excess of active DnaA. Our proteomics analysis showed that the expression levels of DnaA did not change considerably between the wild-type and the triple deletion mutant (Supplementary Figure 5E). At the same time, the activity of the *dnaAp* promoter in *E. coli* Δ 3D during slow growth was slightly higher than the wild-type (Supplementary Figure 5B), pointing at

a lower DnaA-ATP/DnaA-ADP ratio compared to the wild-type. These observations are in contrast with the conditions that generated the rifampicin resistant phenotype in the mentioned work from the Skarstad group.

Regarding the possibility of an increase in the C period, previous reports showed this parameter tends to be quite constant regardless on many perturbations, including repression of DnaA expression (Si *et al.*, *Curr. Biol.*, 2017). We thus believe that it is unlikely that such a change could play a role in the increase cell size of *E. coli* $\Delta 3D$. Rather, we explain this phenotype as a delayed initiation of DNA replication, consistent with the higher activity from the *dnaAp* promoter and thus lower DnaA-ATP abundance. The text reads:

The increase in both average cell size and DNA content can be reconciled by noting that the activity of the *dnaAp* promoter increased during slow growth (Supplementary Figure 5B). This increase points at a lower abundance of DnaA-ATP in *E. coli* $\Delta 3D$ compared to the wild-type, leading to delayed initiations and thus higher values of cell area. Still, the higher DNA content is in apparent contradiction with the lower abundance of DnaA-ATP and could be explained by the presence of frequent re-initiation events.

Lines 235-236, these results would also be consistent with a scenario where there is more DnaA-ATP in the cell (because of less RIDA at slow growth and no *dataA* (also see Skartad's paper above)) and all the sites are saturated.

Our response: As previously mentioned, our proteomics analysis points to near-wild-type levels of DnaA expression, while our LacZ assay showed mildly higher activity of the *dnaAp* promoter compared to wild-type, pointing at an average lower DnaA-ATP/DnaA-ADP ratio, excluding this possibility. These observations are also consistent with the larger cell size, pointing at delayed initiation. However, after enough DnaA-ATP is accumulated to initiate replication, the cell can only inactivate DnaA through RIDA and, minorly, the intrinsic ATPase activity of DnaA (Boesen *et al.*, *Proc. Natl. Acad. Sci. U.S.A.* 2024). We added these points in our revised Discussion:

At this point, *E. coli* is mostly relying on the action of RIDA to inactivate DnaA, together with its intrinsic ATPase activity¹⁷. In this scenario, most DnaA is still freely diffusing rather than bound on the chromosome (Figure 4D) and modest accumulation of more DnaA-ATP could then stimulate cells to initiate new rounds of replications after the release of SeqA from *dnaAp* and *oriC*¹⁷ (Figure 7, bottom). Such frequent re-initiations would then explain the observed increase in DNA content in *E. coli* $\Delta 3D$ during slow growth.

Line 296, Shank *et al.*'s study is on *B. subtilis* DnaA, long distant cousin with very different mechanisms of regulation. This is not a valid comparison, or at least it should be specified.

Our response: The Reviewer is correct in mentioning that the study from Schenk *et al.* focusses mostly on *B. subtilis*. However, the authors also used a previously generated *E. coli* strain carrying a *dnaA-eyfp* fusion (Nozaki *et al.*, *J. Bacteriol.*, 2009) and obtained some insights on its mobility (e.g., Table 3 or Figure 7). We believe it was important to refer to their study and its limitation as the first single-molecule tracking effort of DnaA in live *E. coli* cells.

Lines 300-301, unfortunately the results of this study do not allow for an improved understanding of the impact of titration on DNA replication compared to previous work.

Our response: See our response to the last main comment of the Reviewer.

Lines 306-307: (i) balanced biosynthesis has not been shown directly; furthermore, the concentration of DnaA has been shown to change as a function of growth rate (Zheng et al 2020) (ii) the speed of DNA replication is not constant as a function of growth rate (Michelsen, O., Teixeira de Mattos, M.J., Jensen, P.R., and Hansen, F.G. (2003). Precise determinations of C and D periods by flow cytometry in *Escherichia coli* K-12 and B/r. *Microbiology (Reading)* 149, 1001–1010. <https://doi.org/10.1099/mic.0.26058-0>.) see also Skarstad's work on the *datA* mutation.

Our response: As mentioned in a previous comment, nearly balanced synthesis was recently observed for DnaA in live cells thanks to a fusion to GFP (Boesen *et al.*, *Proc. Natl. Acad. Sci. U.S.A.* 2024), albeit in a single growth condition. We added this reference in the text to support the statement.

On the other hand, the duration of DNA replication has been shown to be approximately constant at 38.0 ± 4.5 min in a wide range of growth conditions (Si *et al.*, *Curr. Biol.*, 2017). We also now refer to that study to support our statement.

We modified the text to read as:

This phenomenon is the result of two previously described *E. coli* characteristics: (i) near-balanced biosynthesis^{17,71}, leading to constant intracellular DnaA concentration during the cell cycle, and (ii) the approximately constant speed of DNA replication elongation⁷.

Line 311, the rate at which new DnaA boxes are replicated per origin is different from the speed at which new boxes are replicated -per cell- or -per volume- due to the increase in the origin to terminus ratio with growth rate.

Our response: We agree with the Reviewer, yet we are unsure as of how they would like us to acknowledge this fact in the text.

Line 320-321, the presence of the high affinity boxes towards *oriC* just means that the genes regulated by DnaA need to be near the origin in order to keep their copy number increasing with growth rate. This fact alone does not support the existence of a titration-based form of control.

There is no evidence that these boxes are responsible for the binding of over half of DnaA proteins in the cell.

Our response: Regarding the position of the boxes on the chromosome, see our new section "Changes in activity or titration of DnaA do not impact its function as transcription factor". Here, we analysed the positions of the genes regulated by DnaA and found that only *dnaA* itself, *mioC* and *polA* are part of the enriched region, for a total of only 12 of the 72 DnaA boxes present there. Thus, we exclude that the skew we observed can be attributed *exclusively* to the regulation of the transcriptional repressor function of DnaA.

We agree that we do not have direct evidence of the use of specific boxes by DnaA. However, due to the DNA-binding nature of DnaA, we believe that our bound fraction can be mostly attributed to its interaction with the *E. coli* chromosome. Therefore, it is reasonable to state that the lowest observed bound fraction of >50% comes from the boxes present on the chromosome, especially considering that the deletion of *datA*, *DARS1* and *DARS2* did not impact this value at faster growth rates. We modified the section of text to acknowledge that this statement refers to our interpretation of our single-molecule data, now reading:

Still, the bound fraction accounted for more than 50% of visible DnaA proteins in all experimental conditions (Figure 3C, 4D, 5B). We hypothesise that the binding of DnaA to boxes scattered throughout the chromosome is responsible for this observation, with pairs of closely spaced boxes⁷² and specific loci (e.g. *mioC*⁷³ and *oriC*^{74,75}) mediating more stable forms of binding. The preferential accumulation of boxes towards *oriC* (Figure 2B) then means that new DnaA boxes can rapidly be synthesised after DNA replication initiation, enhancing the titration power of the chromosome at all growth rates. This genomic configuration and the trends of association rates and DnaA bound fractions in different growth regimes (Figure 3C) are in accordance with previous modelling efforts^{34,36} predicting the existence of a titration-based control mechanism.

Lines 338-339, see the comments above about lines 202-203.

Our response: See our response to the comment regarding lines 202-203.

Lines 345-347, the regulatory mechanisms at fast and slow growth are not the same, therefore one should not expect the same phenotype.

Our response: The Discussion has been reworked to include our latest experimental addition (proteomics and LacZ assay). Moreover, we acknowledge that RIDA and *DARS2* play a minor role during slow growth in the Results section. Referring to the changes observed in *E. coli* Δ 3D during slow growth, the text reads:

The *dnaAp* promoter was also more active in *E. coli* Δ 3D than in the wild-type (Supplementary Figure 5B), suggesting lower levels of DnaA-ATP as observed during intermediate and fast growth. The change in activity was milder compared to the other growth condition, likely due to a smaller contribution of RIDA and *DARS2* during slow growth.

While the cells were grown at 37 °C, the measurements were carried out at room temperature. The binding of DnaA-ATP to DNA is temperature dependent, binding with a lower affinity at lower temperatures (Saggiaro, C., Olliver, A., and Sclavi, B. (2013). Temperature-dependence of the DnaA-DNA interaction and its effect on the autoregulation of *dnaA* expression. *Biochem. J.* 449, 333–341. <https://doi.org/10.1042/BJ20120876>.) This suggests that these measurements could overestimate the amount of free DnaA in the cell.

Our response: We acknowledge this temperature-dependent effect in the Results section, reading:

The binding of DnaA to DNA has been reported to be temperature-dependent *in vitro*, with weaker binding occurring at lower temperature⁵³. Therefore, our single-particle experiments performed at room temperature could have led to an underestimation of the bound fractions. However, this temperature-dependent effect was constant throughout all our experiments and does not invalidate the observed growth rate-dependent behaviour.

The ideal, but maybe impossible, experiment would be to measure the diffusion constant of DnaA of cells growing in their growth medium, instead of PBS, at a temperature between 30 and 37°C, as a function of time and thus follow the change in diffusion constant in the same cell as the cell size increases. Even more ideally, one would need a fluorophore specific to the nucleotide bound state of DnaA, something that fluorescence only when DnaA dimerizes or oligomerizes (FRET?).

Our response: We agree with the Reviewer that a microfluidic cultivation-based pipeline would be ideal to investigate these interactions. We are planning to apply these for future experiments, as now mentioned in the discussion as follows:

Finally, implementing microfluidics-based cultivation strategies⁸⁰ could then allow to probe variations in DnaA abundance¹⁷ and the DnaA-ATP/DnaA-ADP ratio³³, as well as to connect changes in DnaA mobility with key moments of the *E. coli* cell cycle.

Regarding FRET, dimerization assays are very difficult to set up for probing homodimerization and are even more challenging in the context of utilising single-molecule readouts. Of note, an approach on studying intermolecular interactions was recently reported (Yan *et al.*, PNAS 2025).

Several citations in the methods are wrong. Something must have happened when the methods were pasted into the main document.

Our response: We thank the reviewer for spotting this. We carefully checked the entire manuscript again to ensure the correct order and formatting of the references.

Reviewer #1 (Remarks on code availability):

My level of expertise does not allow me to properly review code.

Reviewer #2

This paper provides experimental evidence supporting the titration of the replication initiator protein DnaA in *Escherichia coli* and explores how elements regulating the DnaA-ATP/DnaA-ADP switch (dataA, DARS1, and DARS2) affect DnaA titration and initiation. The authors address three main questions:

1. Distribution of DnaA boxes on the chromosome: Using a comparative genomic approach, the researchers analyzed the *E. coli* MG1655 genome alongside other *E. coli* strains, a related species

(*Salmonella enterica*), and random permutations. They identified a conserved enrichment of DnaA boxes near the origin of replication (*oriC*), suggesting a universal bias.

2. Fraction of DnaA bound to the chromosome: Employing single-particle tracking photoactivatable localization microscopy (sptPALM), the authors measured the mobility of individual DnaA proteins. They found that in wild-type *E. coli*, 55-75% of DnaA proteins are chromosome-bound, with the bound fraction decreasing as growth rate increases.

3. Effects of deleting switching elements (*datA*, *DARS1*, *DARS2*): In a mutant lacking all three elements ($\Delta 3D$), the bound fraction of DnaA remained at ~60% across growth regimes. The researchers also noted that the combined effects of these deletions on DnaA binding were not simply additive.

This study fills a critical gap by providing direct quantitative evidence of DnaA titration in live cells and highlights its importance as a major mechanism for replication initiation control. The experimental results are robust, though the interpretations of growth-regime-specific mechanisms are somewhat intricate in the Discussion section. A straightforward takeaway is that initiator titration is significant in all conditions and mutants tested (with over 50% of DnaA proteins bound), but its strength varies moderately with growth rate and the presence of switching elements. Overall, we recommend publication in *Nature Communications* once the authors have addressed the following minor concerns.

Our response: We would like to thank the Reviewer for their efforts in assessing our work, and for the supportive words and suggestions.

Minor concerns:

1. Figure 2A lacks clarity regarding the meaning of the radius in the circular plot. From the context, it seems to be something like the number of DnaA boxes per unit length of the chromosome or the DnaA box density, but this is not explicitly defined. Please define the radius of the circle in Figure 2A explicitly in the main text and in the figure legend to avoid misinterpretation.

Our response: We adapted the main text as suggested. Regarding figure 2A, the text now reads: We thus compared the number of DnaA boxes per position (radius of circular plot) in windows of 10 centisomes on the *E. coli* chromosome with the number expected when sampling from a uniform distribution (Figure 2A).

Regarding Figure 2B, the text now reads:

The accumulation of DnaA boxes towards *oriC* was not a characteristic unique to *E. coli* MG1655. We plotted the density of DnaA boxes across the chromosomes of 56 *E. coli* strains and obtained a Z-score (indicated from the radius of the circular plot) (Figure 2B).

We also now mention similar things in the caption of Figure 2. Caption for panel A now reads:

The radius of the circle indicates the number of boxes per centisomes, counted within a 10 centisomes window.

Caption for panel B now reads:

The radius of the circle indicates the number of standard deviations away from the mean (Z-score) per centisome, counted within a 10 centisomes window.

2. Figure 2B: are these all consensus DnaA boxes (i.e., TTWTNCACA)? Additionally, providing the total number of DnaA boxes in each strain would add valuable context.

Our response: Due to the large number of strains analysed, we added a new sheet to Supplementary Table 1, containing the number of boxes counted in each strain. We then clarified the type of motifs used in the main text and added the average number of DnaA boxes present in the genome of *E. coli*. The main text now reads:

As for our previous analysis, we included both consensus motifs and other types of boxes found in *datA*, *DARS1* and *DARS2*, for an average of 902 ± 77 boxes per *E. coli* genome (see Supplementary Table 1 for strain-specific numbers).

A general statement was also added in the caption of Figure 2, saying:

In all panels, we used a TTWTNCACA motif for the consensus DnaA box and a HHMTHCWHVH motif to include the boxes present among *datA*, *DARS1* and *DARS2* in the analysis.

3. Line 136-137: "...rejecting previous assumptions on their homogenous distribution.12,19" This is inaccurate and unnecessarily strong. First, neither Ref. 12 nor Ref. 19 assumed a homogeneous distribution in their study. Rather, both Ref. 12 and Ref. 19 considered the biased distribution of DnaA boxes near ori. Ref. 19 only stated that high-affinity DnaA boxes are randomly distributed, but not for all consensus ones. Second, while heterogeneity exists, the distribution in Figure 2A does not seem to significantly deviate from a homogeneous distribution. Based on this result, it still seems valid to assume a homogenous distribution as a zeroth-order approximation in some physical models such as in Ref. 34 and 35. The emphasis on heterogeneity seems overstated, raising the question of whether incorporating it is crucial for future modeling efforts. A similar overstatement appears in Line 110 and should be addressed for consistency.

Our response: While it is true that a homogeneous distribution can serve as a useful zeroth-order approximation in physical models, our analysis in Figures 2A and 2B demonstrates statistically significant deviations from uniformity. Specifically, it shows an overrepresentation of DnaA boxes near the origin of replication, supported by multiple complementary analyses. First, we simulated box positions from a uniform distribution and found that the observed box counts near *oriC* were significantly higher than expected by chance (Figure 2A, yellow area). Second, we generated 100 shuffled versions of the DnaA box motif and mapped them to the genome to account for eventual biases stemming from nucleotide composition of the *E. coli* genome. Again, the real box distribution showed a distinct enrichment near *oriC* (Figure 2A, orange area). Finally, we calculated Z-scores for DnaA box density across centisome bins in 56 *E. coli* strains and one *Salmonella enterica* genome, revealing a conserved local maximum at *oriC* (Figure 2B, yellow area).

While we agree that a homogeneous assumption remains useful in simplified modelling frameworks, our findings suggest that incorporating this biologically grounded heterogeneity may improve the fidelity of models concerned with DnaA availability and titration dynamics. Nevertheless, we have revised the text as suggested, now reading:

We identified an extended region of significant overrepresentation of DnaA boxes between centisomes 98 and 5, specifically revealing a preferred enrichment of DnaA boxes around *oriC*.

4. Line 213-214: "...while the mobility of DnaA increased throughout the whole cell cycle (Figure 4E)." This statement seems inconsistent with the data. Indeed, in Figure 4E top panel (slow growth), the diffusivity only increases when the cell size is small, and becomes flat for most of the cell cycle. We suspect that the short increasing phase after cell birth is right before initiations when cells need to accumulate a large amount of free DnaA to have enough DnaA-ATP because DnaA-ATP fraction is low due to the absence of *DARS1* and *DARS2* in this mutant.

Our response: We corrected our statement to better represent the data as suggested by the Reviewer. The text now reads:

The average mobility of DnaA increased during the first part of the cell cycle, before stabilising at around $1 \mu\text{m}^2/\text{s}$ (Figure 4E).

Regarding the accumulation of DnaA in general and DnaA-ATP, the total expression of DnaA is unchanged compared to the wild-type (see new data in Supplementary Figure 5C). We believe that the synthesis of new DnaA is necessary for the accumulation of DnaA-ATP due to the deletion of reactivating locus *DARS1* (*DARS2* is not active during slow growth). At the same time, the decrease in titration power from the chromosome leads to the increase in mobility of DnaA, in agreement with the Reviewer suggestion. The stabilisation of DnaA diffusivity could then be due to the initiation of new rounds of replication leading to the accumulation of new boxes. Accordingly, the revised Discussion now reads:

In this mutant, the deletions of *datA* and *DARS1* leave the cell in a deficiency of DnaA-ATP, as indicated by the increased activity of the *dnaAp* promoter (Supplementary Figure 6B). This effect also leads to a matching decrease in titration (see Δ *DARS1* mutant), which is further impacted by the loss of *DARS1*, *DARS2* and especially *datA*^{40,79} as titration loci. Due to the lack of reactivating loci, the only mechanism left for the cell to accumulate sufficient DnaA-ATP is through synthesis of new DnaA²⁵. The accumulation of new DnaA-ATP then leads to the increase in DnaA mobility that we observed at the beginning of *E. coli* Δ 3D cell cycle (Figure 4D). After sufficient DnaA-ATP is accumulated and replication is initiated, new DnaA boxes are rapidly synthesised, thereby stabilising DnaA mobility (Figure 4C) and lowering its activity.

5. Line 369 and 383, also Figure 6: The concept of controlling the first replication event is underexplained and we do not understand the meaning of distinguishing it from other replication events.

Our response: The Discussion has been extensively modified to include our latest experimental addition. As a result, the lines referred to in this comments are no longer present in the current text. On the topic, the revised Discussion now reads:

At this point, *E. coli* is mostly relying on the action of RIDA to inactivate DnaA, together with its intrinsic ATPase activity¹⁷. In this scenario, most DnaA is still freely diffusing rather than bound on the chromosome (Figure 4D) and modest accumulation of more DnaA-ATP could then stimulate cells to initiate new rounds of replications after the release of SeqA from *dnaAp* and *oriC*¹⁷ (Figure 7, bottom). Such frequent re-initiations would then explain the observed increase in DNA content in *E. coli* Δ 3D during slow growth.

6. Including supplementary videos showing examples of DnaA movement with different diffusivity would greatly aid in visualizing the findings and improving reader comprehension.

Our response: We included Supplementary Video 1, a representative example in which non-filtered tracks within the first 3000 frames of one of our experimental conditions are overlaid on a brightfield image showing the immobilised bacteria.

Reviewer #3

This study used newly developed single cell methodology and fluorescence-labeled DnaA to quantitatively analyze intracellular movements of DnaA in growing cells with various doubling times. Also, the authors mathematically deduced DnaA box density for the whole genome of *E. coli* species. Results of these two are not necessarily linked together in this study, but are proposed to be consistent with the idea that the initiator titration model is partly valid for regulation of replication initiation in slowly growing cells. Challenging experiments were well performed and interpretation to the results are overall reasonable. However, I have some questions as follows and am not very convincing to reliability of the values of DnaA bound fraction and a part of the conclusion.

Our response: We would like to thank the Reviewer for their efforts in assessing our work, and for the supportive words and suggestions.

1. Figure 2 and Supplementary Figure 2AC: Include explanation to the numbers of DnaA boxes (20, 40, 60 and 100) indicated by circles in Results or Figure legend. Methods describes 'We calculated the number of boxes per position in a sliding window approach by moving over the scaled chromosome in a 10 centisome window, taking 1 centisome steps.' However, a brief explanation is also required in Results or Figure legend.

Our response: We now describe the meaning of the radius of the circular plots both in the main text (Results) and in the caption of Figure 2. Additional information can be found in the response to the first comment of Reviewer 2.

2. Relating above: I feel that the 10 centisome window is too long. Show several data using several different lengths of the window. Comparing those data are required to make a conclusion for DnaA box density bias

Our response: We performed the same analysis with windows of either 2 or 6 centisomes. We observe that the enriched region around *oriC* persists, although clearly with different sizes due to the employed windows. We decided to leave the plot obtained with a 10 centisomes window in the main text and to add the new plots as Supplementary Figure 2B. The text now refers to it as: The same conclusions were reached when using only the consensus sequence (Supplementary Figure 2A) or when counting DnaA boxes in windows of either 2 or 6 centisomes (Supplementary Figure 2B).

3. Relating above: The *oriC* region should have exceptionally high DnaA box density. This should affect the density levels of the *oriC*-proximal regions by the window. This is not preferable for the intended analysis of the initiator titration model. Thus, these calculations should be done excluding the *oriC* region.

Our response: We performed the same analysis excluding *oriC* from the analysis and found that the adjacent region retained its enrichment for DnaA boxes. We added the related plot as Supplementary Figure 2C. The main text refers to it as: Additionally, we tried excluding *oriC* to test whether its high density of DnaA boxes could bias our analysis. Even without considering the boxes present at the origin, the enrichment in the adjacent region persisted (Supplementary Figure 2C).

4. Figure 3A: The codon no. and domain structure should be indicated to DnaA.

Our response: We modified the figure as suggested to include the codon number in panel A, as well as indicating the domains. We also modified the caption accordingly, now reading: The domains of DnaA are indicated with Roman numerals, excluding the two flexible linkers domain II and domain IIIb, depicted in light grey. On the bottom, the codon numbers of each domain of the wild-type DnaA are indicated.

5. Figure 3A: The cellular number of DnaA-PAmCherry2.1 molecules in MG1655 *dnaA*-PAmCherry2.1 cells growing in the indicated three conditions should be shown by immunoblotting or similar experiments. Also, are DnaA-PAmCherry2.1 molecules stable in cells? Degradation products, if present at a certain level, would adversely affect the microscopic data.

Our response: Due to the lack of an available antibody against DnaA, we performed proteomics to assess the levels of DnaA in our different genetic backgrounds and observe eventual changes. We observed no difference in DnaA expression level between *E. coli* MG1655 and *E. coli* MG1655 *dnaA*-PAmCherry2.1, see Supplementary Figure 4C. Taken together with the unchanged DNA content, cell size and activity from the *dnaAp* promoter (rest of Supplementary Figure 4), we conclude that the fusion construct is stable in the cells.

6. Figure 3C and Results lines 156-164: Understanding how the bound fractions (%) are determined is very difficult. At least brief explanation is required in this part, although Methods includes detailed explanation (lines 587-595).

Our response: We now include an explanation of how the bound fraction is calculated in the Results section as well, reading:

We then used the kinetic rates to estimate the average DnaA bound fraction as

$$\frac{k_{\text{Free} \rightarrow \text{Bound}}}{k_{\text{Free} \rightarrow \text{Bound}} + k_{\text{Bound} \rightarrow \text{Free}}}$$

7. Relating above: The authors assume certain values of diffusion constants/coefficients for distinguishing the bound and free DnaA molecules (2.7 $\mu\text{m}^2/\text{s}$ and 0.12 $\mu\text{m}^2/\text{s}$), based on theoretical assumptions. This is OK, but other different values for these parameters should be used for calculations for comparison.

Our response: A previous study from one of the corresponding authors (Martens... Hohlbein, Nat. Commun., 2019) showed that, although the mobility of a plasmid can be detected, the mobility of the chromosome is below the localisation precision of our set up ($\sigma = 30 \text{ nm}$) due to its size. Therefore, the chromosome and any protein bound to it appear as immobile. DnaA proteins bound to the chromosome then move with a step size of our localisation precision, yielding a value of diffusion coefficient of 0.12 $\mu\text{m}^2/\text{s}$. We modified the text to better explain this constraint:

In our set-up, proteins bound to DNA appear as immobile^{44,99}; therefore, DnaA was considered as immobile in its DNA-bound state, with a diffusion coefficient solely determined by the localisation uncertainty ($D_{\text{Bound}}^* = D_{\text{Immobile}}^* = 0.12 \mu\text{m}^2/\text{s}$).

Regarding the diffusivity of free DnaA, see our answer to the next comment.

8. Relating above: The diffusion constants of free DnaA should be experimentally deduced using DnaA mutant defective in DnaA box binding. DnaA-PAMCherry2.1 variant with such a mutation should be expressed in MG1655 cells and be analyzed.

To confirm the diffusion coefficient for free DnaA, we repurposed an expression plasmid of a previous single-particle tracking study (Olivi *et al.*, *Nucleic Acids Research*, 2024) to express DnaA211 (Hansen *et al.*, *Mol. Gen. Genet.*, 1992), a mutant of DnaA known to be impaired in DNA binding (Roth & Messer, *EMBO J.*, 1995) fused to PAMCherry2.1. We then performed single-particle tracking of the resulting strain to observe the behaviour of DnaA proteins not bound to DNA. In this way, we obtained a range of estimated diffusion coefficients of $2.6 \pm 0.2 \mu\text{m}^2/\text{s}$; our previously used value of diffusion coefficient (2.7 $\mu\text{m}^2/\text{s}$) falls inside this range. The diffusion coefficient histograms of this set of experiments is available as Supplementary Figure 4F. Moreover, we fitted the diffusion distributions of DnaA in *E. coli* MG1655 in all growth rates using the previously obtained kinetic rates and the extremes of the diffusion coefficient ranges obtained with DnaA211 (red line: 2.4 $\mu\text{m}^2/\text{s}$, black line; 2.8 $\mu\text{m}^2/\text{s}$).

Theoretical distributions obtained with values falling within the range (either 2.4 or $2.8 \mu\text{m}^2/\text{s}$ in this figure and $2.7 \mu\text{m}^2/\text{s}$ in the manuscript) well describe our experimental data. Thus, we maintained the original value of $D_{\text{Free}}^* = 2.7 \mu\text{m}^2/\text{s}$ in all instances in the manuscript.

Regarding the additional control using DnaA211, the revised Methods section now reads:

We further confirmed this value by using the pDnaA211-PAmCherry2.1 plasmid to express DnaA211⁹⁶, a DnaA mutant impaired in DNA binding and thus solely moving in a free state. We then analysed the diffusional distributions of the protein as described above, simulating a static species ($k_{\text{Free} \rightarrow \text{Bound}} = k_{\text{Bound} \rightarrow \text{Free}} = 0 \text{ s}^{-1}$) and estimating diffusion coefficients that would best fit the experimental data. In this way, we obtained a diffusion coefficient of $2.6 \pm 0.2 \mu\text{m}^2/\text{s}$ across all growth regimes (Supplementary Figure 4F).

9. The chapter of lines 187-241 and Figure 4 and 5: The duration of chromosomal replication is thought to be relatively constant in cells with different growth rate at the same temperature. So, there is another hypothesis; i.e., In delta-3D cells growing slowly, the ATP-DnaA level is inferred to be elevated because

RIDA works only for a limited period in the long doubling time. In contrast, in delta-3D cells growing at intermediate and fast rates with the shorter doubling time, RIDA works relatively more efficiently, reducing the ATP-DnaA level and repressing overinitiation. This hypothesis can explain the cause of the overinitiation in delta-3D cells growing slowly. The increased *oriC* copies move for partition in cells, which enhances movement of DnaA bound to the chromosome. In panel D, the indicated 'bound' DnaA does not necessarily mean DNA-bound molecules but indeed does slowly moving molecules. Such chromosome dynamics for partition of the *oriC* copies and then chromosomes can explain partial enhancement of movements of DnaA molecules even bound to the chromosome, which can be recognized as free DnaA molecules in the diffusion constants/coefficients. Note that this hypothesis is valid also to explain the results of Figure 5. Thus, these data do not necessarily mean the DnaA titration is reduced in the delta-3D cells growing slowly. Thus, the chapter should be revised fundamentally.

Our response: We agree that the form of nucleotide bound to DnaA seems to have an impact on the titration of DnaA, with an increase of DnaA-ATP leading to a higher binding to the chromosome (see our previous answers to Reviewer 1). We assessed the DnaA-ATP/DnaA-ADP ratio of the strains in this study using the expression of the LacZ β -galactosidase from the *dnaAp* promoter as a proxy for this ratio (Charbon *et al*, *Mol. Microbiol.* 2011; Boesen *et al.*, *Proc. Natl. Acad. Sci. U.S.A.* 2024). We note that this assay is a bulk measurement and, as such, cannot provide information on the temporal change of this ratio. Yet, we observed a net increase in *dnaAp* activity in *E. coli* Δ 3D, indicative of averagely lower levels of DnaA-ATP compared to the wild-type. This effect is indeed more pronounced at higher growth rates (Supplementary Figure 5B), due to the mentioned increased activity of Hda, as well as the one of *DARS2*. Still, we believe that the combinations of temporally high levels of DnaA-ATP and a decrease in titration generates the observed increase in DNA content during slow growth, as mentioned in the revised Discussion (see response to comment number 4 and 5 of Reviewer 2 for relevant new text). Moreover, the deletion of *datA* alone leads to a substantial decrease in *dnaAp* activity (i.e., high levels of DnaA-ATP; Supplementary Figure 6B), yet generates a lower average number of origins than *E. coli* Δ 3D (Figure 5C). For this reason, we exclude that perturbations of the switch mechanism alone can generate the severely re-initiating phenotype of *E. coli* Δ 3D.

Regarding the increase of mobility due to *oriC* partitioning, see our answer regarding the diffusion coefficient of bound DnaA proteins, appearing as immobile in our set-up. A recent study also observed how, after DNA replication is initiated, *oriC* displaces of only 0.4 μ m over the course of 20 min (Gras *et al.*, *Nat Commun*, 2024). Estimating the displacement for the frame time that we follow our PAmCherry2.1 molecules for (10 ms) leads to a value of $\sim 3 \cdot 10^{-4}$ nm, once again below our localisation precision of 30 nm. Both these observations mean that the chromosome appears as immobile for the time window employed in this study, discarding the possibility that the increase in DnaA mobility is due to its localisation of a fast-moving chromosomal locus. Regarding the changes in DnaA bound fraction upon deletion of single switch loci (Figure 5), the Discussion now reads:

The Δ *datA* mutation led to both an increase in DnaA-ATP/DnaA-ADP ratio (Supplementary Figure 6B) and in the bound fraction of DnaA (Figure 5). The increase in DnaA titration could affect

replication initiation by limiting the binding of DnaA-ATP to *oriC* and thus re-initiation events (Figure 7, top), consistent with the low fraction of cells with more than two origins we observed (Supplementary Figure 6A). Conversely, the deletion of *DARS1* resulted in a lower abundance of DnaA-ATP (Supplementary Figure 6B) and caused a decrease in the bound fraction of DnaA (Figure 5). In this sense, when the cell lacks a major DnaA reactivating component, a decrease in titration could lead to higher amounts of free DnaA-ATP and mitigate delays in replication initiation (Figure 7, middle), thus providing robustness.

10. Relating above: lines 235-231. If overinitiation occurs and the initiator titration model is valid, the elevated copies of the *oriC*-proximal region (as shown in Figure 2) should titrate DnaA more. Thus, logic of this part is inconsistent. The *datA*, *DARS1* and *DARS2* are not included in the DnaA box-dense region (Figure 2). Overall plot of this study is confusing.

Our response: As suggested by reviewer 2, we suspect that the stabilisation of DnaA mobility in *E. coli* $\Delta 3D$ after an initial phase of constant increase might be due to the initiation of DNA replication and thus the accumulation of new DnaA boxes. See our previous response for the cited text.

11. Relating above: Efficient titration of DnaA by *datA*, *DARS1* and *DARS2* is difficult to imagine because of the limited number of DnaA boxes. Indeed Ogawa et al, demonstrates that *datA* binds at most 30 DnaA molecules (Genes to Cells (2009) 14, 329–341). However, the authors should perform CHIP experiments to assess DnaA binding to these loci and *oriC* (as a control) in cells with different growth rates.

Our response: We agree with the Reviewer that chromatin immunoprecipitation would help in elucidating which DnaA boxes on the chromosome of *E. coli* are prevalently used in binding the initiator protein. However, the lack of a commercially available anti-DnaA antibody means that to perform efficient CHIP we would have to generate a new mutant of DnaA fused to an affinity tag, characterise its phenotype and then introduce a similar mutation in all genetic backgrounds of this study before performing CHIP in all the relevant growth conditions. Due to the extent of such plans, we believe that these experimental efforts are better suited for a follow-up study on this topic. As such, we now make a mention to them in the main text, reading:

Extending previous attempts of chromatin immunoprecipitation of *E. coli* DnaA⁷² to different growth rates and genetic backgrounds could also provide a better understanding of the usage of each box.

Reviewer #1

The authors have addressed most of the comments, however there are still a couple of issues with this new version of the manuscript.

Line 382: “Both the *fliC* and *glpD* genes are known to be positively regulated by DnaA. As such, it is surprising that mutations that lower the DnaA-ATP fraction and thus its capacity to oligomerise on DNA would result in their overexpression.”

The authors should cite the following study: Stringer et al (Microbiology 2024): “We did not detect an association of DnaA with previously described sites upstream of *aldA*, *glpD*, *fliC*, *proS* and *iraD*, or within *guaB*, strongly suggesting that the associated genes are not part of the DnaA regulon.”

The authors do not describe how they define the DnaA regulon. Did they compare the DnaA binding sites found by Stringer et al. in addition to their bioinformatics search? How many of these are in promoter regions? How do these compare to the protein expression profiles?

Our response: We modified the text to better acknowledge the insights obtained in the mentioned study. Regarding the considered composition of the DnaA regulon, we used the list of DnaA-affected genes mentioned in the introduction of the Stringer 2024 study. We now modified the text by acknowledging the new proposed genes:

“Recently, chromatin immunoprecipitation experiments performed during fast growth provided evidence that the *purH*, *rne* and *nrdD* genes are also regulated by DnaA⁷¹. Notably, only four of these genes (*mioC*, *polA*, *purH* and *dnaA*) fall within the chromosomal region with a high density of DnaA boxes (Figure 6A). In total, the DnaA boxes on the promoter region of these genes account for 14 of the 72 motifs found around *oriC*; the total increases to 22 when considering *oriC*.”

We further extended our analysis to the proteins expressed by the new genes listed (Rne, PurH and NrdD). We found that the abundance of Rne and PurH did not change in any of the tested conditions. On the other hand, NrdD was enriched in the *E. coli* Δ 3D during intermediate and fast growth and in *E. coli* Δ DARS2 during slow growth. Our original observation that *DARS2* deletion is sufficient to explain the observed changes in DnaA regulon is still valid also for the extended number of genes considered. We also acknowledge that *fliC* and *glpD* are potentially not regulated by DnaA. The text now reads:

“Yet, the notion that these genes are part of the DnaA regulon, at least during fast growth, has been recently called into question⁷³. We also detected an enrichment in the expression of the anaerobic ribonucleoside-triphosphate reductase NrdD during intermediate and fast growth. [...] Conversely, the deletion of *DARS2* induced a large change in *E. coli* expression profile, including the overexpression of *FliC*, *GlpD* and *NrdD*.”

In their rebuttal the authors say: Our analysis did not reveal major changes in DnaA abundance at different growth rates or in the different genetic backgrounds used in our study (Supplementary Figure

4D, 5D and 6C). Still, we refer to Zheng and colleagues (Nat. Micro., 2020) for a more precise estimation of these numbers across growth rates.

From Materials and Methods:

“Growth conditions to achieve different growth regimes were: M9 supplemented with 4 g/L glucose and 1x RPMI 1640 amino acids at 37 °C for fast growth regime; M9 supplemented with 4 g/L succinate and 1x RPMI 1640 amino acids at 30 °C for intermediate growth regime; and M9 supplemented with 4 g/L acetate at 37 °C for slow growth regime.”

I refer the authors to Figure 3 from Zheng et al as an example of how temperature might not be the best way to achieve different growth rates for comparison of DnaA activity. They should cite them and point out in their results section that the intermediate growth rate was obtained by lowering the temperature. In addition to the known temperature dependence of DnaA activity.

Our response: In the main text, we now acknowledge the potential effect of temperature, by citing the original study on the topic also referenced by Zheng *et al.*. After the new section on the temperature-dependent binding behaviour of DnaA, the text now reads:

“We further note that, together with changing the carbon source, intermediate growth was achieved by lowering the culture temperature to 30 °C (see Materials and Methods). Whereas changes in culture temperature affect growth rate, these changes have been reported to have a smaller impact on cellular mass^{10,54}. Still, *E. coli* cells exhibited the characteristic trend of larger cell size with increasing growth rate^{7,10}, with the measured areas progressively increasing from slow to fast growth conditions (Figure 4B).”.

The authors should cite the following study when comparing fast and slow growth conditions: Stepankiw, N., Kaidow, A., Boye, E., and Bates, D. (2009). The right half of the Escherichia coli replication origin is not essential for viability, but facilitates multi-forked replication. Mol Microbiol 74, 467–79. doi: 10.1111/j.1365-2958.2009.06877.x

Our response: We agree and added the suggested reference (number 78).

In the rebuttal the authors say:

“We also believe that DnaA cannot just be assimilated to any other DNA binding protein: its specific form of control (near-balanced expression) is not a trait confirmed for all DNA binding protein, as well as the non-monotonic dependence of its concentration on growth rate.”

A lot of transcription factors (40%) are negatively autoregulated to maintain a near-balanced expression, it is not unique of DnaA, and the growth rate dependence of its concentration is not related to the titration effect but the increase in origins per cell. See: Rosenfeld, N., Elowitz, M. B., and Alon, U. (2002). Negative autoregulation speeds the response times of transcription networks. J Mol Biol 323, 785–93. doi: 10.1016/S0022-2836(02)00994-4

Our response: We agree with the reviewer. To the best of our knowledge, we do not claim in the main text that these characteristics are unique to DnaA. Still, we modified the text to

include the suggested reference at the beginning of the Results section “Changes in DnaA ATP/ADP ratio or titration do not impact its function as transcription factor”. The text reads: “Additionally, negative autoregulation is a characteristic shared between DnaA and a variety of other *E. coli* transcription factor⁶²”.

“the synthesis of new titration sites per origin is set by the speed of DNA replication and is independent of growth rate. As a result, the synthesis rate of DnaA increases faster with growth rate than the synthesis rate of titration sites, leading to a reduction in the effective titration power of the chromosome.”

The synthesis of new titration sites is set by the speed of DNA replication, yes, but also by the number of sites. The number of origins per cell increases with increasing growth rate. It also increases at slow growth in the 3D strain. However, the bound fraction decreases and the average diffusion coefficient increases.

In the first round of review I made the following comment:

Line 311, the rate at which new DnaA boxes are replicated per origin is different from the speed at which new boxes are replicated -per cell- or -per volume- due to the increase in the origin to terminus ratio with growth rate.

Our response: We agree with the Reviewer, yet we are unsure as of how they would like us to acknowledge this fact in the text.

The -number- of titrating sites is dependent on growth rate. How does this affect the regulation of DnaA activity? It should at least be mentioned in the discussion.

Our response: We added this important consideration in the text. Regarding the behaviour of wild-type *E. coli*, we modified the Discussion to read:

“Whereas the number and rate of synthesis of DnaA boxes per origin is constant across growth rates, the same does not hold for the total number of DnaA boxes per cell. First, the DNA content of *E. coli* cells increases with increasing growth rates (Figure 4A). Second, cells eventually enter a regime of overlapping replication forks, in which the synthesis of new DNA proceeds through parallel replication rounds. The resulting increase in the total number of DnaA boxes per cell and in the rate at which these boxes are synthesised during progressively faster growth likely has an impact on the bound fraction of DnaA. Moreover, we hypothesise that the binding of DnaA to boxes scattered throughout the chromosome is also responsible for the overall high bound fractions, with pairs of closely spaced boxes⁷³ and specific loci (e.g. *mioC*⁷⁵ and *oriC*⁷⁶⁻⁷⁸) mediating more stable forms of binding”.

Regarding *E. coli* Δ3D, we modified the text in the Result section to read:

“During slow growth, we observed a decreased bound fraction and increased mobility of DnaA, despite the higher DNA content indicating that more DnaA boxes are present compared to the wild-type condition. These observations lead us to believe that the contribution of the three switch control loci on titration is substantial under these conditions.”.

Regarding the single deletion mutants, we already addressed how the changes in DNA content might reflect on the bound fraction of DnaA via increases or decreases in the number of DnaA boxes.

Reviewer #2

The authors have addressed all our concerns.

Reviewer #3

Overall the authors made significant improvements by additional experiments and revising the text. However, the following points should be further considered.

1. Structure of DnaA-mCherry. Descriptions in the main text and Figure 3A are inconsistent. Unlike the explanation in the text (lines 171-173), Figure 3A and its legend indicated deletion of DnaA domain IIIb, in addition to DnaA domain II. Deletion of DnaA domain IIIb should be lethal. Precise structure of the dnaA-mCherry region on the genome and strain construction process should be described.

Our response: Please note that we already provide an explanation of the construction process in the Materials and Methods section (“Creation of the *dnaA-PAFP* fusion mutants”). To bring this more clearly to the reader, we now added a brief explanation in the Results section, reading:

“To this end, we generated several fusions of DnaA in domain II^{48,49} with photoactivatable or photoswitchable fluorescent proteins. The fusions were created by replacing amino acids 87-106^{48,49} with the different fluorescent proteins (see Methods for details on the construction process). We ultimately selected PAmCherry2.1⁴⁴ for its low impact on DNA content (Supplementary Figure 4B).”

Additionally, we modified the caption of Figure 3 to clarify that linker domains were still present in the final *dnaA-PAmCherry2.1* construct, but are not indicated in the figure for visualisation purposes. The text reads:

“The domains of DnaA are indicated with Roman numerals. Domain II and IIIb are depicted in light grey, yet are not specifically indicated for visualisation purposes due to their small size. On the bottom, the codon numbers of each domain of the wild-type DnaA are indicated, not accounting for the PAmCherry2.1 insertion.”

2. As I mentioned in the previous comments, I am not still fully convinced in the idea that “The decreased bound fraction and increased mobility of DnaA suggest that the contribution of the three switch control loci on titration is substantial during slow growth”. (lines 295-296). Suggested experiments were not performed. This might be OK, but as the result, the idea was not reinforced.

There should be a reasonable possibility that due to the lack of *datA*, the rate of DnaA-ATP hydrolysis which is caused only by RIDA and the intrinsic ATPase is somehow decreased, causing re-initiations. Although the overall level of the ATP form is decreased in the $\Delta 3D$ mutant, the ATP-DnaA level should be elevated temporarily even in a delayed manner during the cell cycle. If the decreasing rate of the level is slowed, re-initiations can occur. These ideas and the results by Ogawa et al. (2009) (as indicated in the e previous comments) should be discussed in the text.

Our response: We refer to our detailed response to the last comment of reviewer 1 for our reasoning that the decrease in DnaA bound fraction in *E. coli* $\Delta 3D$ during slow growth could be attributed to the titration power of the three control loci.

We do agree that our measurements of the state of DnaA rely on bulk assays and, as such, cannot reveal transient peaks in the levels of either of the two forms. For this reason, at the end of our Discussion we mention how microfluidics-based cultivation strategy would be best posed to investigate this type of dynamics, as already reported by Iuliani and colleagues (*Sci Adv.*, 2024). Earlier in the Discussion section, we indeed considered transient high abundance of DnaA-ATP while providing our explanation of the behaviour of *E. coli* $\Delta 3D$. Yet, the frequently re-initiating phenotype is unique of the *E. coli* $\Delta 3D$ strain. If a decrease in inactivating power would be the sole cause of the re-initiations, then *E. coli* $\Delta datA$ should show a similar, if not aggravated behaviour, due to the fact that cells lack *datA* while the major reactivating locus *DARS1* is still present. Yet, they mostly cycle between 1 and 2 origins per cell and have an asynchrony index similar to the wild-type. We expanded our Discussion to acknowledge these points, now reading:

“It is possible that the decreased rate of DnaA-ATP hydrolysis caused by the absence of *datA* is sufficient to explain the re-initiating phenotype of the triple deletion mutant. It should further be noted that the *dnaAp* activity assay employed here relies on bulk measurements that could mask transient peaks in the active form of DnaA in key moments of the cell cycle. However, if high DnaA-ATP levels would be the sole determinant of re-initiation frequency, *E. coli* $\Delta datA$ would show the highest DNA content out of all the strains examined during slow growth. In *E. coli* $\Delta datA$, the effect of the absence of the inactivating locus is further aggravated by the presence of the reactivating *DARS1* locus, leading to the highest DnaA-ATP levels observed (Supplementary Figure 6B). Yet, *E. coli* $\Delta datA$ showed a lower number of origins than *E. coli* $\Delta 3D$ (Figure 5C) and near-wild-type values of asynchrony index (Supplementary Figure 6D). We believe that this surprising difference can also serve as indirect evidence for the role of DnaA titration on the coordination of DNA replication.”.